

# Bubbling saddles of the gravitational index

**Davide Cassani[1], Alejandro Ruipérez[3] and Enrico Turetta[1,2]**

**1** INFN, Sezione di Padova, Via Marzolo 8, 35131 Padova, Italy
**2** Dipartimento di Fisica e Astronomia "Galileo Galilei", Università di Padova,
Via Marzolo 8, 35131 Padova, Italy
**3** Dipartimento di Fisica, Università di Roma "Tor Vergata" & Sezione INFN Roma 2,
Via della Ricerca Scientifica 1, 00133, Roma, Italy

## Abstract

We consider the five-dimensional supergravity path integral that computes a supersymmetric index, and uncover a wealth of semiclassical saddles with bubbling topology. These are complex finite-temperature configurations asymptotic to $S^1 \times \mathbb{R}^4$, solving the supersymmetry equations. We assume a $U(1)^3$ symmetry given by the thermal isometry and two rotations, and present a general construction based on a rod structure specifying the fixed loci of the $U(1)$ isometries and their three-dimensional topology. These fixed loci may correspond to multiple horizons or three-dimensional bubbles, and they may have $S^3$, $S^2 \times S^1$, or lens space topology. Allowing for conical singularities gives additional topologies involving spindles and branched spheres or branched lens spaces. As a particularly significant example, we analyze in detail the configurations with a horizon and a bubble just outside of it. We determine the possible saddle-point contribution of these configurations to the gravitational index by evaluating their on-shell action and the relevant thermodynamic relations. We also spell out two limits leading to well-defined Lorentzian solutions. The first is the extremal limit, which gives the known BPS black ring and black lens solutions. The on-shell action and chemical potentials remain well-defined in this limit and should thus provide the contribution of the black ring and black lens to the gravitational index. The second is a limit leading to horizonless bubbling solutions, which have purely imaginary action.

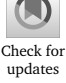

# 1 Introduction

The gravitational path integral

$$Z = \int \mathcal{D}g_{\mu\nu} \dots \, \mathrm{e}^{-I[g_{\mu\nu}, \dots]}, \tag{1.1}$$

introduced by Gibbons and Hawking almost fifty years ago [1], lies at the heart of Euclidean Quantum Gravity. As such, it has played a central role in black hole thermodynamics, quantum cosmology, and holography. Recently, it has found new applications in the derivation of the Page curve through replica wormholes [2,3] and the description of strong quantum effects in near-extremal black holes [4,5], among others. In supergravity, one can impose supersymmetric boundary conditions and attempt to evaluate the path integral beyond the semiclassical approximation, possibly even exactly if the theory admits a UV completion. This idea is particularly appealing in the context of black hole microstate counting, since the path integral with supersymmetric boundary conditions should then compute an index (see e.g. [6] for a review). When the boundary conditions are those satisfied in the near-horizon of BPS black holes, and thus the geometry asymptotes to an AdS$_2$ factor, the path integral calculates a microcanonical

index and is directly related to the black hole degeneracy [7,8]. Boosted by the use of supersymmetric localization, initiated in [9], the evaluation of the supergravity path integral with these microcanonical boundary conditions has now reached a very high accuracy [10], despite many complications.

In this paper, we work in pure five-dimensional supergravity and impose supersymmetric boundary conditions defining a *grand-canonical* supersymmetric index. We will work in the semiclassical approximation and explore the expansion over the saddles given by supersymmetric solutions to the classical equations of motion with finite action.

Let us set the stage by briefly recalling how one can define a supergravity path integral calculating the supersymmetric index of interest. We consider a five-dimensional asymptotically flat spacetime where one can define the energy $E$, two independent angular momenta $J_1, J_2$, and an electric charge $Q$. These conserved quantities generate evolution along Lorentzian time $t$, translations along $2\pi$-periodic angular coordinates $\phi_1, \phi_2$, and gauge transformations of the supergravity abelian vector field, respectively. Now we Wick-rotate $t = -i\tau$ and compactify the Euclidean time $\tau$ to a circle $S^1$ of period $\beta$, which has a standard interpretation as inverse temperature. We specify the angular coordinate identifications, including those for a revolution around the thermal $S^1$, as

$$(\tau, \phi_1, \phi_2) \sim (\tau + \beta, \phi_1 - i\omega_1, \phi_2 - i\omega_2) \sim (\tau, \phi_1 + 2\pi, \phi_2) \sim (\tau, \phi_1, \phi_2 + 2\pi), \quad (1.2)$$

where $\omega_i$, $i = 1, 2$, are dimensionless angular velocities, related to the usual angular velocities $\Omega_i$ appearing in black hole thermodynamics as $\omega_i = \beta\Omega_i$. We can also introduce an electrostatic potential $\Phi$ via the holonomy of the supergravity abelian gauge field around the asymptotic circle, $\beta\Phi = -i\int_{S^1_\infty} A$. With these boundary conditions, together with suitable Dirichlet-like boundary conditions for the fields in the theory, the gravitational path integral (1.1) can be seen as a thermal partition function in the grand-canonical ensemble, depending on the potentials, $Z = Z(\beta, \omega_1, \omega_2, \Phi)$. When the theory is embedded in a UV-complete theory such as string theory, this partition function may be interpreted as a trace over microstates of the form

$$Z(\beta, \omega_1, \omega_2, \Phi) = \text{Tr}\, e^{-\beta(E - \Phi Q) + \omega_1 J_1 + \omega_2 J_2}. \quad (1.3)$$

We choose supersymmetric boundary conditions, so that the partition function above reduces to a refined Witten index [11], receiving contributions from supersymmetric states only. In order to achieve this, we take a supercharge $\mathcal{Q}$ satisfying the superalgebra relations

$$\{\mathcal{Q}, \bar{\mathcal{Q}}\} = E - \sqrt{3}Q, \qquad [J_i, \mathcal{Q}] = \frac{1}{2}\mathcal{Q}, \quad i = 1, 2, \qquad [Q, \mathcal{Q}] = 0. \quad (1.4)$$

We need the argument of the trace (1.3) to anticommute with the supercharge, so that pairs of bosonic and fermionic states related to each other by the action of $\mathcal{Q}$ give opposite contributions and cancel out. In the present setup, this is obtained by demanding that the angular velocities satisfy

$$\omega_1 + \omega_2 = 2\pi i \qquad (\text{mod } 4\pi i). \quad (1.5)$$

Solving (1.5) for $\omega_1$ and noting that $e^{\pm 2\pi i J_1} = (-1)^{\mathsf{F}}$, where $\mathsf{F}$ is the fermion number operator, we can write $e^{\omega_1 J_1 + \omega_2 J_2} = (-1)^{\mathsf{F}} e^{\omega_2 (J_2 - J_1)}$. Also using the expression for $E$ from the first in (1.4), and redefining the electrostatic potential as $\varphi = \beta(\Phi - \sqrt{3})$, the trace (1.3) eventually takes the form of a refined Witten index,

$$\mathcal{I}(\omega_2, \varphi) = \text{Tr}\, (-1)^{\mathsf{F}} e^{-\beta\{\mathcal{Q}, \bar{\mathcal{Q}}\}} e^{\omega_2(J_2 - J_1) + \varphi Q}. \quad (1.6)$$

By a standard argument, this partition function only receives contributions from states preserved by the action of $\mathcal{Q}, \bar{\mathcal{Q}}$ and is independent of $\beta$. We will denote the gravitational path

integral with chemical potentials $\omega_1, \omega_2, \varphi$ satisfying the constraint (1.5) the *gravitational index*.

Although the index is a much simpler object than the non-supersymmetric thermal partition function, computing it exactly would be very difficult for a gravitational theory like the one at hand. We will thus study this gravitational partition function in the semiclassical approximation, where it is expected to be dominated by saddles given by regular solutions to the classical equations of motion satisfying the assigned boundary conditions. We would like to determine the structure of the saddles, their contribution to the path integral, and the different phases arising when dialing the chemical potentials. Saddles of the index should be supersymmetric solutions with finite $\beta$. These non-extremal supersymmetric solutions are not well-defined in Lorentzian signature. However, it has recently been appreciated that they can arise as complex field configurations, which are naturally defined in Euclidean signature [12–14]. We nevertheless expect that upon taking a $\beta \to \infty$ limit, some of these finite-$\beta$ configurations should be connected with supersymmetric extremal solutions having a good interpretation as Lorentzian supersymmetric solutions. This provides an effective prescription to assign a finite on-shell action and chemical potentials to the latter.

Since we eventually want to make the connection with supersymmetric solutions in Lorentzian signature, it may be useful to briefly summarize the state of the art in the Lorentzian setup. In five dimensions, the celebrated uniqueness theorems that hold in four dimensions do not apply. This means that specifying the conserved quantities measured at infinity (mass, angular momenta, electric charge) is not enough for determining an asymptotically flat solution with an event horizon. Relatedly, horizons can have more general topology than in four dimensions. In fact, the first example of a regular solution violating black hole uniqueness was provided by a black ring, namely a solution with an $S^2 \times S^1$ horizon topology, which can have the same conserved quantities measured at infinity as a black hole with $S^3$ horizon topology [15]. Let us focus on minimal five-dimensional supergravity, and consider asymptotically flat field configurations having two independent axial symmetries (leading to the angular momenta $J_1, J_2$) in addition to time translation invariance, so that the symmetry group includes $\mathbb{R} \times U(1) \times U(1)$. With these assumptions, it has been shown that the cross-section of supersymmetric horizons can have the topology of $S^3$, $S^2 \times S^1$, or the lens space $L(n, 1) \simeq S^3/\mathbb{Z}_n$ [16]. Explicit supersymmetric solutions with these horizon topologies have been constructed: the $S^3$ horizon is realized by the BMPV black hole [17], the $S^2 \times S^1$ topology is realized by the black ring of [18], while the $L(n, 1)$ topology is realized by the black lens solutions of [19,20].

There are also regular supersymmetric solutions with non-trivial topology and charges despite the absence of a horizon, see e.g. [21–24]. These are called topological solitons, or "bubbling" solutions, due to the presence of a "foam" of two-cycles threaded by flux of the supergravity gauge field, and play a central role in the fuzzball and microstate geometry programs [25–27]. Horizons can also coexist with bubbling topology in the domain of outer communication; this was first emphasized in [28], where a solution with a bubble outside a black hole horizon was presented. More generally, it is possible to add a black hole horizon to a background topological soliton containing many bubbles. Additionally, one can construct solutions with multiple disconnected horizons, the first examples being the concentric black rings of [29]. This variety of combinations is enabled by the linearity of the supersymmetry equations, which allows one to sum up the functions that determine the different horizons and bubbles (much in the same way as how the Majumdar-Papapetrou multi-center solution is obtained from the extremal Reissner-Nordström black hole).

Besides the conserved quantities at infinity, the information needed to specify a solution is effectively encoded in the *rod structure*, based on a formalism originally developed for pure gravity in [30–32]. Given the two-dimensional orbit space obtained by modding out the five-dimensional spacetime by the action of the $\mathbb{R} \times U(1) \times U(1)$ isometries, the rod structure consists

of specifying the isometries which degenerate and their fixed loci in the orbit space. These fixed loci can be arranged into consecutive segments – the rods – forming an infinite line. If it is a spacelike U(1) that degenerates, then the corresponding rod together with the orbits of the surviving U(1) forms a fixed two-cycle in the spatial section of the geometry. Depending on whether the latter U(1) also degenerates at the endpoints of the rod or not, the cycle can have the topology of a two-sphere, a disc, or a tube.[1] If, instead, the Killing vector becoming null is timelike, then the corresponding rod together with the axial U(1) × U(1) forms an event horizon, whose topology depends on which of these axial U(1)'s shrinks at the rod endpoints. Extremal horizons (including all supersymmetric horizons in Lorentzian signature) correspond to the limiting case where the horizon rod has vanishing length, namely they are given by special isolated points on the rod axis.

In order to specify the solution, one should supplement the rod structure with the magnetic flux of the supergravity gauge field through the non-contractible fixed loci, or alternatively provide the conjugate potentials. A classification of asymptotically flat, biaxisymmetric supersymmetric solutions to minimal supergravity using the rod structure formalism was given in [16].[2] These magnetic fluxes and their conjugate potentials have been argued to also play a role in black hole thermodynamics, although they are not associated to an asymptotic symmetry: a generalization of the first law of black hole mechanics in a background topological soliton has been proven in [34, 35].

In this paper, we join the two threads reviewed above and investigate how the cornucopia of supersymmetric Lorentzian solutions relates to saddles of the gravitational index. We are thus led to study finite-temperature extensions of the extremal solutions, where by "finite temperature" we mean that the parameter $\beta$ introduced in (1.2) is finite, and may even be complex. We also assume the solution has $U(1)^3$ symmetry, so that we can rely on the rod structure illustrated above. Our task is thus to consider supersymmetric horizons containing a rod of finite size, and study the associated regularity conditions.[3] This is a non-trivial task since the solutions have a complex metric, however we will still be able to address their topology. The extremal limit, leading to solutions that have a Lorentzian counterpart, can then be understood as a limit where the rod contracts to a point.

Our study shows that the set of saddles of the gravitational index with fixed $\beta$, $\omega_1$, $\omega_2$, $\varphi$ is much richer than the class identified as "black hole saddles" in [38–42] (see also [14, 43–45] for related studies in four dimensions). By combining different types of rods, namely *horizon rods* and *bubbling rods*, we provide a general framework for constructing solutions with multiple disconnected horizons of different topologies and an arbitrary number of bubbles. These are distinguished by the number of times the orbits of the associated rod vector wind around the three basis U(1)'s in the geometry. The winding numbers needed to specify this information characterize the topology of the fixed loci. In particular, if the orbits of the rod vector wind around the thermal circle, then the fixed locus is a horizon, while if they do not, then it is a three-dimensional bubble. In this framework, the black hole saddles of [38–42] correspond to the case where there is a unique compact rod, and the orbits of the rod vector wind just once around the thermal circle. A simple generalization of this case is one where the orbits wind multiple times around the thermal circle as well as around an axial circle; this corresponds to a supersymmetric orbifold of the black hole solution, and provides a saddle which only makes sense in the complexified setup.

---

[1]When the solutions are embedded in string theory, one may also allow for orbifolds of the sphere and disc topologies, which yield conical singularities.

[2]Supersymmetric solutions with just one axial symmetry have been studied in [33]. Although we will not consider such solutions in this paper, we observe that they share many features with the biaxisymmetric ones, so we anticipate that at least part of our analysis will go through in that case as well.

[3]In the context of minimal five-dimensional supergravity, the rod formalism including non-supersymmetric, non-extremal horizons is discussed e.g. in [36, 37].

We work out the topology of the rod vector fixed loci within the five dimensional geometry, including the Euclidean time circle fibration in the description. We find that in general the topology is the one of an $L(\mathrm{p}, \mathrm{q})$ lens space, and allows for $S^3$ and $S^2 \times S^1$ as special cases. The integers p, q characterizing the lens space are determined by the winding numbers associated with the rod under study and the two adjacent ones, in a way that we discuss in detail. One can also consider non-freely-acting orbifolds of the above topologies, which produce horizons and bubbles with conical singularities, namely branched spheres, branched lens spaces and products of $S^1$ and a spindle.[4]

The characterization of the solutions also requires to specify the electric flux through the three-dimensional bubbles or, alternatively, certain conjugate potentials that we define and compute via equivariant localization. When the abelian gauge group is compact, namely U(1), this leads us to introduce an additional integer for each compact rod. For bubbling rods, the potential conjugate to the electric flux is now quantized and directly specified by the additional integer. For horizon rods, the integer enters in the linear map relating the horizon electrostatic potential and the chemical potential $\varphi$ appearing in the index (1.6).

We then focus on the possible saddle-point contribution of our solutions to the gravitational index. This requires computing the supergravity on-shell action and expressing it in terms of the variables appearing in the index. For the case with just one horizon but arbitrarily many bubbles, we infer a formula for the on-shell action. Relatedly, we study the thermodynamics of our solutions in the presence of non-trivial topology outside the horizon, focussing on the contribution of the electric flux through the three-dimensional bubbles.

As particularly significant examples of our general construction, we study in detail the class of solutions with two compact rods, one being a horizon rod and the other being a bubbling rod. This includes the supersymmetric non-extremal versions of the BPS black ring and black lens of [18,19]. By a direct computation we evaluate the on-shell action of these solutions. This requires some care, since the gauge Chern-Simons term of five-dimensional supergravity has to be evaluated patchwise. The expression we obtain for the on-shell action reads

$$I = \frac{\pi}{12\sqrt{3}} \left[ \frac{\varphi^3}{\omega_1 \omega_2} - \frac{\left(p_1 \varphi - \omega_2 \Phi^{I_1}\right)^3}{p_1^2 \omega_2 \left(p_1 \omega_1 + (p_1 - 1)\omega_2\right)} \right], \qquad (1.7)$$

where $p_1 \in \mathbb{Z}$ and $\Phi^{I_1}$ is the potential conjugate to the electric flux through the bubble. We verify that by taking the extremal $\beta \to \infty$ limit of the solutions thus constructed and demanding regularity outside the horizon we land indeed on the BPS black ring and black lens. These correspond to the cases $p_1 = 1$ and $p_1 = -1$ in (1.7), respectively. The on-shell action remains well-defined in the limit and may thus be seen as the saddle-point contribution of these BPS solutions to the index. As a check of our on-shell action formula, we recover the correct Bekenstein-Hawking entropy of the black ring and black lens by a Legendre transformation.

In addition to the extremal $\beta \to \infty$ limit, we consider a different limit leading to Lorentzian solutions, which involves sending $\beta \to 0$. This limit transforms horizon rods into bubbling rods, and requires that $\omega_2$, $\varphi$ satisfy suitable quantization conditions. We thus obtain horizonless bubbling solutions, belonging to the family found in [21,22]. In this way, we can assign chemical potentials and an on-shell action to these solutions as well. However, these quantities are purely imaginary, raising the question of whether the gravitational index can be extended to this regime and the horizonless solutions be regarded as acceptable saddles.

The rest of the paper is organized as follows. In section 2 we introduce some general features of supersymmetric solutions to pure five-dimensional supergravity having the assumed symmetry. In section 3 we illustrate the rod structure and analyze the topology of the rod

---

[4]Starting with [46,47], geometries based on spindles have recently become part of supersymmetric holography. Quantum field theories on branched spheres and branched lens spaces have been studied e.g. in [48–51].

vector fixed loci, distinguishing between horizon rods and bubbling rods. A summary of the general classification which follows from this analysis is given in section 3.1. We also illustrate the global properties of the gauge field and its role in the thermodynamics. In section 4 we revisit the two-center black hole solution emphasizing the need to complexify it, and discussing its shifts and orbifolds satisfying the same boundary conditions. In section 5 we discuss in detail the three-center solution, providing the on-shell action and specifying the Lorentzian limits leading to the black ring, the black lens and a horizonless topological soliton. We draw our conclusions and discuss some open questions in section 6. Three appendices contain some details on the derivation of the non-extremal solution, a brief account on lens spaces, and the explicit evaluation of the on-shell action for the three-center solution.

## 2 Supersymmetric solutions with biaxial symmetry

The bulk Euclidean action of pure five-dimensional supergravity reads

$$I_{\text{bulk}} = -\frac{1}{16\pi} \int \left( R \star_5 1 - \frac{1}{2} F \wedge \star_5 F + \frac{i}{3\sqrt{3}} A \wedge F \wedge F \right), \tag{2.1}$$

where $A$ is an abelian gauge field, and $F = \mathrm{d}A$.

We start by outlining our strategy for obtaining supersymmetric solutions to Euclidean (complexified) supergravity. In suitable conventions, the Killing spinor equation reads

$$\left[ \nabla_\mu - \frac{i}{8\sqrt{3}} \left( \gamma_\mu{}^{\nu\rho} - 4\delta_\mu^\nu \gamma^\rho \right) F_{\nu\rho} \right] \epsilon = 0, \tag{2.2}$$

where $\epsilon$ is a Dirac spinor. In the general framework of Euclidean supergravity, all degrees of freedom should be doubled, i.e. the metric and the gauge field become complex, while the gravitino and its Lorentzian charge-conjugate should be seen as independent. It follows that the supersymmetry spinor parameter and its Lorentzian charge-conjugate should also be taken as independent in Euclidean signature. We denote by $\epsilon$ and $\tilde{\epsilon}$ these two a priori independent Dirac spinors. It turns out that the Lorentzian charge-conjugate equation of the Killing spinor equation (2.2) takes exactly the same form, so $\epsilon$ and $\tilde{\epsilon}$ need to satisfy the very same Killing spinor equation. This implies that if we have a Lorentzian supersymmetric solution depending on some parameters, any analytic continuation of those parameters to the complex domain will give a field configuration solving the doubled Killing spinor equations, as well as the equations of motion following from (2.1), where the fields are assumed complex. This will be our working framework, namely we will consider complex solutions which arise as analytic continuations of Lorentzian supersymmetric solutions. We will not impose any global a priori condition on these Lorentzian supersymmetric solutions, these will be studied after the analytic continuation.

According to the classification of [52], supersymmetric timelike solutions with an extra symmetry commuting with the supercharges take the form

$$\begin{aligned}
\mathrm{d}s^2 &= f^2 \left( \mathrm{d}\tau + i\omega_\psi (\mathrm{d}\psi + \chi) + i\breve{\omega} \right)^2 + f^{-1} \left[ H^{-1} (\mathrm{d}\psi + \chi)^2 + H \, \mathrm{d}s^2_{\mathbb{R}^3} \right], \\
\frac{i}{\sqrt{3}} A &= -f \left( \mathrm{d}\tau + i\omega_\psi (\mathrm{d}\psi + \chi) + i\breve{\omega} \right) + iH^{-1} K (\mathrm{d}\psi + \chi) + i\breve{A} + \mathrm{d}\alpha,
\end{aligned} \tag{2.3}$$

where $\tau = it$ is the Euclidean time parameterizing the orbits of the Killing vector $\partial_\tau$, arising as a Killing spinor bilinear, while $\psi$ parameterizes the orbits of the additional isometry preserving the Killing spinor. The four-dimensional metric in the brackets is a hyperkähler basis of Gibbons-Hawking form. Moreover, $\mathrm{d}\alpha$ is a closed one-form to be discussed later, and all

other functions and one-forms appearing in (2.3) are determined by four harmonic functions $(H, K, L, M)$ on $\mathbb{R}^3$. Specifically, the functions $\omega_\psi$ and $f$ are fixed as

$$\omega_\psi = \frac{K^3}{H^2} + \frac{3}{2}\frac{KL}{H} + M, \qquad f^{-1} = \frac{K^2}{H} + L,\tag{2.4}$$

while $\chi, \breve{\omega}, \breve{A}$ are local one-forms on $\mathbb{R}^3$ satisfying the equations

$$\star_3 d\chi = dH, \qquad \star_3 d\breve{\omega} = HdM - MdH + \frac{3}{2}(KdL - LdK), \qquad \star_3 d\breve{A} = -dK,\tag{2.5}$$

with $\star_3$ the Hodge dual in $\mathbb{R}^3$. These are the same conditions appearing in [52]. As customary, we shall assume a multi-center ansatz for the harmonic functions,

$$H = h_0 + \sum_{a=1}^{s}\frac{h_a}{r_a}, \qquad iK = k_0 + \sum_{a=1}^{s}\frac{k_a}{r_a}, \qquad L = \ell_0 + \sum_{a=1}^{s}\frac{\ell_a}{r_a}, \qquad iM = m_0 + \sum_{a=1}^{s}\frac{m_a}{r_a},\tag{2.6}$$

with $s$ being the number of sources – usually called "centers" in this context – on the $\mathbb{R}^3$ base-space, and $r_a$ representing the distance to each of the centers. In Lorentzian signature, all harmonic functions would be taken real; in particular $K$ and $M$ would be expressed in terms of coefficients $k_0 = ik_0$, $k_a = ik_a$, $m_0 = im_0$, $m_a = im_a$, with real $k_0, k_a, m_0, m_a$. Here, we have chosen to work with the analytically continued parameters $k_0, k_a, m_0, m_a$, which satisfy the condition that if all of them are real, then the Euclidean metric in (2.3) is real. However, below we will also allow for complex choices of these parameters.

In this paper we shall restrict to solutions admitting an additional isometry, rotating a plane in the $\mathbb{R}^3$ base. This implies that all the centers are placed on the rotation axis, which we identify with the $z$-axis. The location of the $a$-th center is thus specified by its position $z = z_a$ on this axis. In addition to the natural cylindrical coordinates on $\mathbb{R}^3$ given by $(\rho, \phi, z)$, where $(\rho, \phi)$ are polar coordinates in the $\mathbb{R}^2$-plane orthogonal to the $z$-axis, we will also find it useful to introduce sets of spherical coordinates: we will denote by $(r, \theta, \phi)$ the standard spherical coordinates centered at the origin of $\mathbb{R}^3$, while we will use $(r_a, \theta_a, \phi)$ for those with origin at each of the Gibbons-Hawking centers. They are related to the cylindrical coordinates by

$$\rho = r\sin\theta, \quad z = r\cos\theta \quad \leftrightarrow \quad r = \sqrt{\rho^2 + z^2}, \quad \cos\theta = \frac{z}{\sqrt{\rho^2 + z^2}},\tag{2.7}$$

and

$$\rho = r_a\sin\theta_a, \quad z = z_a + r_a\cos\theta_a \quad \leftrightarrow \quad r_a = \sqrt{\rho^2 + (z - z_a)^2}, \quad \cos\theta_a = \frac{z - z_a}{\sqrt{\rho^2 + (z - z_a)^2}}.\tag{2.8}$$

In terms of these coordinates, one readily finds that the following expressions solve the equations for $\chi$ and $\breve{A}$:

$$\chi = \sum_{a=1}^{s} h_a\cos\theta_a\, d\phi, \qquad i\breve{A} = -\sum_{a=1}^{s} k_a\cos\theta_a\, d\phi.\tag{2.9}$$

Solving the equation for $\breve{\omega}$ requires a bit more effort. We relegate the details to appendix A, and present here just the final result, which is:

$$i\breve{\omega} = \sum_{a} iw_a\cos\theta_a d\phi + \sum_{a}\sum_{b>a}\frac{C_{ab}}{\delta_{ab}}(1 + \cos\theta_a)\left(1 - \frac{r_a + \delta_{ab}}{r_b}\right)d\phi,\tag{2.10}$$

where

$$iw_a = h_0 m_a - m_0 h_a + \frac{3}{2}(k_0 \ell_a - \ell_0 k_a) - \sum_{b \neq a} \frac{C_{ab}}{|\delta_{ab}|}, \tag{2.11}$$

$$C_{ab} = h_a m_b - h_b m_a + \frac{3}{2}(k_a \ell_b - k_b \ell_a), \tag{2.12}$$

and $\delta_{ab} \equiv z_a - z_b$. In deriving (2.10), it has been assumed that $\delta_{ab} > 0$ if $a < b$. Thus, only the absolute value of $\delta_{ab}$ appears in (2.10). We have also fixed an integration constant such that

$$\sum_a w_a = 0. \tag{2.13}$$

**Asympotic flatness condition.** We shall consider solutions that asymptote to $S^1 \times \mathbb{R}^4$, where $S^1$ is parameterized by the $\tau$ coordinate. This implies, in particular, that the Gibbons-Hawking base space must be asymptotically $\mathbb{R}^4$. This is achieved imposing

$$h_0 = 0, \qquad \sum_a h_a = 1, \tag{2.14}$$

and that the metric functions $f$ and $\omega_\psi$ tend to constant values at infinity (1 and 0 respectively). The latter conditions fix the constant terms of the remaining harmonic functions as

$$k_0 = 0, \qquad \ell_0 = 1, \qquad m_0 = -\frac{3}{2}\sum_a k_a. \tag{2.15}$$

It is convenient to recall that the metric and gauge field strength are invariant under the following shifts of the harmonic functions,

$$\begin{aligned}
K &\to K + \mu H, \\
L &\to L - 2\mu K - \mu^2 H, \\
M &\to M - \frac{3}{2}\mu L + \frac{3}{2}\mu^2 K + \frac{1}{2}\mu^3 H,
\end{aligned} \tag{2.16}$$

where $\mu$ is a constant. We fix this redundancy by imposing

$$\sum_a k_a = 0. \tag{2.17}$$

**Cap condition.** One of the main advantages of working in Euclidean signature is that the solution one obtains after Wick-rotating a thermal black hole smoothly caps off at the position of the horizon, thereby excising the singularity. Since the goal of this paper is to construct gravitational solitons of this kind, we have to demand that the solution has a cap, instead of the infinite $AdS_2$ throat characteristic of the near-horizon of supersymmetric and extremal black holes. As it turns out, this is equivalent to demanding

$$\lim_{r_a \to 0} f^{-1} = \mathcal{O}(r_a^0), \qquad \lim_{r_a \to 0} \omega_\psi = \mathcal{O}(r_a^0), \tag{2.18}$$

at each center, which in turn implies that the coefficients of the harmonic functions $L$ and $M$ are determined in terms of those of $H$ and $K$ by

$$\ell_a = \frac{k_a^2}{h_a}, \qquad m_a = -\frac{k_a^3}{2h_a^2}. \tag{2.19}$$

Not surprisingly, one can verify that these conditions are also satisfied by the smooth horizonless "microstate" geometries constructed in [21, 22]. Indeed, a central feature of these solutions is the absence of an AdS$_2$ throat. We further discuss the relation between our solutions and those of [21, 22] in the explicit examples presented in sections 4 and 5.

Additionally, in this class of solutions the following property holds

$$-h_a \lim_{r_a \to 0} \omega_\psi = w_a,\tag{2.20}$$

which will be relevant when analyzing regularity of the metric around the centers in the next section.

**The solution so far.** Imposing the above conditions, the harmonic functions specialize to

$$H = \sum_a \frac{h_a}{r_a}, \qquad iK = \sum_a \frac{k_a}{r_a}, \qquad L = 1 + \sum_a \frac{k_a^2}{h_a r_a}, \qquad iM = -\frac{1}{2}\sum_a \frac{k_a^3}{h_a^2 r_a},\tag{2.21}$$

with $\sum_a h_a = 1$ and $\sum_a k_a = 0$. The solution then depends on $2s - 2$ harmonic function coefficients, as well as $s - 1$ distances between the centers. In the Euclidean setup we will let these parameters take complex values. The coefficients (2.11), (2.12) determining the local one-form $\breve{\omega}$ given in (2.10) boil down to

$$iw_a = -\frac{3}{2}k_a - \sum_{b \neq a} \frac{C_{ab}}{|\delta_{ab}|},\tag{2.22}$$

and

$$C_{ab} = \frac{h_a h_b}{2}\left(\frac{k_a}{h_a} - \frac{k_b}{h_b}\right)^3.\tag{2.23}$$

The coefficients $w_a$ satisfy $\sum_a w_a = 0$ and also determine the function $\omega_\psi$ near to the $a$-th center via (2.20).

**Asymptotic charges.** The mass and angular momenta, associated with the symmetries generated by the Killing vectors $\partial_t$, $\partial_\phi$ and $\partial_\psi$, can be read from the asymptotic expansion of the Lorentzian metric. To this aim, we introduce the coordinates

$$\tilde{r}^2 = 4r, \qquad \phi_1 = \frac{\phi - \psi}{2}, \qquad \phi_2 = \frac{\phi + \psi}{2},\tag{2.24}$$

and use the Lorentzian time $t = -i\tau$. The asymptotic expansion of the relevant components of the metric then reads

$$g_{tt} = -1 + \frac{8\sum_a \ell_a}{\tilde{r}^2} + \dots,$$

$$g_{t\phi_1} = \frac{4i\sum_a (-2m_a + 3k_a z_a)\sin^2\frac{\theta}{2}}{\tilde{r}^2} + \dots,\tag{2.25}$$

$$g_{t\phi_2} = \frac{4i\sum_a (2m_a + 3k_a z_a)\cos^2\frac{\theta}{2}}{\tilde{r}^2} + \dots,$$

from which we extract that the mass $E$ and the angular momenta $J_\pm = \frac{J_1 \pm J_2}{2}$:

$$E = 3\pi\sum_a \frac{k_a^2}{h_a}, \qquad J_+ = -3\pi i\sum_a k_a z_a, \qquad J_- = -\pi i\sum_a \frac{k_a^3}{h_a^2}.\tag{2.26}$$

Since the solutions are supersymmetric, according to (1.4) the electric charge $Q$ must be related to the mass by

$$E = \sqrt{3}\,Q\,. \tag{2.27}$$

We can verify this by considering the asymptotic expansion of the field strength,

$$F_{t\tilde{r}} = \sqrt{3}\,\partial_{\tilde{r}}f = \frac{8\sqrt{3}}{\tilde{r}^3}\sum_a \frac{\mathsf{k}_a^2}{h_a} + \dots\,, \tag{2.28}$$

from which it follows that the electric charge is

$$Q = \sqrt{3}\,\pi \sum_a \frac{\mathsf{k}_a^2}{h_a}\,. \tag{2.29}$$

## 2.1 Global identifications and asymptotic boundary conditions

Using the coordinates introduced above, the asymptotic metric and gauge field read

$$
\begin{aligned}
\mathrm{d}s^2 &\longrightarrow \mathrm{d}\tau^2 + \mathrm{d}\tilde{r}^2 + \tilde{r}^2 \mathrm{d}s_{S^3}^2\,,\\
A &\longrightarrow \sqrt{3}\,\mathrm{i}(\mathrm{d}\tau - \mathrm{d}\alpha_\infty)\,,
\end{aligned}
\tag{2.30}
$$

where again $\tilde{r}^2 = 4r$, while $\alpha_\infty$ is defined as the asymptotic value of $\alpha$, and

$$\mathrm{d}s_{S^3}^2 = \frac{1}{4}\left[\mathrm{d}\theta^2 + \sin^2\theta\,\mathrm{d}\phi^2 + (\mathrm{d}\psi + \cos\theta\,\mathrm{d}\phi)^2\right]. \tag{2.31}$$

For this to describe the unit metric on a smooth $S^3$ parameterized by $(\theta, \phi, \psi)$, we must take $\theta \in [0, \pi]$ and demand regularity at the poles in $\theta = 0$ and $\theta = \pi$, which leads to the identifications

$$(\phi, \psi) \sim (\phi + 2\pi, \psi - 2\pi) \sim (\phi, \psi + 4\pi)\,. \tag{2.32}$$

Crucially, since we are compactifying the Euclidean time we must also specify how the angular coordinates are identified when going around the thermal $S^1$. This can be expressed as

$$(\tau, \phi, \psi) \sim (\tau + \beta, \phi - \mathrm{i}\omega_+, \psi + \mathrm{i}\omega_-)\,, \tag{2.33}$$

where $\beta$ is interpreted as inverse temperature, while $\omega_\pm$ play the role of angular velocities (at least when these are real quantities). Overall, the angular coordinates satisfy the identifications,[5]

$$(\tau, \phi, \psi) \sim (\tau + \beta, \phi - \mathrm{i}\omega_+, \psi + \mathrm{i}\omega_-) \sim (\tau, \phi + 2\pi, \psi - 2\pi) \sim (\tau, \phi, \psi + 4\pi)\,. \tag{2.34}$$

---

[5]Making the transformation $\phi = \phi_1 + \phi_2$, $\psi = \phi_2 - \phi_1$, we obtain standard $2\pi$-periodic coordinates $\phi_1, \phi_2$ satisfying the more symmetric identifications

$$(\tau, \phi_1, \phi_2) \sim (\tau + \beta, \phi_1 - \mathrm{i}\omega_1, \phi_2 - \mathrm{i}\omega_2) \sim (\tau, \phi_1 + 2\pi, \phi_2) \sim (\tau, \phi_1, \phi_2 + 2\pi)\,,$$

while the metric (2.31) becomes

$$\mathrm{d}s_{S^3}^2 = \mathrm{d}\vartheta^2 + \sin^2\vartheta\,\mathrm{d}\phi_1^2 + \cos^2\vartheta\,\mathrm{d}\phi_2^2\,, \qquad \text{with} \quad \vartheta = \theta/2\,.$$

To the vectors $\partial_{\phi_1} = \partial_\phi - \partial_\psi$ and $\partial_{\phi_2} = \partial_\phi + \partial_\psi$ we associate the angular momenta $J_1, J_2$ and the angular velocities $\omega_1, \omega_2$, which are those appearing in the introduction. These are related to the quantities appearing above as

$$\omega_\pm = \omega_1 \pm \omega_2\,, \qquad J_\pm = \frac{1}{2}(J_1 \pm J_2)\,,$$

which explains the origin of the $\pm$ labels used in the main text. Note that in our conventions, $J_+$ advances $\phi$, while $J_-$ advances $-\psi$.

A basis of independent Killing vectors having closed $2\pi$-periodic orbits under the identifications above is given by

$$\frac{1}{2\pi}\left(\beta\partial_\tau - \mathrm{i}\omega_+\partial_\phi + \mathrm{i}\omega_-\partial_\psi\right), \qquad \partial_\phi - \partial_\psi, \qquad \partial_\phi + \partial_\psi. \tag{2.35}$$

These generate the $U(1)^3$ isometry which will play a central role in the following. We will denote the first as the *thermal isometry*, and the other two as axial isometries. The orbits of $\partial_\phi \pm \partial_\psi$ are contractible, since they collapse to zero size at either one of the poles of the asymptotic $S^3$. Then fermion fields must be antiperiodic when going around such orbits. Recalling (2.34), we infer that shifts of the form $\omega_\pm \to \omega_\pm + 4\pi\mathrm{i}n_\pm$, with $n_\pm \in \mathbb{Z}$, are an invariance of the boundary conditions for both bosonic and fermionic fields. As far as the boundary conditions are concerned, we can thus take

$$\operatorname{Im}\omega_\pm \in [0, 4\pi), \tag{2.36}$$

with no loss of generality. Regarding the thermal isometry, we assume that the theory admits a supersymmetric non-extremal black hole solution having precisely inverse temperature $\beta$ and angular velocities $\omega_\pm$ (this will be discussed in section 4). Then the first vector in (2.35) is proportional to the generator of the horizon, and its orbits are contractible in the bulk of this solution. Well-definiteness of the spinorial supersymmetry parameter requires it to be antiperiodic when completing one revolution around the contractible orbits, which imposes

$$\omega_+ = 2\pi\mathrm{i}. \tag{2.37}$$

In order to see this, note that the supersymmetry parameter is neutral under $\partial_\tau$, while (in our conventions) it has charge $+1/2$ under both $\partial_\phi \pm \partial_\psi$, namely it has charge $1/2$ under $\partial_\phi$ and charge $0$ under $\partial_\psi$; hence the choice (2.37) is needed for the spinor to acquire the correct phase $\mathrm{e}^{2\pi\mathrm{i}\cdot\frac{1}{2}} = -1$. This argument, first noted for gauged supergravity in [12] and used for the ungauged case in [14], is very general and applies to many different situations, see [6] for a discussion.

The boundary conditions are completed by specifying the holonomy of the supergravity gauge field $A$ around the asymptotic $S^1$, which measures an electrostatic potential. We define it as

$$\varphi = -\mathrm{i}\int_{S^1_\infty}\left(A - \sqrt{3}\,\mathrm{i}\,\mathrm{d}\tau\right) = -\sqrt{3}\int_{S^1_\infty}\mathrm{d}\alpha_\infty, \tag{2.38}$$

where in the second equality we have used the asymptotic form of the gauge field given in (2.30).[6]

Note that the quantities $\omega_\pm$ and $\varphi$ are related to the ordinary angular velocities $\Omega_\pm$ and electrostatic potential $\Phi$ appearing in black hole thermodynamics as $\omega_\pm = \beta\Omega_\pm$ and $\varphi = \beta(\Phi - \sqrt{3})$. The shift in the electrostatic potential is chosen so that $\varphi$ corresponds to the potential appearing in the supersymmetric index, as discussed in the introduction.

Regularity in the bulk provides a map between $\beta, \omega_\pm, \varphi$ and the parameters of the solution introduced above. This depends on which combinations of the $U(1)$ isometries degenerate, which leads us to discuss the rod structure of the solution in the next section.

---

[6]An alternative way of introducing the same electrostatic potential would be to prescribe that the field identifications around the thermal circle involve a gauge transformation with gauge function $\alpha = -\mathrm{i}\Phi\tau$.

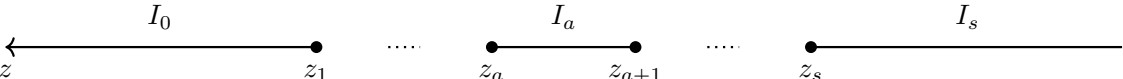

Figure 1: Generic representation of the rod structure. The compact rod $I_a$ joins the Gibbons-Hawking centers placed at $z_a$ and $z_{a+1}$, while the dots represents any possible arrangement of compact rods. The first and last ones are semi-infinite segments originating from $z_1$ and $z_s$.

## 3 Rod structure and topology of fixed loci

### 3.1 The rod structure

The global properties of the solutions are characterized by a rod structure. This is closely related to the one introduced for Lorentzian solutions [16, 31, 32], however we will include the Euclidean time circle in the description. In order to illustrate the rod structure in our setup, recall that the solutions we consider have a $U(1)^3$ isometry generated by linear combinations of the supersymmetric Killing vector $\partial_\tau$, the vector $\partial_\psi$ pointing along the Gibbons-Hawking fibre and the vector $\partial_\phi$ generating rotations around the $z$-axis in the $\mathbb{R}^3$ base of the Gibbons-Hawking space, as seen in the previous section. Denoting by $\mathcal{M}$ the five-dimensional space-time manifold, we can thus introduce the orbit space $\mathcal{M}/U(1)^3$. Parameterizing the $\mathbb{R}^3$ base of the Gibbons-Hawking space with cylindrical coordinates $(\rho \geq 0, \phi, z)$, the orbit space is the half-plane spanned by $(\rho, z)$, with boundary the infinite $z$-axis. The action of the isometries generates a three-torus at each interior point of the orbit space, while it is degenerate at the boundary. The boundary line is divided into segments – the rods – characterized by the specific $U(1)$ isometry which degenerates there. At the intersection of two adjacent rods, a $U(1) \times U(1)$ isometry degenerates, hence intersection points are corners of the orbit space. In our supersymmetric setup, these points coincide with the centers $z_a$ of the harmonic functions introduced in section 2. We call *rod vectors* the Killing vectors whose closed orbits collapse at a rod. The fixed locus of a rod vector is the three-dimensional space made by the rod and the two-torus foliated over it. At the intersection of two rods, a combination of the adjacent rod vectors degenerates; the corresponding fixed locus is a one-dimensional circle. Borrowing the terminology of [53], we will call *bolts* and *nuts* these co-dimension two and co-dimension four loci, respectively.

We label by $I_a = [z_a, z_{a+1}]$ the rod joining the centers at $z_a$ and $z_{a+1}$, with $a = 0, 1, \ldots, s$, as illustrated in figure 1. Our convention is that $z_0 = +\infty$ and $z_{s+1} = -\infty$; hence $I_0$ and $I_s$ denote the semi-infinite rods $(+\infty, z_1]$ and $[z_s, -\infty)$, respectively. After fixing an arbitrary overall constant, the Killing vector degenerating at the rod $I_a$ is

$$\xi_{I_a} = \partial_\phi - \mathrm{i}\breve{\omega}_{I_a} \partial_\tau - \chi_{I_a} \partial_\psi, \tag{3.1}$$

where $\breve{\omega}_{I_a}$ and $\chi_{I_a}$ denote the $\phi$-component of the one-forms $\breve{\omega}$ and $\chi$ evaluated at the rod $I_a$. These are constant and read

$$\breve{\omega}_{I_a} = \sum_{b>a} w_b - \sum_{b \leq a} w_b = -2 \sum_{b \leq a} w_b, \qquad \chi_{I_a} = \sum_{b>a} h_b - \sum_{b \leq a} h_b = 1 - 2 \sum_{b \leq a} h_b. \tag{3.2}$$

Notice that the rod vectors associated with the two semi-infinite rods $I_0$ and $I_s$ are always $\partial_\phi - \partial_\psi$ and $\partial_\phi + \partial_\psi$, respectively (indeed, these are precisely the vectors having fixed points at infinity).

The rod structure of a supersymmetric finite-temperature solution is fixed once certain integers are specified. Indeed, regularity requires that the orbits of $\xi_{I_a}$ close, hence (3.1) must

be a linear combination of the three U(1) generators (2.35) with integer coefficients, where we should recall that we have fixed $\omega_+ = 2\pi\mathrm{i}$. This combination can be parameterized as[7]

$$\xi_{I_a} = n_a\left(\frac{\beta}{2\pi}\partial_\tau + \partial_\phi + \frac{\mathrm{i}\omega_-}{2\pi}\partial_\psi\right) + (1-n_a)\partial_\phi + (2p_a+n_a-1)\partial_\psi\,, \qquad n_a, p_a \in \mathbb{Z}\,. \tag{3.3}$$

Comparing with (3.1) and using (3.2), we find that for each rod $I_a$ we have

$$\sum_{b\leq a}\mathrm{i}w_b = n_a\frac{\beta}{4\pi}\,, \qquad \sum_{b\leq a}h_b = p_a + \frac{n_a}{2}\left(1+\frac{\mathrm{i}\omega_-}{2\pi}\right)\,. \tag{3.4}$$

These relations provide a map between the solution parameters (on the left hand side) and a choice of integers $(n_a, p_a)$ for each rod, together with the boundary conditions $\beta, \omega_-$ which are fixed at infinity and are independent of the rod considered (on the right hand side).

In sections 3.2, 3.3 we perform a detailed regularity analysis and show how it constrains the rod structure data and determines the topology of the fixed loci associated with the rod vectors. Before diving into this, for the reader's convenience we provide a summary of the outcome of the analysis.

We will distinguish between *horizon rods* and *bubbling rods*, depending on the value of $n_a$. Let us discuss them in turn.

**Bubbling rods.** Bubbling rods have $n_a = 0$, hence the rod vector does not have a component along the thermal direction and reduces to

$$\xi_{I_a} = \partial_\phi + (2p_a-1)\partial_\psi\,. \tag{3.5}$$

From (3.4) it follows that

$$\sum_{b\leq a}w_b = 0\,, \qquad \sum_{b\leq a}h_b = p_a \in \mathbb{Z}\,. \tag{3.6}$$

Note that the definition includes the semi-infinite rods $I_0$ with $p_0 = 0$, and $I_s$ with $p_s = 1$. Moreover, if $I_{a-1}$ and $I_a$ are adjacent bubbling rods, the relations above imply

$$w_a = 0\,, \tag{3.7}$$

which is known as *bubble equation* [23], as well as

$$h_a = p_a - p_{a-1} \in \mathbb{Z}\,. \tag{3.8}$$

Studying the solution near the intersection point at $z = z_a$, one can see that there is a $\mathbb{C}^2/\mathbb{Z}_{|h_a|}$ orbifold singularity, unless one chooses $h_a = \pm 1$, in which case the metric is smooth around that point. We will allow for such orbifold singularities in our general discussion, since they may make sense when the solution is uplifted to string theory. The conditions for bubbling rods summarized here are equivalent to those known from previous analyses in Lorentzian signature, see e.g. [23, 32].

---

[7]This is obtained as follows. In terms of the basis vectors (2.35) generating the independent $2\pi$-periodic U(1)'s, each rod vector $\xi_I$ must be of the form

$$\xi_I = N_0\left(\frac{\beta}{2\pi}\partial_\tau + \partial_\phi + \frac{\mathrm{i}\omega_-}{2\pi}\partial_\psi\right) + N_1\left(\partial_\phi - \partial_\psi\right) + N_2\left(\partial_\phi + \partial_\psi\right),$$

with integer $N_0, N_1, N_2$ satisfying $N_0 + N_1 + N_2 = 1$, so that the coefficient of $\partial_\phi$ is 1. In (3.3), we have used $N_1 = 1 - N_0 - N_2$ and renamed $N_0 \equiv n, N_2 \equiv p$.

**Horizon rods.** Horizon rods have $n_a \neq 0$, hence their rod vector contains the generator of the thermal circle and the corresponding bolt is a Euclidean horizon. We find that if $I_a$ is a horizon rod, then the two adjacent rods $I_{a-1}$ and $I_{a+1}$ must be bubbling rods. From (3.4) it follows that the solution parameters associated to the horizon rod endpoints $z_a$ and $z_{a+1}$ satisfy

$$w_a = -w_{a+1} = n_a \frac{\beta}{4\pi i}, \tag{3.9}$$

$$h_a = p_a - p_{a-1} + \frac{n_a}{2}\left(1 + \frac{i\omega_-}{2\pi}\right), \qquad h_{a+1} = p_{a+1} - p_a - \frac{n_a}{2}\left(1 + \frac{i\omega_-}{2\pi}\right), \tag{3.10}$$

implying

$$h_a + h_{a+1} = p_{a+1} - p_{a-1} \in \mathbb{Z}. \tag{3.11}$$

We emphasize that these relations must be satisfied by each horizon rod $I_a$ with the same $\beta$, $\omega_-$, since these are boundary conditions fixed at infinity.

We define the angular velocities of the horizon associated with $I_a$ as the constant coefficients $\omega_\pm^{I_a}$ that can be read from the rod vector expressed as[8]

$$\xi_{I_a} = \frac{1}{2\pi}\left(n_a\beta\,\partial_\tau - i\omega_+^{I_a}\partial_\phi + i\omega_-^{I_a}\partial_\psi\right). \tag{3.12}$$

These are given by

$$\omega_+^{I_a} = 2\pi i, \qquad \omega_-^{I_a} = n_a\omega_- + 2\pi i(1 - n_a - 2p_a). \tag{3.13}$$

Hence the horizon angular velocity $\omega_+^{I_a}$ is the same as the chemical potential $\omega_+ = 2\pi i$, while $\omega_-^{I_a}$ is related to the chemical potential $\omega_-$ by a linear combination controlled by the integers $n_a, p_a$.

As we will illustrate in the examples, the information above gives a systematic way to construct solutions by introducing one rod after the other, starting from the semi-infinite rod $I_0$, assigning integers $n_a$ and $p_a$ to each new rod $I_a$, and ending with the other semi-infinite rod, $I_s$. We emphasize that in order to conclude that a solution with a given rod structure actually exists, one has to solve the algebraic equations (3.7), (3.9), where the expression for $w_a$ was given in (2.22). These equations fix the distances between the centers in terms of the other parameters and, for horizon rods, in terms of the chosen $\beta$. When there are many centers, the equations become hard to solve (a method has been proposed in [54]). A complete analysis of these equations in the complexified setup goes beyond the scope of the present work; in particular, one should check if they are solved by real or complex values of $\delta_{ab}$. We will comment more on this issue while discussing some explicit examples in the next sections.

---

[8] These definitions assume that $n_a$ is positive. In general, one should define the horizon angular velocities as $\xi_{I_a} = \frac{\text{sign}(n_a)}{2\pi}\left(|n_a|\beta\,\partial_\tau - i\omega_+^{I_a}\partial_\phi + i\omega_-^{I_a}\partial_\psi\right)$, since the vector in parenthesis now generates evolution along the thermal circle in the same direction as the first of the basis vectors in (2.35). Accordingly, the expressions for the angular velocities become

$$\omega_+^{I_a} = 2\pi i\,\text{sign}(n_a), \qquad \omega_-^{I_a} = |n_a|\omega_- + 2\pi i[\,\text{sign}(n_a)(1 - 2p_a) - |n_a|\,].$$

Therefore, by choosing the sign of $n_a$, we can engineer different horizons in a multi-horizon solution to realize both allowed possibilities for the angular velocity: $\omega_+^{I_a} = \pm 2\pi i$. To avoid cluttering the notation, we focus in the main text on the case $n_a > 0$, although all of the expressions we derive can be readily extended to the general case. Note that configurations such that $\omega_+^{I_a}$ changes sign while $\omega_-^{I_a}$ remains invariant are obtained by sending $n_a \to -n_a$ and $p_a \to 1 - p_a$. Making this transformation for all rods $I_a$, together with $z_a \to -z_a$, sends the solution into itself, since it is equivalent to performing the $\pi$-rotation in $\mathbb{R}^3$ which sends $z \to -z$ and $\phi \to -\phi$.

**Topology of bolts.** For each rod vector, we can discuss the topology of the associated fixed locus in the five-dimensional solution. This is a three-dimensional bolt, made by a foliation along the rod of the two circles that are non-degenerate in the interior of that rod. We will distinguish between *bubbling bolts* and *horizon bolts*, associated with bubbling rods and horizon rods, respectively.

We have mentioned that horizons can only sit between two bubbling bolts. On the other hand, next to a bubbling rod $I_a$ one can have either a bubbling rod or a horizon rod, and the topology of the bolt associated with $I_a$ is sensitive to the neighbours. Indeed, the torus foliation along $I_a$ making the bubbling bolt involves the closed orbits of a Killing vector involving $\partial_\tau$; these collapse at one of the two rod endpoints if the adjacent rod is a horizon rod, while they remain of finite size if the adjacent rod is a bubbling one. We should therefore distinguish between the following cases, depending on the nature of the adjacent bolts.

- **bubble$_{a-1}$–HORIZON$_a$–bubble$_{a+1}$.** We find that the bolt associated with a horizon rod $I_a$ has lens space topology

$$L\left(|n_a(p_{a+1}-p_{a-1})|,\, 1+\mathtt{a}\,(p_{a+1}-p_{a-1})\right) \simeq S^3/\mathbb{Z}_{|n_a(p_{a+1}-p_{a-1})|}, \tag{3.14}$$

where $\mathtt{a}$ is an integer defined via the equation

$$\mathtt{a}\,(p_a-p_{a+1})+\mathtt{b}\,n_a\,(p_{a-1}-p_{a+1})=1\,, \qquad \mathtt{a},\mathtt{b}\in\mathbb{Z}\,. \tag{3.15}$$

In the special case $|n_a(p_{a-1}-p_{a+1})|=1$, the horizon has $S^3$ topology, while for $p_{a-1}=p_{a+1}$ the topology is that of $S^1\times S^2$. For $|n_a|=1$, this reduces to the classification of (non-extremal) horizon topologies appearing in the Lorentzian analysis of [32], as depicted in figure 2.

- **horizon$_{a-1}$–BUBBLE$_a$–horizon$_{a+1}$.** The bolt associated to the bubbling rod $I_a$ sitting between two horizons also has lens space topology

$$L\left(|n_{a-1}(p_a-p_{a+1})-n_{a+1}(p_a-p_{a-1})|,\, 1+\mathtt{a}\,(p_{a+1}-p_{a-1})\right), \tag{3.16}$$

with

$$\mathtt{a}\,(p_a-p_{a+1})+\mathtt{b}\,[n_{a-1}(p_a-p_{a+1})-n_{a+1}(p_a-p_{a-1})]=1\,, \qquad \mathtt{a},\mathtt{b}\in\mathbb{Z}\,. \tag{3.17}$$

The circles that collapse smoothly at the rod endpoints intersecting the two neighbouring horizons are both generated by vectors involving $\partial_\tau$. This tells us that if we consider the hypersurface at constant $\tau$ then we find a 2-tube [16].

- **horizon$_{a-1}$–BUBBLE$_a$–bubble$_{a+1}$,** or **bubble$_{a-1}$–BUBBLE$_a$–horizon$_{a+1}$.** The bolts have topology

$$L\left(|n_{a-1}|,\, p_{a-1}-p_a\right), \qquad \text{or} \qquad L\left(|n_{a+1}|,\, p_{a+1}-p_a\right), \tag{3.18}$$

respectively. This assumes $|h_{a+1}|\equiv|p_{a+1}-p_a|=1$ (in the first case) or $|h_a|\equiv|p_a-p_{a-1}|=1$ (in the second case), so that we have a smooth topology. If these conditions are not met, then there is a conical singularity $\mathbb{C}/\mathbb{Z}_{|h_{a+1}|}$ (in the first case), or $\mathbb{C}/\mathbb{Z}_{|h_a|}$ (in the second case) at the bubbling centers, and the lens space is branched. For $|n_{a\mp1}|=1$ this reduces to a branched $S^3$. The circle that smoothly collapses at the rod endpoint touching the horizon is generated by a vector involving $\partial_\tau$, while the one at the other endpoint does not. This implies that the hypersurface at constant $\tau$ has disc topology, possibly with a conical singularity at the tip.

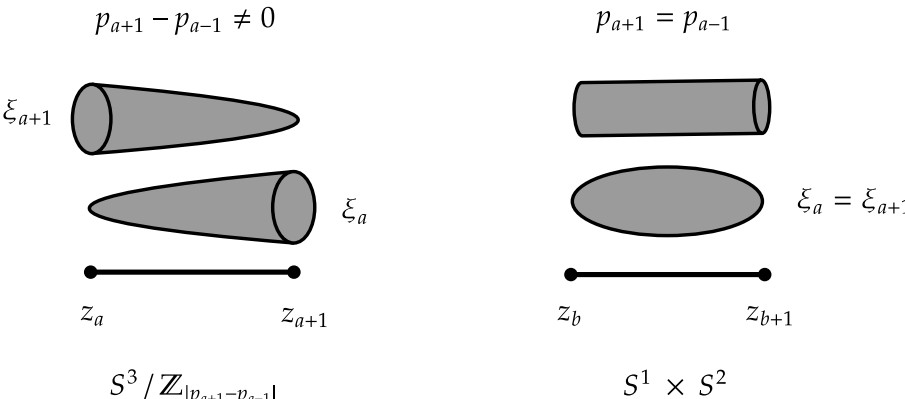

Figure 2: For $|n_a| = 1$, bolts associated to horizon rods have $L(|p_{a+1}-p_{a-1}|, 1)$ topology. In the case $p_{a+1}-p_{a-1} \neq 0$, each of the two axial U(1) isometries shrinks at one rod endpoint, realizing the lens space topology $S^3/\mathbb{Z}_{|p_{a+1}-p_{a-1}|}$. When $p_{a+1} = p_{a-1}$, instead, the same U(1) shrinks at both endpoints, while the other never does; then, the horizon has $S^1 \times S^2$ topology.

- **bubble$_{a-1}$–BUBBLE$_a$–bubble$_{a+1}$.** The bolt associated with the rod $I_a$ generically has topology

$$S^1 \times \Sigma_{[h_a, h_{a+1}]}, \tag{3.19}$$

where $\Sigma_{[h_a, h_{a+1}]} = \mathbb{WCP}^1_{[h_a, h_{a+1}]}$ denotes a complex weighted projective space, also known as *spindle*. This is topologically a sphere with axial symmetry and orbifold singularities at the two poles, of the type $\mathbb{C}/\mathbb{Z}_{|h_a|}$ and $\mathbb{C}/\mathbb{Z}_{|h_{a+1}|}$, leading to conical deficit angles $2\pi(1-1/|h_a|)$ and $2\pi(1-1/|h_{a+1}|)$, respectively. If $|h_a| = 1$ and $|h_{a+1}| \neq 1$ (or vice-versa) then we have a teardrop. If $|h_a| = |h_{a+1}| = 1$ then we have a smooth $S^2$.

## 3.2 Regularity of the metric

We next come to our detailed regularity analysis, proving the claims summarized above. This consists of checking absence of Dirac-Misner strings as well as smoothness of the geometry at the rod endpoints – up to orbifold singularities.[9] We stress that while these regularity conditions are standard when the metric and the gauge field are real, they become much less obvious for a complexified solution. However, we explicitly show how implementing suitable *complex* changes of coordinates one can reduce to the usual analysis in a real space near to the potentially singular points. We will comment further on the need to consider complex solutions in section 4.

In order to study regularity of the geometry, it is convenient to choose coordinates adapted to the rod vectors $\xi_{I_a}$ by making the transformation

$$\tau_a = \frac{2\pi}{\beta}\left(\tau + i\breve{\omega}_{I_a}\phi\right), \qquad \psi_a = \psi + \chi_{I_a}\phi + c_a\left(\tau + i\breve{\omega}_{I_a}\phi\right), \qquad \phi_a = \phi, \tag{3.20}$$

where $c_a$ is a constant that will be fixed momentarily. Note that this is a linear transformation with a priori complex coefficients. This change of coordinates leads to $\xi_{I_a} = \partial_{\phi_a}$, and is related

---

[9]In principle, one should also check whether the solution admits critical surfaces where $H$ and $f$ vanish, and if so analyze the regularity of the metric and gauge field there, as done in [16] in the Lorentzian case.

to cancellation of Dirac-Misner string singularities. The latter arise when the connection one-forms $\chi$ and $\breve{\omega}$ have a non-vanishing $\phi$-component on the $z$-axis, where $\mathrm{d}\phi$ is not well-defined. A string singularity on the $z$-axis is just a coordinate singularity if there exists a coordinate transformation that removes it, and the transformation (3.20) indeed does the job. However, we still have to show compatibility of the new coordinates in the overlaps between the different regions containing the different rods $I_a$. Let us prove this. We start by specifying the condition for the new coordinates to obey untwisted periodic identifications. Recalling (2.34), we deduce the identifications

$$(\tau_a, \psi_a, \phi_a) \sim (\tau_a + 2\pi, \psi_a + \mathrm{i}\omega_- + 2\pi + \beta c_a, \phi_a) \sim (\tau_a, \psi_a + 4\pi, \phi_a)$$
$$\sim \left( \tau_a + \frac{4\pi^2 \mathrm{i}\, \breve{\omega}_{I_a}}{\beta}, \psi_a + 2\pi \left( \chi_{I_a} - 1 + \mathrm{i}\breve{\omega}_{I_a} c_a \right), \phi_a + 2\pi \right). \tag{3.21}$$

These are untwisted provided

$$\mathrm{i}\omega_- + 2\pi + \beta c_a = 4\pi \tilde{p}_a, \qquad \frac{2\pi \mathrm{i}\breve{\omega}_{I_a}}{\beta} = -n_a, \qquad \chi_{I_a} - 1 + \mathrm{i}\breve{\omega}_{I_a} c_a = -2p_a, \tag{3.22}$$

where $n_a, p_a, \tilde{p}_a$ are a set of convenient integers. Indeed in this way we have

$$(\tau_a, \psi_a, \phi_a) \sim (\tau_a + 2\pi, \psi_a, \phi_a) \sim (\tau_a, \psi_a + 4\pi, \phi_a) \sim (\tau_a, \psi_a, \phi_a + 2\pi). \tag{3.23}$$

With the conditions (3.22) imposed, the Killing vector $\xi_{I_a}$ contracts smoothly on the rod $I_a$, so that the normal space close to the bolt forms a smooth $\mathbb{R}^2$:

$$\mathrm{d}s^2 = \ldots + f^{-1} H \left( \mathrm{d}\rho^2 + \rho^2 \mathrm{d}\phi_a^2 \right). \tag{3.24}$$

Solving the first in (3.22) for $c_a$,

$$\beta c_a = 4\pi \tilde{p}_a - 2\pi - \mathrm{i}\omega_-, \tag{3.25}$$

we end up with

$$\frac{2\pi \mathrm{i}\breve{\omega}_{I_a}}{\beta} = -n_a, \qquad \chi_{I_a} = 1 - 2p_a - n_a \left( 1 + \frac{\mathrm{i}\omega_-}{2\pi} - 2\tilde{p}_a \right). \tag{3.26}$$

We note that we can set $\tilde{p}_a = 0$ without loss of generality, since it can always be reabsorbed in a redefinition of $p_a$. We will do so in the following. Eqs. (3.26) imply that on the overlap between two charts containing the rod $I_a$ and the rod $I_b$, respectively, one has

$$\tau_b = \tau_a - (n_b - n_a)\phi_a, \qquad \phi_b = \phi_a, \qquad \psi_b = \psi_a - 2(p_b - p_a)\phi_a, \tag{3.27}$$

which ensures compatibility of these coordinate patches, as we wanted to show.

Let us comment on the meaning of discussing regularity conditions for a complexified metric. We note that the original coordinates $(\tau, \phi, \psi)$ should take complex values, since they satisfy the generically complex identifications (2.34). However, we can assume that the new coordinates $(\tau_a, \phi_a, \psi_a)$ introduced by the complex transformation (3.20) are real, since they have real period; this can be done in every patch, since the map (3.27) between $(\tau_b, \phi_b, \psi_b)$ and $(\tau_a, \phi_a, \psi_a)$ is real. We conclude that if we describe the solution using the real coordinates $(\tau_a, \phi_a, \psi_a)$ in every patch, then the complexification is pushed into the metric (and gauge field) components, and a regularity statement such as the one associated with (3.24), where we check that the orbits of the rod vector $\xi_{I_a} = \partial_{\phi_a}$ smoothly shrink to zero size, appears to make sense.

Using (3.2) and setting $\tilde{p}_a = 0$, eqs. (3.26) reduce to (3.4), showing that the condition for absence of Dirac-Misner string singularities leads us to introduce a pair of independent integers $(n_a, p_a)$ for each rod $I_a$, which determine the rod vectors as in (3.3). In general, eqs. (3.26) form a set of $2(s-1)$ non-trivial equations, the ones associated to the first $(I_0)$ and last $(I_s)$ rods being trivially solved as they do not involve the parameters of the solution.[10] The remaining $2(s-1)$ equations fall in two categories. The first set in (3.26) can be solved for the $h_a$, providing an expression of the latter in terms of $\omega_-$ and the integers $n_a, p_a$ (or viceversa). The second set of equations corresponds to a finite-$\beta$ version of the *bubble equations* arising in the context of horizonless topological solitons [21, 22, 24].

We now study the metric near the Gibbons-Hawking centers. At this stage, it is useful to distinguish two types of centers, *horizon poles* if $w_a \neq 0$, and *bubbling centers* if $w_a = 0$.

**Smoothness at bubbling centers ($w_a = 0$).** We start by analyzing the geometry near a bubbling center. To this aim, we introduce the new radial coordinate $\tilde{r}_a = 2\sqrt{r_a}$. Keeping only the relevant terms in the $\tilde{r}_a \to 0$ limit, one finds that

$$
ds^2 = f_a^2 \, d\tau_a'^2 + \frac{h_a}{f_a} \left\{ d\tilde{r}_a^2 + \frac{\tilde{r}_a^2}{4} \left[ \left( d\psi_a' + \cos\theta_a \, d\phi_a' \right)^2 + d\theta_a^2 + \sin^2\theta_a \, d\phi_a'^2 \right] \right\} + \dots, \quad (3.28)
$$

where, using (3.4), we have introduced the coordinates

$$
\tau_a' = \tau - \frac{\beta n_{a-1}}{2\pi} \phi, \qquad \psi_a' = \frac{\psi + (1 - 2Y_a - h_a)\phi - \frac{i\omega_- + 2\pi}{\beta}\tau}{h_a}, \qquad \phi_a' = \phi, \quad (3.29)
$$

and

$$
Y_a = \sum_{b<a} h_b. \quad (3.30)
$$

The periodic identifications of the new coordinates are inherited from (2.32) and (2.33), namely:

$$
\left( \tau_a', \psi_a', \phi_a' \right) \sim \left( \tau_a' + \beta, \psi_a', \phi_a' \right) \sim \left( \tau_a', \psi_a' + \frac{4\pi}{h_a}, \phi_a' \right)
$$
$$
\sim \left( \tau_a' - \beta n_{a-1}, \psi_a' - \frac{2\pi(2Y_a + h_a)}{h_a}, \phi_a' + 2\pi \right). \quad (3.31)
$$

Demanding regularity of the metric at $\theta_a = 0, \pi$ yields the following conditions

$$
n_{a-1} = \frac{4\pi}{\beta} \sum_{b<a} i w_b = 0, \qquad Y_a = p_{a-1} \in \mathbb{Z}, \qquad h_a = \pm 1, \quad (3.32)
$$

which imply that the metric near a bubbling center is that of $S^1 \times \mathbb{R}^4$. The last condition in (3.32) can be relaxed to allow for integer values of $h_a$, if one admits discrete orbifold singularities. In such case, the periodic identifications (3.31) become

$$
\left( \tau_a', \psi_a', \phi_a' \right) \sim \left( \tau_a' + \beta, \psi_a', \phi_a' \right) \sim \left( \tau_a', \psi_a' + \frac{4\pi}{h_a}, \phi_a' \right) \sim \left( \tau_a', \psi_a' + 2\pi, \phi_a' + 2\pi \right), \quad (3.33)
$$

and the geometry near the center is that of $S^1 \times \mathbb{R}^4/\mathbb{Z}_{|h_a|}$, where $h_a \in \mathbb{Z}$ and the $\mathbb{Z}_{|h_a|}$ quotient acts in the same way on the polar angles of the two orthogonal $\mathbb{R}^2$ planes in $\mathbb{R}^4$.

The first condition in (3.32), when combined with (3.26) and with the requirement $w_a = 0$ characterizing bubbling centers, imposes that both the integers $n_a$ and $n_{a-1}$, associated to the two rods adjacent to the center, must vanish:

$$
n_{a-1} = n_a = 0. \quad (3.34)
$$

---

[10]It is sufficient to take $p_0 = n_0 = n_s = 0$ and $p_s = 1$, since $\breve{\omega}_{I_0} = \breve{\omega}_{I_s} = 0$ and $\chi_{I_0} = -\chi_{I_s} = 1$.

This implies that no bubbling center can be placed inside a horizon. Finally, let us further note that the integers $p_a$ can be expressed in terms of $h_a$ and $p_{a-1}$, by the relation

$$p_a = h_a + p_{a-1} = \sum_{b \leq a} h_b \in \mathbb{Z}, \tag{3.35}$$

as it follows from (3.32). Therefore, the local analysis near a bubbling center gives the conditions listed around (3.6).

**Smoothness at horizon poles ($w_a \neq 0$).**  The metric near a horizon pole is given by

$$ds^2 = \left(\frac{i w_a f_a}{h_a}\right)^2 d\psi_a'^2 + \frac{h_a}{f_a}\left[d\tilde{r}_a^2 + \tilde{r}_a^2\left(\frac{d\theta_a^2}{4} + \sin^2\frac{\theta_a}{2}\,d\phi_a'^2 + \cos^2\frac{\theta_a}{2}\,d\tau_a'^2\right)\right] + \dots, \tag{3.36}$$

where we have introduced coordinates such that

$$\begin{aligned}
\psi_a' &= \psi + (1 - 2Y_a - 2h_a X_a)\,\phi - \frac{h_a \tau}{i w_a}, \\
\tau_a' &= \frac{\tau}{2 i w_a} + X_a \phi, \qquad \phi_a' = (1 - X_a)\,\phi - \frac{\tau}{2 i w_a},
\end{aligned} \tag{3.37}$$

with

$$X_a = -\sum_{b<a} \frac{w_b}{w_a}, \qquad Y_a = \sum_{b<a} h_b. \tag{3.38}$$

The periodic identifications of the new coordinates, inherited from (2.32), (2.33), are given by

$$\begin{aligned}
\left(\psi_a', \tau_a', \phi_a'\right) &\sim \left(\psi_a' + 4\pi, \tau_a', \phi_a'\right) \\
&\sim \left(\psi_a' - 4\pi\,(Y_a + h_a X_a),\, \tau_a' + 2\pi X_a,\, \phi_a' + 2\pi\,(1 - X_a)\right) \\
&\sim \left(\psi_a' + i\omega_- + 2\pi - \frac{\beta h_a}{i w_a},\, \tau_a' + \frac{\beta}{2 i w_a},\, \phi_a' - \frac{\beta}{2 i w_a}\right).
\end{aligned} \tag{3.39}$$

Demanding regularity of the metric at $\theta_a = 0, \pi$ boils down to the condition

$$X_a \in \{0, 1\}. \tag{3.40}$$

The choice determines whether the horizon pole is a north or a south pole.

- **Horizon north pole.** A horizon north pole is defined by the condition $X_a = 0$, which implies

$$n_{a-1} = 0, \qquad w_a = n_a \frac{\beta}{4\pi i}. \tag{3.41}$$

These follow after using the definition of $X_a$ in (3.38), together with (3.26) and (3.2), and simply tell us that $I_{a-1}$ must be a bubbling rod, while $I_a$ is a horizon rod. Moreover, from the second of (3.26), one deduces

$$Y_a = \sum_{b \leq a} h_a = p_{a-1}, \qquad h_a = p_a - p_{a-1} + \frac{n_a}{2}\left(1 + \frac{i\omega_-}{2\pi}\right). \tag{3.42}$$

Putting this information together, the above identifications (3.39) reduce to

$$\begin{aligned}
\left(\psi_a', \tau_a', \phi_a'\right) &\sim \left(\psi_a' + 4\pi, \tau_a', \phi_a'\right) \sim \left(\psi_a', \tau_a', \phi_a' + 2\pi\right) \\
&\sim \left(\psi_a' + \frac{4\pi\,(p_{a-1} - p_a)}{n_a},\, \tau_a' + \frac{2\pi}{n_a},\, \phi_a' - \frac{2\pi}{n_a}\right),
\end{aligned} \tag{3.43}$$

which inform us that the space near a horizon north pole is $\left(S^1 \times \mathbb{R}^4\right)/\mathbb{Z}_{|n_a|}$. The orbifold action is free as long as $p_{a-1} \neq p_a$.

Table 1: Classification of centers and their adjacent rods. Centers are distinguished by which rods meet there. This is reflected in the allowed values for the coefficients $w_a$ of the one-form $\breve{\omega}$ and the integer $n_a$, as shown by the regularity analysis.

| $z_a$ | $w_a$ | $n_{a-1}$ | $I_{a-1}$ | $n_a$ | $I_a$ |
|---|---|---|---|---|---|
| Horizon north pole | $\neq 0$ | $= 0$ | bubble | $\neq 0$ | horizon |
| Horizon south pole | $\neq 0$ | $\neq 0$ | horizon | $= 0$ | bubble |
| Bubbling center | $= 0$ | $= 0$ | bubble | $= 0$ | bubble |

- **Horizon south pole.** A horizon south pole $z_a$ is in turn defined by the condition $X_a = 1$, which implies

$$w_a = -n_{a-1}\frac{\beta}{4\pi\mathrm{i}}\,, \qquad n_a = \sum_{b<a} w_b = 0\,. \tag{3.44}$$

Then, $I_{a-1}$ and $I_a$ must be horizon and bubbling rods respectively. This observation also implies that the center $a-1$ must necessarily be a horizon north pole. Namely, horizon poles are always paired. On top of these conditions, one further deduces from (3.26) that

$$Y_a + h_a X_a = \sum_{b\leq a} h_b = p_a\,, \qquad h_a = p_a - p_{a-1} - \frac{n_{a-1}}{2}\left(1 + \frac{\mathrm{i}\omega_-}{2\pi}\right)\,. \tag{3.45}$$

Thus, the periodic identifications of the coordinates read

$$\begin{aligned}\left(\psi_a', \tau_a', \phi_a'\right) &\sim \left(\psi_a' + 4\pi, \tau_a', \phi_a'\right) \sim \left(\psi_a', \tau_a', \phi_a' + 2\pi\right)\\ &\sim \left(\psi_a' + \frac{4\pi(p_a - p_{a-1})}{n_{a-1}}, \tau_a' - \frac{2\pi}{n_{a-1}}, \phi_a' + \frac{2\pi}{n_{a-1}}\right)\,.\end{aligned} \tag{3.46}$$

Then the space near the south pole is $(S^1 \times \mathbb{R}^4)/\mathbb{Z}_{|n_{a-1}|}$, acting freely provided $p_{a-1} \neq p_a$.

Thus, the summary of our findings (see also table 1) is that a regularity analysis does not allow horizon rods to be adjacent to each other: there must be bubbling rods right before and right after. Moreover, if $|n_a| \neq 1$, there will appear orbifold singularities at a pole of the horizon if either $p_{a+1}$ or $p_{a-1}$ are equal to $p_a$. If $|n_a| = 1$, there is no such regularity condition on the $p$'s of the adjacent rods.

It can readily be checked that the expressions found here directly imply eqs. (3.9), (3.10), (3.11). This shows the consistency of our previous analysis with the local study around the horizon poles.

## 3.3 Classification of bolt topologies

We now come to the study of the three-dimensional bolts associated with each rod vector $\xi_{I_a}$ given by (3.1), with the aim of classifying the possible topologies. It is convenient to write down the metric induced at the rod $I_a$ using the adapted coordinates introduced in (3.20):

$$\mathrm{d}s_{I_a}^2 = f^{-1}H^{-1}\left(\mathrm{d}\psi_a - \frac{\beta c_a}{2\pi}\mathrm{d}\tau_a\right)^2 + f^{-1}H\,\mathrm{d}z^2 + f^2\left[\frac{\beta}{2\pi}\mathrm{d}\tau_a + \mathrm{i}\omega_\psi\left(\mathrm{d}\psi_a - \frac{\beta c_a}{2\pi}\mathrm{d}\tau_a\right)\right]^2\,, \tag{3.47}$$

where $c_a$ is the constant given in (3.25), and all the functions are evaluated on the rod.

### 3.3.1 Local analysis at rod endpoints

In order classify the possible topologies for the fixed loci of the rod vectors $\xi_{I_a}$, we need to specify the local form of the metric near to the centers $z_a$ and $z_{a+1}$ where the bolt caps off, as well as the periodic identifications when going around the contracting circles. We first analyze the induced metric (3.47) near to the rod endpoints.

**Bubbling center.** Condition (2.20) guarantees that near the center the metric reads,

$$ds_a^2 = \frac{h_a}{f_a}\left[d\tilde{r}_a^2 + \tilde{r}_a^2\left(\frac{d\psi_a}{2h_a}\right)^2\right] + \left(\frac{\beta f_a}{2\pi}\right)^2 d\tau_a^2 + \dots, \tag{3.48}$$

where $\tilde{r}_a = 2\sqrt{r_a}$. This shows that the Killing vector with $2\pi$-periodic orbits contracting at a bubbling center is

$$\xi_a = 2\partial_{\psi_a} = 2\partial_\psi, \tag{3.49}$$

while the $2\pi$-periodic vector

$$\tilde{\lambda}_a = \partial_{\tau_a} = \frac{\beta}{2\pi}\partial_\tau + \left(1 + \frac{i\omega_-}{2\pi}\right)\partial_\psi, \tag{3.50}$$

generates a fixed $S^1$. The geometry near to the center is then that of $S^1 \times \mathbb{R}^2/\mathbb{Z}_{|h_a|}$. The orbifold action is singular (unless $h_a = \pm 1$), reflecting the presence of an $S^1 \times \mathbb{R}^2/\mathbb{Z}_{|h_a|} \times \mathbb{R}^2$ orbifold singularity in the full five-dimensional geometry [21].

**Horizon north pole.** For a horizon rod $I_a$, the following Killing vector contracts at the horizon north pole $z_a$:

$$\begin{aligned}
\xi_a = \xi_{I_a} - \xi_{I_{a-1}} &= \frac{\beta n_a}{2\pi}\partial_\tau + \left[2(p_a - p_{a-1}) + n_a\left(1 + \frac{i\omega_-}{2\pi}\right)\right]\partial_\psi \\
&= n_a\partial_{\tau_a} + 2(p_a - p_{a-1})\partial_{\psi_a}.
\end{aligned} \tag{3.51}$$

Here, $(\tau_a, \psi_a)$ are the adapted rod coordinates introduced in (3.20). In order to exhibit the local metric near the pole, we define the new angles

$$\tilde{\psi}_a = \psi_a - 2(p_a - p_{a-1})\frac{\tau_a}{n_a}, \qquad \tilde{\tau}_a = \frac{\tau_a}{n_a}, \tag{3.52}$$

and expand the metric for $\tilde{r}_a = 2\sqrt{r_a} \to 0$. This takes the form

$$ds_a^2 = \frac{h_a}{f_a}\left(d\tilde{r}_a^2 + \tilde{r}_a^2 d\tilde{\tau}_a^2\right) + \left(\frac{f_a i w_a}{h_a}\right)^2 d\tilde{\psi}_a^2 + \dots \tag{3.53}$$

It is easy to show that it corresponds to an orbifold $\left(S^1 \times \mathbb{R}^2\right)/\mathbb{Z}_{|n_a|}$, with the orbifold action specified by the identifications:

$$(\tilde{\tau}_a, \tilde{\psi}_a) \sim \left(\tilde{\tau}_a + \frac{2\pi}{n_a}, \tilde{\psi}_a - \frac{4\pi}{n_a}(p_a - p_{a-1})\right) \sim (\tilde{\tau}_a, \tilde{\psi}_a + 4\pi). \tag{3.54}$$

The orbifold is freely acting if $(p_a - p_{a-1})$ and $n_a$ are coprime. Eq. (3.54) identifies a $2\pi$-periodic Killing vector whose orbits describe the fixed $S^1$ at the endpoint:

$$\begin{aligned}
\tilde{\lambda}_a &= \frac{1}{n_a}\left(\partial_{\tilde{\tau}_a} - 2(p_a - p_{a-1})\partial_{\tilde{\psi}_a}\right) \\
&= \frac{\beta}{2\pi}\partial_\tau + \left(1 + \frac{i\omega_-}{2\pi}\right)\partial_\psi.
\end{aligned} \tag{3.55}$$

Note that this is the same vector as (3.50). Finally, the $2\pi$-periodic contracting vector (3.51) in these coordinates reads $\xi_a = \partial_{\tilde{\tau}_a}$.

**Horizon south pole.** The Killing vector contracting at a horizon south pole reads

$$\xi_a = \xi_{I_{a-1}} - \xi_{I_a} = \frac{\beta n_{a-1}}{2\pi}\partial_\tau + \left[2(p_{a-1}-p_a)+n_{a-1}\left(1+\frac{\mathrm{i}\omega_-}{2\pi}\right)\right]\partial_\psi$$
$$= n_{a-1}\,\partial_{\tau_a} + 2(p_{a-1}-p_a)\,\partial_{\psi_a}\,, \tag{3.56}$$

where $(\tau_a, \psi_a)$ are part of a system of adapted rod coordinates for the bubbling rod $I_a$, with $a$ being a south pole. The metric near the pole takes the same form as in (3.53), where $(\tilde{\tau}_a, \tilde{\psi}_a)$ are now given by

$$\tilde{\psi}_a = \psi_a - 2(p_{a-1}-p_a)\frac{\tau_a}{n_{a-1}}\,, \qquad \tilde{\tau}_a = \frac{\tau_a}{n_{a-1}}\,. \tag{3.57}$$

The Killing vector generating the fixed $S^1$ is the same as in the second line of (3.55).

### 3.3.2 Bolt topologies

We now turn to the analysis of global aspects. Aside from the two non-compact bolts associated with $I_0$ and $I_s$,[11] we consider the four types of compact bolts already introduced above, classified by the nature of the adjacent rods. For each bolt we may write relations between the associated Killing vectors $\xi_a$ and $\xi_{a+1}$ of the form:

$$\mathsf{q}_1\,\xi_a = \mathsf{q}_2\,\xi_{a+1} + \mathsf{p}\,\tilde{\lambda}_{a+1}\,, \qquad \mathsf{p},\,\mathsf{q}_{1,2}\in\mathbb{Z}\,. \tag{3.58}$$

We assume that $\mathsf{q}_1\,\mathsf{q}_2$ are coprime to $\mathsf{p}$. As reviewed in appendix B, when the regions bounding the neighborhoods about the centers $a$ and $a+1$ are glued together, the manifold obtained is topologically the lens space

$$L\left(\mathsf{p},\,\mathsf{a}\,\mathsf{q}_2\right)\,, \tag{3.59}$$

where

$$\mathsf{a}\,\mathsf{q}_1 + \mathsf{b}\,\mathsf{p} = 1\,, \qquad \mathsf{a},\,\mathsf{b}\in\mathbb{Z}\,. \tag{3.60}$$

If $|\mathsf{p}|=1$ and $\mathsf{a}\,\mathsf{q}_2\in\mathbb{Z}$, then the space has $S^3$ topology (up to possible conical singularities at the bubbling centers, as we specify below), while if $\mathsf{p}=0$ and $|\mathsf{a}\,\mathsf{q}_2|=1$, then the space topologically becomes $S^1\times S^2$ (again, up to possible conical singularities).

**bubble$_{a-1}$-HORIZON$_a$-bubble$_{a+1}$.** Let us begin by considering a horizon rod, connecting two horizon poles. The vectors contracting smoothly at the horizon north and south pole are given by (3.51) and (3.56), respectively. Then, we can express

$$(p_a-p_{a+1})\xi_a = (p_a-p_{a-1})\xi_{a+1} + n_a(p_{a-1}-p_{a+1})\tilde{\lambda}_{a+1}\,, \tag{3.61}$$

where

$$p_{a+1}-p_{a-1} = h_a + h_{a+1}\in\mathbb{Z}\,, \tag{3.62}$$

as it follows from (3.10), (3.11). This shows that event horizons have lens space topology

$$L\left(|n_a(p_{a+1}-p_{a-1})|,\; 1+\mathsf{a}(p_{a+1}-p_{a-1})\right)\,, \tag{3.63}$$

where $\mathsf{a}$ is an integer such that[12]

$$\mathsf{a}(p_a-p_{a+1}) + \mathsf{b}\,n_a(p_{a-1}-p_{a+1}) = 1\,, \qquad \mathsf{a},\,\mathsf{b}\in\mathbb{Z}\,. \tag{3.64}$$

---

[11]The rod vectors $\xi_{I_0} = \partial_\phi - \partial_\psi$, $\xi_{I_s} = \partial_\phi + \partial_\psi$ associated with the the semi-infinite rods $I_0$, $I_s$ have non-compact bolts with topology $S^1\times\mathbb{R}^2/\mathbb{Z}_{|h_a|}$ (with $a\in\{1,s\}$) if they are adjacent to a bubbling rod, or $S^1\times\mathbb{R}^2$ if they are adjacent to a horizon rod.

[12]We have used that $\mathsf{a}(p_a-p_{a-1}) = \mathsf{a}(p_{a+1}-p_{a-1}) + \mathsf{a}(p_a-p_{a+1}) = 1 + \mathsf{a}(p_{a+1}-p_{a-1}) - \mathsf{b}\,n_a(p_{a-1}-p_{a+1})$, together with the homeomorphism between the lens spaces $L(\mathsf{p},\mathsf{q})\simeq L(\mathsf{p},\mathsf{q}+\mathsf{p}\,\mathbb{Z})$ when writing (3.63).

In the special case where $p_{a+1} - p_{a-1} = 0$, which implies $\xi_a = \xi_{a+1}$, the topology becomes that of $S^1 \times S^2$. When the orbifold is trivial, namely for $n_a = \pm 1$, the horizon reduces to the lens space $L(|p_{a+1} - p_{a-1}|, 1) \simeq S^3/\mathbb{Z}_{|p_{a+1} - p_{a-1}|}$. If then $p_{a+1} - p_{a-1} = h_a + h_{a+1} = 1$, the horizon has $S^3$ topology.

**horizon$_{a-1}$-BUBBLE$_a$-horizon$_{a+1}$.** Next, we consider a bubbling rod whose endpoints coincide with the south pole of a horizon rod $I_{a-1}$ and the north pole of another horizon rod $I_{a+1}$. In this case the Killing vectors smoothly contracting at the endpoints of the rod are, respectively,

$$
\begin{aligned}
\xi_a &= \frac{\beta n_{a-1}}{2\pi} \partial_\tau + \left[ 2(p_{a-1} - p_a) + n_{a-1}\left(1 + \frac{i\omega_-}{2\pi}\right) \right] \partial_\psi, \\
\xi_{a+1} &= \frac{\beta n_{a+1}}{2\pi} \partial_\tau + \left[ 2(p_{a+1} - p_a) + n_{a+1}\left(1 + \frac{i\omega_-}{2\pi}\right) \right] \partial_\psi.
\end{aligned}
\tag{3.65}
$$

These are related as

$$
(p_a - p_{a+1})\xi_a = (p_a - p_{a-1})\xi_{a+1} + [n_{a-1}(p_a - p_{a+1}) - n_{a+1}(p_a - p_{a-1})]\tilde{\lambda}_{a+1}, \tag{3.66}
$$

where

$$
n_{a-1}(p_a - p_{a+1}) - n_{a+1}(p_a - p_{a-1}) = -n_{a+1}h_a - n_{a-1}h_{a+1} \in \mathbb{Z}, \tag{3.67}
$$

as it follows from (3.10). According to our previous discussion, this gives the smooth lens space

$$
L(|n_{a-1}(p_a - p_{a+1}) - n_{a+1}(p_a - p_{a-1})|, \ 1 + \mathtt{a}(p_{a+1} - p_{a-1})), \tag{3.68}
$$

with

$$
\mathtt{a}(p_a - p_{a+1}) + \mathtt{b}[n_{a-1}(p_a - p_{a+1}) - n_{a+1}(p_a - p_{a-1})] = 1, \qquad \mathtt{a}, \mathtt{b} \in \mathbb{Z}. \tag{3.69}
$$

**horizon$_{a-1}$-BUBBLE$_a$-bubble$_{a+1}$, or bubble$_{a-1}$-BUBBLE$_a$-horizon$_{a+1}$.** These bubbling rods connect either a horizon north pole to a bubbling center or a bubbling center to a horizon south pole, respectively. We focus on the former case, **horizon$_{a-1}$-BUBBLE$_a$-bubble$_{a+1}$**, the discussion of the other case being equivalent. The two $2\pi$-periodic Killing vectors contracting at the endpoints of the rod $I_a$ are

$$
\begin{aligned}
\xi_a &= \frac{\beta n_{a-1}}{2\pi} \partial_\tau + \left[ 2(p_{a-1} - p_a) + n_{a-1}\left(1 + \frac{i\omega_-}{2\pi}\right) \right] \partial_\psi, \\
\xi_{a+1} &= 2\partial_\psi.
\end{aligned}
\tag{3.70}
$$

Using the expression in (3.50), we find the relation

$$
\xi_a = (p_{a-1} - p_a)\xi_{a+1} + n_{a-1}\tilde{\lambda}_{a+1}, \tag{3.71}
$$

giving the lens space

$$
L(|n_{a-1}|, p_{a-1} - p_a). \tag{3.72}
$$

We note, however, that there is a conical singularity at the pole associated to the bubbling center (if $|h_{a+1}| \neq 1$, as it follows from (3.48)), giving a branched lens space. For $n_{a-1} = \pm 1$, the bolt reduces to a branched $S^3$.

**bubble$_{a-1}$-BUBBLE$_a$-bubble$_{a+1}$.** Finally, we consider a bubbling rod connecting two bubbling centers. The vector contracting at both ends of the rod $I_a$ is given by

$$\xi_a = \xi_{a+1} = 2\partial_\psi, \tag{3.73}$$

corresponding, in the notation of (3.58), to the case $\mathsf{q}_1 = \mathsf{q}_2 = 1$ and $\mathsf{p} = 0$. From (3.48) we see that the geometry near the center $a$ exhibits a conical singularity of the form $\mathbb{R}^2/\mathbb{Z}_{|h_a|}$. Similarly, a conical singularity $\mathbb{R}^2/\mathbb{Z}_{|h_{a+1}|}$ arises at the other center $a+1$. Then the bubbling bolt has the topology of

$$S^1 \times \Sigma_{[h_a, h_{a+1}]}, \tag{3.74}$$

where the circle $S^1$ is generated by $\partial_{\tau_a}$, and $\Sigma_{[h_a, h_{a+1}]}$ denotes a *spindle* geometry. If either $|h_a| = 1$ or $|h_{a+1}| = 1$, then there is no conical singularity and space caps off smoothly at the corresponding center.

## 3.4 Global properties of the gauge field

Since the supergravity gauge field is also turned on, the rod structure as discussed so far is not sufficient to characterize the solution: additional data encoded in certain gauge fluxes through compact submanifolds in the interior of the spacetime should be specified. Indeed, as we showed above, in the Euclidean setup we consider, a five-dimensional space $\mathcal{M}$ is allowed to have non-trivial compact three-cycles outside the horizons. Therefore, denoting by $\mathcal{B}_a$ the bolt associated to the rod $I_a$, to any compact bolt ($a = 1, 2, ..., s - 1$) we can associate the electric flux

$$\mathcal{Q}^{I_a} \sim \int_{\mathcal{B}_a} \left[ \star F - \frac{\mathrm{i}}{\sqrt{3}} A \wedge F \right]. \tag{3.75}$$

We will fix the overall normalization in a convenient way later in this section. The integrand appearing coincides with the Maxwell three-form, which is closed on-shell. In general, when the gauge field cannot be globally defined over $\mathcal{B}_a$, including the rod endpoints where the bolt caps off, the integral (3.75) needs to be carefully defined patchwise.

Any three-dimensional bolt defines a normal two-dimensional non-compact cycle that intersects the bolt once. In our construction this is obtained by foliating the circle generated by $\xi_{I_a}$ over the radial direction $\rho$ ending on the bolt (at $\rho = 0$), with metric close to the tip given by (3.24). In the following, we will denote by $\mathcal{N}_a \simeq \mathbb{R}^2$ this normal space. The $\mathcal{N}_a$ dual to the bolt $\mathcal{B}_a$ can support a non-trivial gauge flux, defining a potential given by the gauge-invariant expression

$$\Phi^{I_a} \sim \int_{\mathcal{N}_a} F. \tag{3.76}$$

In general, this integral can be computed by means of the BVAB localization theorem [55,56].[13] To do so, we introduce the equivariantly closed and gauge-invariant polyform

$$\Psi^{I_a}_{[F]} = F - \iota_{\xi_{I_a}} A + \iota_{\xi_{I_a}} A \big|_{\rho \to +\infty}, \tag{3.77}$$

---

[13]Let $(\mathcal{M}, g)$ be a Riemannian $D$-dimensional manifold with boundary $\partial\mathcal{M}$, and $\xi$ be a Killing vector generating an infinitesimal isometry. An *equivariant cohomology* can be defined from the $\xi$-equivariant differential $\mathrm{d}_\xi = \mathrm{d} - \iota_\xi$. The localization theorem of equivariant cohomology, due to Berline-Vergne-Atiyah-Bott (BVAB), when applied to Riemannian manifolds with boundaries, states that the integral over $\mathcal{M}$ of a polyform $\Psi = \sum_n \Psi_{(n)}$, which is equivariantly closed, i.e. $\mathrm{d}_\xi \Psi = 0$, receives contributions only from the fixed-point locus, $\mathcal{M}_0$, of the symmetry generated by $\xi$, up to a set of terms on $\partial\mathcal{M}$:

$$\int_{\mathcal{M}} \Psi = \int_{\mathcal{M}_0} \frac{\iota^* \Psi}{e_\xi(\mathcal{N})} - \sum_{j=0}^{\left[\frac{D-2}{2}\right]} \int_{\partial\mathcal{M}} \eta \wedge (\mathrm{d}\eta)^j \wedge \Psi_{(D-2-2j)},$$

where $\iota^*$ denotes the pullback over $\mathcal{M}_0$, $e_\xi(\mathcal{N})$ is the equivariant Euler form of the normal bundle $\mathcal{N}$ to $\mathcal{M}_0$, and $\eta = |\xi|^{-2}\xi_\mu \mathrm{d}x^\mu$.

such that

$$\Phi^{I_a} \sim \int_{\mathcal{N}_a} \Psi_{[F]}^{I_a}.$$
(3.78)

By means of the localization theorem we reduce the integral above to the fixed points of $\xi_{I_a}$, namely at the tip of $\mathcal{N}_a$,

$$\Phi^{I_a} \sim -2\pi \left[\iota_{\xi_{I_a}} A\big|_{\rho \to 0} - \iota_{\xi_{I_a}} A\big|_{\rho \to +\infty}\right] \sim 2\sqrt{3}\pi\, i \left(i\breve{A}_{I_a} - i\breve{\omega}_{I_a}\right),$$
(3.79)

where $i\breve{A}_{I_a}$ denotes the $\phi$-component of the one-form $i\breve{A}$ evaluated along the rod $I_a$, whose explicit expression is

$$i\breve{A}_{I_a} = \sum_{b \leq a} \mathsf{k}_b - \sum_{b > a} \mathsf{k}_b = 2\sum_{b \leq a} \mathsf{k}_b.$$
(3.80)

As we will explain below, if $I_a$ is a horizon rod then the definitions of the electric fluxes (3.75) and the potentials (3.76) reduce to the usual ones of electric charges and electrostatic potentials associated to each horizon. However, also in case $I_a$ is not a horizon rod the above fluxes have been argued to play a role (at least semiclassically) in the first law of thermodynamics [34, 35].

For the moment we will specialize the above definitions to the subclass of single-horizon solutions, for which the thermodynamic relations found in [35] apply. The same expressions for fluxes and potentials can also be considered in the case of multi-horizon solutions, though the thermodynamical interpretation in that context is less clear.

First of all, we should compute the electrostatic potential of the horizon, denoted as $\varphi^{I_a}$, which is related to the boundary condition $\varphi$. Analogously to (2.38), we define the electrostatic potential associated to the horizon rod as

$$\varphi^{I_a} = -i \int_{\partial \mathcal{N}_a} \left(A_a - \sqrt{3}\, i\, d\tau\right), \qquad I_a : \text{horizon rod},$$
(3.81)

where $A_a$ is a regular gauge field in a patch that contains the horizon and extends up to the boundary. Here, $\partial \mathcal{N}_a$ is the asymptotic circle generated by the orbits of the rod vector $\xi_{I_a}$, being $\mathcal{N}_a$ the associated normal space. The integral (3.81) can be mapped, using Stokes' theorem, to a flux integral of the form (3.76):

$$\varphi^{I_a} + \sqrt{3} \int_{\partial \mathcal{N}_a} d\tau = -i \int_{\mathcal{N}_a} F,$$
(3.82)

which can be computed following our equivariant argument discussed above, giving

$$\varphi^{I_a} + \sqrt{3} \int_{\partial \mathcal{N}_a} d\tau = 2\sqrt{3}\,\pi \left(i\breve{A}_{I_a} - i\breve{\omega}_{I_a}\right) = 4\sqrt{3}\,\pi \left(\sum_{b \leq a} \mathsf{k}_b + \frac{n_a \beta}{4\pi}\right).$$
(3.83)

By performing the integral equivariantly we then found a relation between $\varphi^{I_a}$, that relates to a boundary integral (on the left-hand side), and a bulk quantity (on the right-hand side) – the coefficients of the one-forms $i\breve{A}$ and $i\breve{\omega}$ at the horizon rod $I_a$. This relation provides an example of a UV/IR relation (see [57]) obtained through BVAB localization theorem [58]. As a consequence, the electrostatic potential at the horizon is given by

$$\varphi^{I_a} = 4\sqrt{3}\,\pi \sum_{b \leq a} \mathsf{k}_b.$$
(3.84)

We now wish to relate $\varphi^{I_a}$ to the boundary condition $\varphi$, defined in (2.38). The regular gauge field in the patch that contains the horizon rod and extends smoothly up to the asymptotic three-sphere is found by solving the following three conditions

$$\iota_{\xi_{I_a}}\left(\mathrm{i}\breve{A}+\mathrm{d}\alpha_a\right)\Big|_{I_a} = 0 = \iota_{\xi_{I_{0,s}}}\left(\mathrm{i}\breve{A}+\mathrm{d}\alpha_a\right)\Big|_{I_{0,s}}, \qquad I_a : \text{horizon rod}, \tag{3.85}$$

for a closed one-form

$$\mathrm{d}\alpha_a = \alpha_a^{(\tau)}\mathrm{d}\tau + \alpha_a^{(\phi)}\mathrm{d}\phi + \alpha_a^{(\psi)}\mathrm{d}\psi. \tag{3.86}$$

Eqs. (3.85) are solved by taking

$$\alpha_a^{(\phi)} = 0 = \alpha_a^{(\psi)}, \qquad \alpha_a^{(\tau)} = -\frac{4\pi}{n_a\beta}\sum_{b\leq a}\mathsf{k}_b. \tag{3.87}$$

Then, the regular gauge field asymptotes to

$$A_a \to A_a^{(\infty)} = \mathrm{i}\sqrt{3}\left(1+\frac{4\pi}{n_a\beta}\sum_{b\leq a}\mathsf{k}_b\right)\mathrm{d}\tau \equiv \mathrm{i}\Phi^{I_a}\mathrm{d}\tau. \tag{3.88}$$

Here, we introduced the horizon potential $\Phi^{I_a}$, measuring the boundary holonomy of the gauge field along the orbits generated by the rod vector $\xi_{I_a}$.[14]

Finally, to compare (3.84) to (2.38), recall that the rod vector generating the horizon can be expressed as (3.3), showing that its orbits wind around the thermal circle generated by the first vector of (2.35) $n_a$ times. Then, using the expression for the asymptotic gauge field (3.88), we conclude that $\varphi^{I_a}$ is related to the boundary condition $\varphi$ as

$$\varphi = \frac{\varphi^{I_a}}{n_a} = \frac{4\sqrt{3}\pi}{n_a}\sum_{b\leq a}\mathsf{k}_b. \tag{3.89}$$

In order to compute the flux $\mathcal{Q}^{I_a}$ (3.75) across the bolt $\mathcal{B}_a$ when $I_a$ is a horizon rod, we recall that one can always find a spatial hyper-surface $\Sigma_\tau$ (in homological terms, a four-dimensional chain) whose boundary is the disconnected union of the horizon bolt (at $\rho \to 0$) and an asymptotic three-sphere (at $\rho \to +\infty$), $\partial\Sigma_\tau = \mathcal{B}_a \cup S^3_\infty$ (see e.g. [59]). Therefore, by interpreting the flux integral

$$\mathcal{Q}^{I_a} = -\frac{\mathrm{i}}{16\pi}\int_{\mathcal{B}_a}\left[\star F - \frac{\mathrm{i}}{\sqrt{3}}A_a \wedge F\right], \qquad I_a : \text{horizon}, \tag{3.90}$$

as a Page-charge, this turns out to be independent of which specific $\rho = \text{const}$ slice of $\Sigma_\tau$ we choose as integration manifold. In particular, the flux can be evaluated across the asymptotic three-sphere, showing the equivalence with the standard definition of the electric charge

$$Q = -\frac{\mathrm{i}}{16\pi}\int_{S^3_\infty}\star F, \tag{3.91}$$

using that $A \wedge F$ is suppressed asymptotically.

---

[14]Note that this quantity could also be computed by an integral of the form (3.76):

$$\mathrm{i}\Phi^{I_a} = \frac{1}{n_a\beta}\int_{\mathcal{N}_a}F, \qquad I_a : \text{horizon rod}.$$

We next focus on non-horizon rods. In this case we fix the normalization of the integrals (3.76) and (3.75) so as to match the conventions of [35]:[15]

$$\Phi^{I_a} = -\frac{1}{2\pi}\int_{\mathcal{N}_a} F\,, \qquad \mathcal{Q}^{I_a} = -\frac{1}{8}\int_{\mathcal{B}_a}\left[\star F - \frac{i}{\sqrt{3}}A_a \wedge F\right]\,, \qquad I_a : \text{bubble.} \tag{3.92}$$

The *bubble potential* $\Phi^{I_a}$, whose definition, except for the chosen normalization, is analogous to the one for the electrostatic potential, can be computed using the equivariant localization theorem as above, obtaining

$$\Phi^{I_a} = -2\sqrt{3}\,i\sum_{b\leq a}k_b\,. \tag{3.93}$$

Note that the bubble potentials associated with the semi-infinite rods vanish, $\Phi^{I_0} = \Phi^{I_s} = 0$.

Finally, eq. (3.93) may be combined with (3.84) by recalling that if $I_a$ is a horizon rod, then $I_{a-1}$ is not. Doing so, one finds a relation between an electrostatic potential $\varphi^{I_a}$ and the bubble potential associated to the adjacent bubbling rod, $\Phi^{I_{a-1}}$,

$$\varphi^{I_a} = 4\sqrt{3}\pi\,k_a + 2\pi i\,\Phi^{I_{a-1}}\,, \qquad I_a : \text{horizon}, \qquad I_{a-1} : \text{bubble.} \tag{3.94}$$

## 3.5 Thermodynamics of single-horizon solutions

For solutions with a single horizon, a first law of black hole mechanics which takes into account the presence of topologically non-trivial cycles was presented in [35]. In this section, we consider the supersymmetric versions of these thermodynamical relations, and at the same time we extend them to account for the orbifold solutions constructed above (with $n_a \neq \pm 1$). In the remaining part of the section we will denote the horizon rod by $I_a = \mathcal{H}$, and we will also use the label "$\mathcal{H}$" to characterize quantities associated with the horizon.

The Bekenstein-Hawking entropy associated to the horizon rod, defined as $1/4$ the area of the corresponding bolt, reads

$$\mathcal{S} = \pi\beta\,\delta^{\mathcal{H}}\,, \tag{3.95}$$

where $\delta^{\mathcal{H}}$ is the length of the horizon rod. Translated into our conventions, the first law of [35] reads

$$\beta\,\mathrm{d}E = \mathrm{d}\mathcal{S} + \beta\,\Phi^{\mathcal{H}}\mathrm{d}Q + \frac{\omega_+^{\mathcal{H}}}{n_{\mathcal{H}}}\mathrm{d}J_+ + \frac{\omega_-^{\mathcal{H}}}{n_{\mathcal{H}}}\mathrm{d}J_- + \sum_{I_a\neq\mathcal{H}}\mathcal{Q}^{I_a}\mathrm{d}\Phi^{I_a}\,. \tag{3.96}$$

Here, the charges entering the formula are the asymptotic ones given in (2.26) and (2.29), while the horizon angular velocities can be read from (3.13). The electrostatic potential $\Phi^{\mathcal{H}}$ has been defined in (3.88). The unusual ingredient arising because of the non-trivial topology of the solution is the last sum over the non-horizon rods, $I_a \neq \mathcal{H}$. Imposing supersymmetry,

$$E = \sqrt{3}\,Q\,, \tag{3.97}$$

gives us a *supersymmetric version of the first law*,

$$\mathrm{d}\mathcal{S} + \frac{1}{n_{\mathcal{H}}}\left(\varphi^{\mathcal{H}}\mathrm{d}Q + \omega_+^{\mathcal{H}}\mathrm{d}J_+ + \omega_-^{\mathcal{H}}\mathrm{d}J_-\right) + \sum_{I_a\neq\mathcal{H}}\mathcal{Q}^{I_a}\mathrm{d}\Phi^{I_a} = 0\,. \tag{3.98}$$

---

[15]Indeed, $\mathcal{Q}^{I_a}$ is related to the *magnetic flux* introduced in [35]. In the Euclidean finite-$\beta$ setup, the bolt $\mathcal{B}_a$ is a three-cycle, while in the Lorentzian case it reduces to a two-cycle formed by the spatial sections of $\mathcal{B}_a$, times the non-compact time direction. As discussed in section 3.3, this two-cycle can be a disc, a tube, or a spindle. The magnetic flux is defined in [35] by integrating $\iota_t(\star F - \frac{i}{\sqrt{3}}A_a \wedge F)$ over this two-cycle, and corresponds to the Lorentzian counterpart of our integral. In other words, our $\mathcal{Q}^{I_a}$ provides an uplifted Euclidean version of these fluxes, from which it differs by a factor of $\beta$ due to the integration over $\tau$. We also note that the integrals (3.92) are different from those considered in the Lorentzian analysis of [36, 37].

The analysis of [35] also provides a Smarr formula,

$$\beta E = \frac{3}{2}\mathcal{S} + \frac{3}{2n_{\mathcal{H}}}\left(\omega_+^{\mathcal{H}} J_+ + \omega_-^{\mathcal{H}} J_-\right) + \beta\,\Phi^{\mathcal{H}} Q + \frac{1}{2}\sum_{I_a \neq \mathcal{H}} \mathcal{Q}^{I_a}\Phi^{I_a}\,, \tag{3.99}$$

whose supersymmetric version reads

$$\mathcal{S} + \frac{1}{n_{\mathcal{H}}}\left(\omega_+^{\mathcal{H}} J_+ + \omega_-^{\mathcal{H}} J_- + \frac{2}{3}\varphi^{\mathcal{H}} Q\right) + \frac{1}{3}\sum_{I_a \neq \mathcal{H}}\mathcal{Q}^{I_a}\Phi^{I_a} = 0\,. \tag{3.100}$$

This relation can be combined with the quantum statistical relation satisfied by the orbifold saddle,

$$\begin{aligned}
I &= -\mathcal{S} - \frac{1}{n_{\mathcal{H}}}\left(\omega_+^{\mathcal{H}} J_+ + \omega_-^{\mathcal{H}} J_- + \varphi^{\mathcal{H}} Q\right) \\
&= -\mathcal{S} - \frac{2\pi\mathrm{i}}{n_{\mathcal{H}}}\left(J_+ + (1 - n_{\mathcal{H}} - 2p_{\mathcal{H}})J_-\right) - \omega_- J_- - \varphi Q\,,
\end{aligned} \tag{3.101}$$

to derive a simplified expression for the on-shell action that depends only on gauge field data, namely the conserved electric charge and various fluxes:

$$I = \frac{1}{3}\left[-\frac{\varphi^{\mathcal{H}}}{n_{\mathcal{H}}}Q + \sum_{I_a \neq \mathcal{H}}\mathcal{Q}^{I_a}\Phi^{I_a}\right]. \tag{3.102}$$

We will verify the validity of this formula in the explicit examples discussed in the following sections.

The quantum statistical relation (3.101) and the first law (3.98) together imply differential relations for the on-shell action. This is obtained by allowing small enough variations of the continuous boundary conditions $\omega_-$ and $\varphi$ while keeping fixed the discrete integers $(n_a, p_a)$ introduced above. According to (3.13) and (3.89), this corresponds to the variations

$$\omega_-^{\mathcal{H}} \to \omega_-^{\mathcal{H}} + n_{\mathcal{H}}\mathrm{d}\omega_-\,, \qquad \varphi^{\mathcal{H}} \to \varphi^{\mathcal{H}} + n_{\mathcal{H}}\mathrm{d}\varphi\,, \qquad \Phi^{I_a} \to \Phi^{I_a} + \mathrm{d}\Phi^{I_a}\,. \tag{3.103}$$

Assuming the first law holds, we then find

$$\mathrm{d}I = -J_-\,\mathrm{d}\omega_- - Q\,\mathrm{d}\varphi + \sum_{I_a \neq \mathcal{H}}\mathcal{Q}^{I_a}\,\mathrm{d}\Phi^{I_a}\,. \tag{3.104}$$

Therefore, the charges $Q$, $J_-$ and the electric fluxes $\mathcal{Q}^{I_a}$ can also be determined from the on-shell action by taking the partial derivatives with respect to the conjugate variables

$$J_- = -\frac{\partial I}{\partial\,\omega_-}\,, \qquad Q = -\frac{\partial I}{\partial\,\varphi}\,, \qquad \mathcal{Q}^{I_a} = \frac{\partial I}{\partial\,\Phi^{I_a}}\,, \qquad I_a \neq \mathcal{H}\,. \tag{3.105}$$

## 3.6 Quantization conditions for a compact gauge group

In this section, we derive quantization conditions for the bubble potentials (3.93) and allowed shifts for the electrostatic potentials (generalizing (3.89)), arising in case of compact abelian gauge group. This leads us to introduce a new integer, $q_a$, for each compact rod $I_a$.

In the gravitational path integral we sum over all field configurations $\mathcal{X}$, either bosonic or fermionic, that satisfy the following periodic identifications when going once around the three independent U(1)'s generated by the Killing vectors in (2.35):

$$\begin{aligned}
\mathcal{X}(\tau + \beta, \phi + 2\pi, \psi + \mathrm{i}\omega_-) &\sim (-1)^{\mathsf{F}}\mathcal{X}(\tau, \phi, \psi)\,, \\
\mathcal{X}(\tau, \phi + 2\pi, \psi - 2\pi) &\sim (-1)^{\mathsf{F}}\mathcal{X}(\tau, \phi, \psi)\,, \\
\mathcal{X}(\tau, \phi, \psi + 4\pi) &\sim \mathcal{X}(\tau, \phi, \psi)\,.
\end{aligned} \tag{3.106}$$

These identifications are intended to be valid when working in a gauge for the vector field $A$ in (2.3) specified by the choice $\alpha = -\frac{\varphi}{\sqrt{3}\beta}\,\tau$, in which the gauge field $A$ asymptotes to

$$A \to i\left(\frac{\varphi}{\beta} + \sqrt{3}\right)d\tau, \qquad \rho \to \infty, \tag{3.107}$$

where $\varphi$ is the boundary condition defined in (2.38).

The identifications (3.106) must be compatible with the requirement that the geometry supports a smooth spin structure. In our setup, this means that any field must pick a $(-1)^{\mathsf{F}}$ phase when transported around the orbits generated by any rod vector $\xi_{I_a}$, as long as one works in a gauge that is regular in a patch containing the rod $I_a$, where $\xi_{I_a}$ contracts. More concretely, we must impose

$$\mathcal{X}_a\left(\tau + n_a\beta,\, \phi + 2\pi,\, \psi + 2\pi\left[2p_a - 1 + n_a\left(1 + \frac{i\omega_-}{2\pi}\right)\right]\right) \sim (-1)^{\mathsf{F}}\mathcal{X}_a(\tau, \phi, \psi), \tag{3.108}$$

where the subscript $\mathcal{X}_a$ means that we are working in the aforementioned regular gauge. The gauge transformation relating $A$ and $A_a$ is given by

$$A_a = A - i\sqrt{3}\left(d\alpha_a + \frac{\varphi}{\sqrt{3}\beta}d\tau\right), \tag{3.109}$$

where we recall that $\alpha_a$ is a function of the form $\alpha_a = \alpha_a^{(\tau)}\tau + \alpha_a^{(\phi)}\phi + \alpha_a^{(\psi)}\psi$. This implies that

$$\mathcal{X}_a(\tau, \phi, \psi) = \exp\left[\sqrt{3}\,e_{\mathcal{X}}\left(\alpha_a + \frac{\varphi}{\sqrt{3}\beta}\tau\right)\right]\mathcal{X}(\tau, \phi, \psi), \tag{3.110}$$

where $e_{\mathcal{X}}$ is the charge of the field $\mathcal{X}$ under the U(1) gauge symmetry. Compatibility between (3.108) and the identifications (3.106) is equivalent to demanding the transition function in (3.110) to be single-valued when going once around the U(1) generated by $\xi_{I_a}$. We note that in ungauged supergravity these conditions are trivially satisfied since all supergravity fields are uncharged under the U(1) gauge symmetry. However, if we consider the UV completion of the theory — e.g., via an embedding in string theory — then a microscopic description must exist in which microstates charged under the gauge symmetry are present to account for the entropy. Typically, in these examples the gauge group turns out to be a compact U(1), then the states have quantized charges of the type $e_{\mathcal{X}} \in \mathbb{Z}e$, being $e$ the fundamental unit charge in the theory. Then, under these assumptions, the requirements above provide a quantization condition for each rod $I_a$, as we will now discuss considering separately bubbling and horizon rods.

The transition function in (3.110) is single-valued when going once around the U(1) generated by $\xi_{I_a}$ if the following condition is obeyed

$$n_a\varphi + 2\pi\sqrt{3}\,\iota_{\xi_{I_a}}d\alpha_a = 2\pi i\frac{q_a}{e}, \qquad q_a \in \mathbb{Z}. \tag{3.111}$$

For bubbling rods, $n_a = 0$, we can use (3.93) to find

$$\iota_{\xi_{I_a}}d\alpha_a = -2\sum_{b \leq a}\mathsf{k}_b = \frac{\Phi^{I_a}}{\sqrt{3}i}. \tag{3.112}$$

Therefore, eq. (3.111) applied to a bubbling rod yields quantization conditions:

$$e\,\Phi^{I_a} = q_a, \qquad I_a : \text{bubble}. \tag{3.113}$$

On the other hand, for a horizon rod, $n_a \neq 0$, we use (3.84) and (3.85) to derive

$$2\pi\sqrt{3}\,\iota_{\xi_{I_a}}\mathrm{d}\alpha_a = -\varphi^{I_a}\,. \tag{3.114}$$

Using this in (3.111), we conclude that the electrostatic potentials assigned to each horizon $\varphi^{I_a}$ are related to the boundary condition $\varphi$ via

$$\varphi^{I_a} = n_a\varphi - 2\pi\mathrm{i}\frac{q_a}{e}\,, \qquad I_a : \text{horizon}. \tag{3.115}$$

For a horizon rod, we can further use (3.94) to find

$$\mathsf{k}_a = \frac{-\mathrm{i}}{2\sqrt{3}e}\left(q_a + q_{a-1}\right) + \frac{n_a\varphi}{4\pi\sqrt{3}}\,, \tag{3.116}$$

and

$$\mathsf{k}_a + \mathsf{k}_{a+1} = \frac{q_{a-1} - q_{a+1}}{2\sqrt{3}\mathrm{i}e}\,, \tag{3.117}$$

analogously to the expressions we found in (3.10) and (3.11) for the coefficients of the harmonic functions $H$ and $\omega_-$.

We have thus extended the map between the horizon electrostatic potential (3.84) and the boundary condition $\varphi$, previously given by (3.89), by including integer shifts $q_a$. We now examine how these quantized parameters $q_a$ modify the quantum statistical relation (3.101) and its differential form (3.104). Specializing to the case where there is only a single horizon, the quantum statistical relation takes the form

$$\begin{aligned}
I &= -\mathcal{S} - 2\pi\mathrm{i}\tilde{J}_+ - \omega_- J_- - \varphi\,Q\,, \\
\tilde{J}_+ &= \frac{1}{n_{\mathcal{H}}}\Big(J_+ + (1 - n_{\mathcal{H}} - 2p_{\mathcal{H}})J_- - \frac{q_{\mathcal{H}}}{e}Q\Big),
\end{aligned} \tag{3.118}$$

where we use the notation of section 3.5. Combining this with the first law (3.98) yields its differential version. A key distinction now is that the discrete parameters $n_a, p_a$, as well as $q_a/e$ are all held fixed. In particular, the bubble potentials, $\Phi^{I_a}$, are not allowed to vary anymore ($\mathrm{d}\Phi^{I_a} = 0$) due to (3.113), in contrast with the derivation leading to (3.104). As a result, the on-shell action $I$ satisfies the simple differential relation

$$\mathrm{d}I = -J_-\,\mathrm{d}\omega_- - Q\,\mathrm{d}\varphi\,, \tag{3.119}$$

which shows that, once all quantization conditions are imposed, the only continuous variables of $I$ are just the chemical potentials $\omega_-$ and $\varphi$.

# 4  Revisiting two-center solutions

In this section we revisit the two-center supersymmetric non-extremal black hole solution presented in [39, 40]. This comprises just one compact rod, with rod vector the generator of the Euclidean horizon. Our scope is twofold: on the one hand we want to motivate further the need to consider a complexified version of the solution, and on the other hand we wish to illustrate the discrete shifts and orbifolds introduced in section 3 in the simplest possible setup.

## 4.1 The complex saddle

We start by considering the basic rod vector (3.3) with $n = 1, p = 0$, that is

$$\xi_{\mathcal{H}} = \frac{\beta}{2\pi}\partial_\tau + \partial_\phi + \frac{i\omega_-}{2\pi}\partial_\psi\,, \tag{4.1}$$

where we use $\mathcal{H}$ rather than $I_1$ to label the horizon rod. Then the horizon temperature and angular velocities coincide with the asymptotic boundary conditions, specified by demanding that the first basis vector in (2.35) has closed orbits. In [39, 40], reality conditions on the parameters were chosen so that the metric is real and positive definite. This makes it straightforward to study regularity of the solution and also gives a real Euclidean on-shell action. It was noted in [40] that this finite-$\beta$ solution can interpolate between the extremal black hole of BMPV (for which $\beta = \infty$) and a horizonless topological soliton (which is reached via a $\beta \to 0$ limit), however in order to see this one needs to implement suitable analytic continuations of the parameters appearing in the real, Euclidean solution (in addition to Wick-rotating the Euclidean time back to Lorentzian signature). Therefore, in order to be able to reach the Lorentzian solutions by a continuous limit, one must give up reality of the metric and the gauge field. Here we review this argument and further motivate it. While it is not clear in general what conditions should a complex metric satisfy, the discussion in section 3 shows that cancellation of Dirac-Misner strings as well as smoothness along the collapsing orbits of the rod vector can be achieved even in the complexified setup.

The two-center solution is given by choosing the harmonic functions (2.21) as [40]

$$H = \frac{h_N}{r_N} + \frac{h_S}{r_S}\,, \qquad \text{with} \quad h_N + h_S = 1\,,$$

$$iK = \mathsf{k}\left(\frac{1}{r_N} - \frac{1}{r_S}\right)\,, \qquad iM = -\frac{\mathsf{k}^3}{2}\left(\frac{1}{h_N^2 r_N} - \frac{1}{h_S^2 r_S}\right)\,, \tag{4.2}$$

$$L = 1 + \mathsf{k}^2\left(\frac{1}{h_N r_N} + \frac{1}{h_S r_S}\right)\,,$$

where $r_N, r_S$ denote the distance between the two centers (horizon north pole $N$ and south pole $S$) in the $\mathbb{R}^3$ base.

We first consider the case where the parameters $h_N, \mathsf{k}, \delta$ are all real (with $\delta > 0$). Then the functions $H, L$ are real, while $K, M$ are purely imaginary. The metric in (2.3) is real and positive-definite provided

$$f^{-1}H \equiv K^2 + HL > 0\,. \tag{4.3}$$

For our two-center solution, this condition reduces to

$$\frac{\mathsf{k}^2}{h_N h_S} + h_N r_S + h_S r_N > 0\,, \tag{4.4}$$

which must be satisfied for all $r_{N,S} > 0$. This requirement imposes that both coefficients of the harmonic function $H$ be positive, which is equivalent to

$$0 < h_N < 1\,. \tag{4.5}$$

The chemical potentials fixing the boundary conditions are related to the parameters as

$$\omega_+ = 2\pi i\,, \qquad \omega_- = 2\pi i(h_S - h_N)\,, \qquad \varphi = 4\sqrt{3}\pi\mathsf{k}\,, \tag{4.6}$$

$$\beta = 4\pi iw_N = -4\pi iw_S = -2\pi\mathsf{k}\left(3 + \frac{\mathsf{k}^2}{h_N^2 h_S^2 \delta}\right)\,. \tag{4.7}$$

The Euclidean on-shell action reads in terms of the chemical potentials:

$$I = \frac{\pi}{12\sqrt{3}} \frac{\varphi^3}{\omega_1 \omega_2}, \tag{4.8}$$

where we recall that $\omega_\pm = \omega_1 \pm \omega_2$. The conserved charges are:

$$J_+ = -3\pi i k \delta, \qquad J_- = \pi i k^3 \frac{h_N - h_S}{h_N^2 h_S^2}, \qquad Q = \frac{\sqrt{3}\pi k^2}{h_N h_S}. \tag{4.9}$$

The entropy, defined as $1/4$ the area of the bolt of the Killing vector (4.1), is given by

$$\mathcal{S} = \pi\beta\delta = 4\sqrt{\pi}\sqrt{\frac{Q^3}{3\sqrt{3}} - \frac{\pi}{4}J_-^2} - 2\pi i J_+. \tag{4.10}$$

We see there is a direct correspondence between the free coefficients of the harmonic functions $(h_N, k)$, the chemical potentials $(\omega_-, \varphi)$, and the charges $(J_-, Q)$ which play a role in the supersymmetric index.

However, as previously stated, the reality conditions on the parameters assumed above are too limiting. Indeed, they give a real on-shell action $I$, real $\varphi$ and purely imaginary angular velocities $\omega_\pm$, while one may expect that the chemical potentials appearing in the grand-canonical index can take more general complex values. This indicates that we have not considered sufficiently general field configurations. Also, as emphasized in [40], it is not clear if a purely imaginary value for both angular velocities $\omega_{1,2}$ is acceptable, as the fugacity $e^{\omega_2}$ appearing in the index trace (1.6) would be just a phase and the sum over microstates may not converge. Finally, the angular momenta $J_\pm$ are purely imaginary in the solution, which then cannot be directly connected to (rotating) Lorentzian solutions. So at this stage we would not be able to answer the question of whether the extremal solutions can arise as suitable limits of the index saddles. These limitations can be overcome if we allow for complexifications of the parameters. Let us briefly recall how in this way the saddle solution is connected with known Lorentzian solutions, referring to [40] for more details.

In order to reach the extremal black hole we require that the two centers merge, i.e. we take $\delta \to 0$, so that $\beta \to \infty$ as it can be seen from (4.7). Note that now the rod vector (3.3) (rescaled by $\beta$) is just $\partial_\tau$, as it should for an extremal horizon. Then $J_+ \to 0$, however the angular momentum $J_-$ remains finite and purely imaginary.[16] In order to fix this we allow $h_N$ to take complex values and impose the conditions

$$h_S = h_N^*, \qquad k \in \mathbb{R}. \tag{4.11}$$

In this way all conserved quantities (4.9) are real. Then one can check that, after Wick-rotating back to Lorentzian time by taking $\tau = it$, this gives precisely the real solution of BMPV (here in the case of just one independent electric charge), parameterized by $\text{Im}\, h_N$, k.

A different limit leads to a topological soliton, namely a Lorentzian solution having a non-trivial topology despite the absence of a horizon. This is obtained by taking $\beta \to 0$ while keeping $\omega_1, \omega_2, \varphi$ finite. From the middle expression in (4.10) we see that then the entropy vanishes, as it should in a horizonless solution. However, from the expression for $\beta$ in (4.7) we see that $\beta = 0$ is not possible for a positive distance $\delta$, unless we analytically continue $k = ik$ while holding $h_N, h_S$ real, which allows us to solve the condition by setting $\delta = \frac{k^2}{3h_N^2 h_S^2}$ (in Lorentzian signature, this corresponds to solving a bubble equation). This also gives real $J_\pm$ and $Q$. Then, Wick-rotating $\tau = it$, we recover the Lorentzian horizonless solution of [60, 61] (in the case of equal charges). Notice that in this $\beta \to 0$ limit the identifications (2.34) only

---

[16]Except in the case $h_N = h_S = 1/2$, which gives $J_\pm = 0$, that is the static black hole of Strominger-Vafa.

involve the $\phi, \psi$ coordinates, while $\tau$ becomes a mere spectator. Also, the rod vector (4.1) does not involve $\partial_\tau$ anymore and thus transforms into a bubbling rod vector of the type (3.5). The mutual compatibility between these identifications imposes the quantization condition $\omega_- = 2\pi i (1 - 2p)$, with $p \in \mathbb{Z}$, which is equivalent to the quantization of the solution parameters, $h_N \in \mathbb{Z}$ (hence $h_S \equiv 1 - h_N \in \mathbb{Z}$). Another feature of this $\beta \to 0$ limit is that the solution develops orbifold singularities, $\mathbb{C}^2/\mathbb{Z}_{|h_N|}$ at the north pole, and $\mathbb{C}^2/\mathbb{Z}_{|h_S|}$ at the south pole [60, 61].

Given the motivations above, we are led to consider complexifications of the two-center solution. In particular, the parameters $h_N, h_S = 1 - h_N$, k take complex values. In this way, $\omega_-, \varphi$ and $\beta$ are generically complex, as well as the other thermodynamic quantities, namely, the action, the charges and the entropy. This has the advantage to straightforwardly allow for the two different limits giving rise to well-behaved Lorentzian solutions reviewed above.

The complex solutions have the same asymptotic behavior as (2.30), however the coordinates $(\tau, \phi, \psi)$ should be taken complex now, since the identifications (2.34) involve complex periodicities. As argued in section 3, one can nevertheless introduce real, untwisted angular coordinates, cf. (3.20). In the present case these are given by

$$\tau_a = \frac{2\pi}{\beta} \tau - \phi \,, \qquad \psi_a = \phi + \psi + \frac{i}{\beta}(\omega_+ - \omega_-)\tau \,, \qquad \phi_a = \phi \,, \qquad (4.12)$$

and can be used in order to analyze regularity of the metric.

## 4.2 Discrete families of saddles from shifts and orbifolds

We continue revisiting the two-center solutions by elaborating on the infinite family of shifts and orbifolds labelled by the integers $(n, p, q)$ introduced in section 3. These $(n, p, q)$ solutions satisfy the very same boundary conditions, hence they potentially contribute to the same gravitational index. They are analogous to the shifted and orbifolded solutions first discussed in [62] in the context of asymptotically $AdS_5 \times S^5$ black holes solutions to type IIB supergravity, however they have not been studied in the asymptotically flat context so far. Also in our case the Killing spinor is not charged under the supergravity gauge field, implying that the orbifold acts in the five extended dimensions and not in the internal space of any higher-dimensional uplift. This simplifies the study of the topology.

Recall that the two integers $(n, p)$ specify how the rod vector, corresponding to the horizon generator, is related to the basis of $U(1)^3$ isometries. This is illustrated in eq. (3.3). The rod vector reads (cf. (3.12))

$$\xi_{\mathcal{H}} = \frac{1}{2\pi} \left( \beta^{\mathcal{H}} \partial_\tau - i \omega_+^{\mathcal{H}} \partial_\phi + i \omega_-^{\mathcal{H}} \partial_\psi \right) , \qquad (4.13)$$

where the horizon inverse temperature and angular velocities are given by

$$\beta^{\mathcal{H}} = n \beta \,, \qquad \omega_+^{\mathcal{H}} = 2\pi i \,, \qquad \omega_-^{\mathcal{H}} = n \omega_- + 2\pi i (1 - n - 2p) \,. \qquad (4.14)$$

Also, the horizon electrostatic potential reads

$$\varphi^{\mathcal{H}} = n \varphi - 2\pi i q \,, \qquad (4.15)$$

where the shift by the additional integer $q$ should be introduced when the gauge group is $U(1)$.[17]

We will assume that both $p$ and $p - 1$ are coprime to $n$. According to the analysis in section 3 (cf. (3.63), (3.64)), the horizon has lens space topology

$$L(|n|, 1 + a) \simeq S^3/\mathbb{Z}_{|n|} \,, \qquad (4.16)$$

---

[17]Compared to the previous section we are setting $e = 1$.

where a is an integer defined through the equation

$$\mathsf{a}(p-1) - \mathsf{b}\,n = 1\,, \qquad \mathsf{a}, \mathsf{b} \in \mathbb{Z}\,. \tag{4.17}$$

The local form of the solution is exactly the same as in subsection 4.1, however the values taken by the parameters $h_N, h_S, \mathsf{k}, \delta$ in terms of the (fixed) boundary conditions now depends on $n, p, q$. Indeed, the expressions in (4.6), (4.7) now provide the horizon inverse temperature $\beta^{\mathcal{H}}$, angular velocities $\omega_{\pm}^{\mathcal{H}}$, and electrostatic potential $\varphi^{\mathcal{H}}$, which are related to the chemical potentials $\beta, \omega_{\pm}, \varphi$ as in (4.14), (4.15). Solving for $h_N, h_S, \mathsf{k}$, we obtain the expressions

$$h_N = n\frac{\omega_2}{2\pi\mathrm{i}} + p\,, \qquad h_S = n\frac{\omega_1}{2\pi\mathrm{i}} + 1 - n - p\,, \qquad \mathsf{k} = \frac{\mathrm{i}}{2\sqrt{3}}\left(n\frac{\varphi}{2\pi\mathrm{i}} - q\right)\,, \tag{4.18}$$

which are the specialization of (3.10) and (3.116) to the present case.[18] The remaining parameter, $\delta$, is given by

$$\delta = -\frac{2\pi\mathsf{k}^3}{h_N^2 h_S^2 (n\beta + 6\pi\mathsf{k})}\,, \tag{4.19}$$

where one should use (4.18) in order obtain an expression in terms of the chemical potentials and the integers $(n, p, q)$.

The saddle-point contribution of the $(n, p, q)$ solutions to the gravitational index is given by the Euclidean on-shell action. This reads

$$
\begin{aligned}
I_{n,p,q}(\omega, \varphi) &= \frac{\pi}{12\sqrt{3}} \cdot \frac{1}{n} \cdot \frac{(\varphi^{\mathcal{H}})^3}{\omega_1^{\mathcal{H}} \omega_2^{\mathcal{H}}} \\
&= \frac{\pi}{12\sqrt{3}} \frac{(n\,\varphi - 2\pi\mathrm{i}q)^3}{n[n\,\omega_1 + 2\pi\mathrm{i}(1 - n - p)][n\,\omega_2 + 2\pi\mathrm{i}p]}\,,
\end{aligned} \tag{4.20}
$$

the overall factor of $\frac{1}{n}$ being due to the action of the orbifold.

The expressions (4.9) for the angular momenta and the electric charge in terms of the solution parameters remain unchanged, however one should again recall that the map between the parameters and the chemical potentials $\beta, \omega_{\pm}, \varphi$ depends on $(n, p, q)$. The entropy, defined as 1/4 the area of the bolt of the horizon generator, is given by

$$\mathcal{S}_{n,p,q} = \frac{1}{n}\pi\beta^{\mathcal{H}}\delta = \pi\beta\delta\,. \tag{4.21}$$

In terms of the charges of the $(n, p, q)$-solution, the same expression reads

$$\mathcal{S}_{n,p,q} = \frac{1}{n}\left(4\sqrt{\pi}\sqrt{\frac{Q^3}{3\sqrt{3}} - \frac{\pi}{4}J_-^2} - 2\pi\mathrm{i}J_+\right)\,. \tag{4.22}$$

The quantum statistical relation (3.101) is then satisfied.

It would be interesting to compute non-perturbative corrections to the action of these $(n, p, q)$ solutions by evaluating the contribution of wrapped branes in a concrete scenario where the solution is uplifted to string or M-theory, as done in [62] in the asymptotically $\mathrm{AdS}_5 \times S^5$ type IIB setup. These corrections may help determine which of the solutions are stable against brane nucleation and thus be regarded as genuine saddles of the gravitational path integral.

---

[18]Note that in the regime where the metric is real and thus both $\omega_1, \omega_2$ are purely imaginary, the condition $0 < h_N < 1$ ensuring a positive-definite metric can be satisfied by fixing the integer $p$ so that $0 < n\frac{\omega_2}{2\pi\mathrm{i}} + p < 1$. This fails if $\frac{\omega_2}{2\pi\mathrm{i}} \in \mathbb{Z}$.

# 5 Three-center solutions, black ring and black lens

This section is devoted to the study of solutions with three centers, that is two compact rods. From the general analysis of section 3, we know that if one is a horizon rod then the other must be a bubbling rod, since two horizon rods cannot be next to each other. Hence, the configuration we are going to study is unique up to orientation (the presence of the additional bubbling center breaks the equatorial $\mathbb{Z}_2$ symmetry of the two-center solutions).

In section 4 the relevance of regarding the non-extremal two-center solution as a complex solution was emphasized, although there is a choice of the parameters such that the metric is real and positive definite. We find this is even more compelling in the three-center case, since it is not even possible to define a Euclidean section such that the metric is real and positive.

The section is organized as follows. First, we present the solution and provide the relations between its parameters and the grand-canonical variables, specializing the general expressions obtained in section 3 to the case with three centers. Then, we analyze in detail the thermo-dynamic properties of the solution; in particular, we compute its on-shell action. Finally, we discuss how the finite-$\beta$ solutions interpolate between extremal black holes with non-spherical horizon topology, such as black rings [18] and lenses [19,20], and horizonless topological soli-tons [21–23], which arise in the $\beta \to 0$ limit.

## 5.1 Three-center solutions

Let us begin by specifying the general class of solutions described in section 2 to the case of three centers. We take the first center, placed at $z_1 = \delta_1$, to be the bubbling center. Instead, the second and third centers, placed at $z_2 = \frac{\delta}{2}$ and $z_3 = -\frac{\delta}{2}$, will correspond to the north ($N$) and south ($S$) poles of the Euclidean horizon.

We next restrict our attention to solutions with $n_{\mathcal{H}} = 1$ and $p_{\mathcal{H}} = 0$, implying that the rod vector (3.3) degenerating at the horizon, $I_N = \mathcal{H}$, is given by

$$\xi_{\mathcal{H}} = \frac{\beta}{2\pi}\partial_\tau + \partial_\phi + \frac{i\omega_-}{2\pi}\partial_\psi\,. \tag{5.1}$$

Instead, the Killing vector contracting at the bubbling rod, $I_1$, is

$$\xi_{I_1} = \partial_\phi + (2p_1 - 1)\partial_\psi\,. \tag{5.2}$$

This rod structure is illustrated in figure 3. The harmonic functions of the three-center solution are given by

$$H = \frac{h_1}{r_1} + \frac{h_N}{r_N} + \frac{h_S}{r_S}\,, \qquad iK = \frac{k_1}{r_1} + \frac{k_N}{r_N} + \frac{k_S}{r_S}\,,$$

$$L = 1 + \frac{k_1^2}{h_1 r_1} + \frac{k_N^2}{h_N r_N} + \frac{k_S^2}{h_S r_S}\,, \qquad iM = -\frac{k_1^3}{2h_1^2 r_1} - \frac{k_N^3}{2h_N^2 r_N} - \frac{k_S^3}{2h_S^2 r_S}\,, \tag{5.3}$$

where we recall that the parameters $h_a$ and $k_a$ satisfy

$$h_1 + h_N + h_S = 1\,, \qquad k_1 + k_N + k_S = 0\,. \tag{5.4}$$

To express the coefficients of the harmonic functions in terms of the boundary conditions (and the integers characterizing the rod structure) we specify (3.8), (3.10), (3.89) and (3.93) to the case at hands, which yields

$$h_1 = p_1\,, \qquad h_N = \frac{1}{2} - p_1 - \frac{\omega_-}{4\pi i}\,, \qquad h_S = \frac{1}{2} + \frac{\omega_-}{4\pi i}\,, \tag{5.5}$$

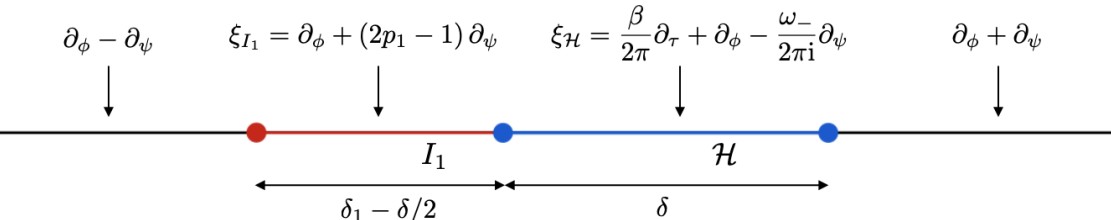

Figure 3: Rod structure of the three-center solution, with the bubbling rod $I_1$ and the horizon rod $\mathcal{H}$. For each rod we indicate the corresponding rod vector.

and

$$k_1 = -\frac{\Phi^{I_1}}{2\sqrt{3}i}, \qquad k_N = \frac{\Phi^{I_1}}{2\sqrt{3}i} + \frac{\varphi}{4\sqrt{3}\pi}, \qquad k_S = -\frac{\varphi}{4\sqrt{3}\pi}. \tag{5.6}$$

Finally, $\delta$ and $\delta_1$ are determined in terms of $\beta$ and the remaining parameters via the first in (3.26), which gives the conditions

$$w_1 = 0, \qquad \beta = 4\pi i w_N = -4\pi i w_S. \tag{5.7}$$

These can be explicitly expressed in terms of the parameters of the solution via (2.22),

$$0 = \frac{3k_1}{2h_1} + \frac{h_N}{2\delta_1 - \delta}\left(\frac{k_1}{h_1} - \frac{k_N}{h_N}\right)^3 + \frac{h_S}{2\delta_1 + \delta}\left(\frac{k_1}{h_1} - \frac{k_S}{h_S}\right)^3, \tag{5.8}$$

$$\beta = -2\pi\left[3k_N + \frac{h_N h_S}{\delta}\left(\frac{k_N}{h_N} - \frac{k_S}{h_S}\right)^3 + \frac{2h_N h_1}{2\delta_1 - \delta}\left(\frac{k_N}{h_N} - \frac{k_1}{h_1}\right)^3\right]. \tag{5.9}$$

As already discussed in section 3.3, the bolt associated with the bubbling rod is topologically a branched $S^3$. In turn, the allowed topologies for the horizon bolt are $S^1 \times S^2$ (corresponding to $p_1 = 1$), or $L(|1-p_1|, 1) \simeq S^3/\mathbb{Z}_{|1-p_1|}$.

## 5.2 Thermodynamics

The mass $E = \sqrt{3}Q$, electric charge $Q$ and angular momenta $J_\pm$ of the solutions were already given in (2.26) and (2.29). Specializing to the case at hands, one gets

$$J_+ = -3\pi i\left[k_1\left(\delta_1 + \frac{\delta}{2}\right) + k_N \delta\right], \qquad J_- = -\pi i\left(\frac{k_1^3}{h_1^2} + \frac{k_N^3}{h_N^2} + \frac{k_S^3}{h_S^2}\right),$$

$$Q = \pi\sqrt{3}\left(\frac{k_1^2}{h_1} + \frac{k_N^2}{h_N} + \frac{k_S^2}{h_S}\right), \tag{5.10}$$

where in the expression for $J_+$ we have used the second in (5.4) in order to explicitly show that it only depends on the distances between the centers. The topology of this three-center solution allows us to associate a non-trivial electric flux $\mathcal{Q}^{I_1}$ to the bubbling rod. This was defined in (3.92), reported again here for convenience,

$$\mathcal{Q}^{I_1} = -\frac{1}{8}\int_{\mathcal{B}_{I_1}}\left[\star F - \frac{i}{\sqrt{3}}A_1 \wedge F\right]. \tag{5.11}$$

Let us recall that the notation $A_1$ stands for a gauge in which the vector field is regular in a patch containing the integration region, $\mathcal{B}_{I_1}$. This amounts to fixing the closed one-form $d\alpha$

appearing in (2.3) by demanding that the contraction $\iota_{\xi_{I_1}} A_1$ vanishes at the fixed locus of any U(1) isometry degenerating inside the patch. Since this must contain $\mathcal{B}_{I_1}$, a first condition is

$$\iota_{\xi_{I_1}} A_1 \Big|_{I_1} = \iota_{\xi_{I_1}} \left( i\breve{A} + d\alpha_1 \right) \Big|_{I_1} = 0. \tag{5.12}$$

On top of this, one has two additional conditions coming from the extra U(1) isometries that degenerate at the endpoints of the bubbling rod,

$$\iota_{\xi_1} A_1 \Big|_{\rho=0, z=\delta_1} = 0, \qquad \iota_{\xi_N} A_1 \Big|_{\rho=0, z=\delta/2} = 0, \tag{5.13}$$

where

$$\xi_1 = 2\partial_\psi, \quad \text{and} \quad \xi_N = \frac{\beta}{2\pi} \partial_\tau + 2h_N \partial_\psi, \tag{5.14}$$

are the $2\pi$-periodic U(1) Killing vectors degenerating at the bubbling center and the north pole, respectively. A closed one-form $d\alpha_1$ satisfying the above regularity requirements is

$$d\alpha_1 = -\frac{k_1}{h_1}\left( d\psi + d\phi - \frac{4\pi h_N}{\beta} d\tau \right) - \frac{4\pi k_N}{\beta} d\tau. \tag{5.15}$$

A convenient way of computing $\mathcal{Q}^{I_1}$ is to note that the integral in (5.11) can be manipulated as follows:

$$\mathcal{Q}^{I_1} = -\frac{1}{8}\int_{\mathcal{B}_{I_1}} G_1 = -\frac{\pi}{4}\int_D \iota_{\tau_1} G_1 = -\frac{\beta}{8}\int_{\partial D} \nu_V, \qquad G_1 \equiv \star F - \frac{i}{\sqrt{3}} A_1 \wedge F, \tag{5.16}$$

where $D$ denotes the disc parametrized by the coordinate $z \in [\delta/2, \delta_1]$ and $\psi_1$, whose boundary $\partial D$ is the circle parametrized by $\psi_1$ at $z = \delta/2$ (to be precise, this is a disc with a conical deficit). Given a Killing vector $\xi$, the local one-form $\nu_\xi$ is defined by [40]

$$d\nu_\xi = \iota_\xi G. \tag{5.17}$$

For the specific choice $\xi = \partial_{\tau_1}$, we have

$$\nu_\xi = \frac{\beta}{2\pi}\nu_V + \left( \frac{i\omega_-}{2\pi} + 1 \right)\nu_U, \tag{5.18}$$

where $V = \partial_\tau$ and $U = \partial_\psi = \partial_{\psi_1}$. The second term in the above equation cannot contribute to the integral over $\partial D$ in (5.16), which explains why only $\nu_V$ appears in the last equality. The expression for $\nu_V$ was already computed in [40], which we just quote:

$$\nu_V = \sqrt{3} i f^2 \left( d\tau + i\omega_\psi (d\psi + \chi) + i\breve{\omega} \right) - \left( f + \alpha_1^{(\tau)} \right) A_1. \tag{5.19}$$

Thus, we have all the ingredients to evaluate the integral, which yields the following result

$$\mathcal{Q}^{I_1} = 2\sqrt{3}\pi^2 i h_N \left( \frac{k_N}{h_N} - \frac{k_1}{h_1} \right)^2. \tag{5.20}$$

Finally, the Bekenstein-Hawking entropy $\mathcal{S}$ is computed as $1/4$ the area of the horizon bolt,

$$\mathcal{S} = \pi \beta \delta. \tag{5.21}$$

Expressed in terms of the charges $Q, J_\pm$ and $\Phi^{I_1}$, it reads

$$\mathcal{S} = -2\pi i \left[ J_+ - p_1 J_- + \frac{\Phi^{I_1}}{2}\left( Q - \frac{\pi}{12\sqrt{3}}\frac{(\Phi^{I_1})^2}{p_1^2} \right) \right] \tag{5.22}$$

$$+ 4\sqrt{\pi}\sqrt{(1-p_1)\left[ \frac{Q}{\sqrt{3}} + \frac{\pi(\Phi^{I_1})^2}{12 p_1(1-p_1)} \right]^3 - \frac{\pi}{4}\left[ (p_1-1)J_- - \frac{Q\Phi^{I_1}}{2} - \frac{\pi(1+p_1)(\Phi^{I_1})^3}{24\sqrt{3} p_1^2(1-p_1)} \right]^2}.$$

This expression will be derived again via the Legendre transform of the on-shell action in appendix C.2.

Having explicitly computed all thermodynamic quantities, we can explicitly verify that the supersymmetric first law (3.98),

$$d\mathcal{S} + \varphi\, dQ + \omega_+\, dJ_+ + \omega_-\, dJ_- + \mathcal{Q}^{I_1}\, d\Phi^{I_1} = 0\,, \tag{5.23}$$

is satisfied if $h_1$ (which is an integer) is held fixed in the variation. Assuming now the quantum statistical relation

$$I = -\mathcal{S} - \varphi\, Q - \omega_+ J_+ - \omega_- J_-\,, \tag{5.24}$$

we can obtain a prediction for the on-shell action prior to embarking ourselves in the direct computation. Plugging the expressions we have obtained for the thermodynamic variables, we find that it is given by

$$I = \frac{\pi}{12\sqrt{3}}\left[\frac{\varphi^3}{\omega_1\omega_2} - \frac{\left(p_1\varphi - \omega_2\Phi^{I_1}\right)^3}{p_1^2\,\omega_2\left(p_1\omega_1 + (p_1-1)\,\omega_2\right)}\right]\,, \tag{5.25}$$

which is the expression advertised in (1.7). Moreover, we have verified that $I = \frac{1}{3}\left(\mathcal{Q}^{I_1}\Phi^{I_1} - \varphi\,Q\right)$, consistently with the Smarr formula of [35], whose supersymmetric version has been given in eq. (3.100).

In appendix C we show that the above expression for the on-shell action also follows from a direct computation, i.e. not relying on the validity of the quantum statistical relation. This computation requires some care since the gauge Chern-Simons term in the supergravity action needs to be evaluated patchwise.

One can check that regimes of the parameters exist such that the real part of the on-shell action is positive while the real part of the chemical potentials is negative, as it is required by naive convergence criteria of the partition function.

## 5.3 Relation to the two-center black hole saddle

As a sanity check, we can verify how the relations above reduce to those for the two-center black hole saddle of section 4.1 in the appropriate limit. In order to achieve this, we should demand that the on-shell action (or, equivalently, the entropy) does not depend on $\Phi^{I_1}$. Namely, treating $\Phi^{I_1}$ as a continuous variable, as in (5.23), we demand that $\mathcal{Q}^{I_1} = \frac{\partial I}{\partial\Phi^{I_1}} = 0$. This condition is solved by

$$\Phi^{I_1} = p_1\frac{\varphi}{\omega_2}\,. \tag{5.26}$$

In this regime, the on-shell action (5.25) simplifies to $I = \frac{\pi}{12\sqrt{3}}\frac{\varphi^3}{\omega_1\omega_2}$, which reproduces the result for the black hole saddle found in [40]. Notably, in the black hole saddle, the bubble potential $\Phi^{I_1}$ is not set to zero, whereas the flux $\mathcal{Q}^{I_1}$ is.

At the level of the parameters of the solutions, imposing (5.26) is equivalent to setting

$$k_1 = -\frac{h_1}{h_1 + h_N}\,k_S\,. \tag{5.27}$$

With this choice, the two centers at $z_1 = \delta_1$ and $z_2 = \frac{\delta}{2}$ coincide. This follows from eq. (5.8), which is now solved by taking

$$2\delta_1 = \delta\,. \tag{5.28}$$

Thus, the three-center solution collapses into a two-center configuration, which corresponds to the black hole saddle in the non-extremal case. In this limit, the harmonic functions simplify as:

$$
\begin{aligned}
H &= \frac{h_1 + h_N}{r_N} + \frac{h_S}{r_S}\,, & iK &= -\frac{k_S}{r_N} + \frac{k_S}{r_S}\,, \\
L &= 1 + \frac{k_S^2}{(h_1 + h_N)\,r_N} + \frac{k_S^2}{h_S\,r_S}\,, & iM &= \frac{k_S^3}{2}\left(\frac{1}{(h_1 + h_N)^2\,r_N} - \frac{1}{h_S^2\,r_S}\right).
\end{aligned}
\tag{5.29}
$$

From these expressions, we identify the black hole parameters as

$$
h_N^{\text{BH}} = h_1 + h_N\,, \qquad h_S^{\text{BH}} = h_S\,, \qquad k^{\text{BH}} = -k_S\,.
\tag{5.30}
$$

## 5.4 Connecting to Lorentzian solutions

In this section we investigate which well-behaved Lorentzian solutions are continuously connected to the Euclidean saddle constructed above. By well-behaved Lorentzian solution we mean a solution admitting a real metric after Wick-rotating back to Lorentzian time, and carrying real conserved charges, entropy and fluxes. From the expression of the entropy in terms of the charges (5.22), it is evident that it is not possible to have real charges and entropy simultaneously unless some constraint on the former is imposed, which is equivalent to fixing $\beta$. As in the two-center case, there are two physically-inequivalent constraints. The first corresponds to taking $\beta \to \infty$, and gives rise to extremal black holes with non-spherical horizon topology [18–20, 63, 64]. The second corresponds to taking $\beta \to 0$ and leads to horizonless topological solitons [21, 22] carrying zero entropy. This means that, as we vary $\beta$, the Euclidean saddle continuously interpolates between the two physical configurations (up to a specific analytic continuation of parameters), extending the findings of [40] to more general classes of solutions. Since the on-shell action (5.25) is independent of $\beta$, as expected for the saddle of a supersymmetric index, the same expression continues to hold after both limits.

### 5.4.1 Extremal limit

Let us assume for a moment that there exists a complexification of the solution yielding real and finite charges and entropy in the extremal limit. From the expression of the entropy in (5.22), one concludes that in this limit the charges must satisfy the constraint

$$
(1 - p_1)J_1 + (1 + p_1)J_2 + \Phi^{I_1}\left(Q - \frac{\pi}{12\sqrt{3}}\frac{(\Phi^{I_1})^2}{p_1^2}\right) = 0\,,
\tag{5.31}
$$

otherwise the entropy would not be real. Using the explicit expression for the charges (5.10), the potential (5.6) and the inverse temperature (5.9), one can indeed verify that the above constraint is equivalent to taking the extremal limit, in which the horizon rod contracts to a point,

$$
\delta \to 0 \qquad \Rightarrow \qquad \beta \to \infty\,.
\tag{5.32}
$$

As we explicitly show below, the resulting two-center solution describes an extremal and supersymmetric (BPS) black ring/lens. For $p_1 \neq 1$, we can solve (5.31) for $J_1$ and obtain the entropy from (5.22),

$$
\mathcal{S} = 4\sqrt{\pi}\sqrt{(1 - p_1)\left(\frac{Q}{\sqrt{3}} + \frac{\pi(\Phi^{I_1})^2}{12\,p_1(1 - p_1)}\right)^3 - \frac{\pi}{4}\left(J_2 - \frac{\pi(\Phi^{I_1})^3}{12\sqrt{3}\,p_1^2(1 - p_1)}\right)^2}\,.
\tag{5.33}
$$

Alternatively, for $p_1 \neq -1$ we can solve the constraint for $J_2$, obtaining

$$\mathcal{S} = 2\pi \sqrt{\frac{4(1-p_1)}{\pi} \left[ \frac{Q}{\sqrt{3}} + \frac{\pi(\Phi^{I_1})^2}{12 p_1 (1-p_1)} \right]^3 - \left[ \frac{1-p_1}{1+p_1} J_1 + \frac{\Phi^{I_1}}{1+p_1} \left( Q + \frac{\pi(\Phi^{I_1})^2}{6\sqrt{3} p_1 (1-p_1)} \right) \right]^2}. \tag{5.34}$$

Let us now discuss a complexification of the solution leading to real charges and entropy in the extremal limit. One way to achieve this is by imposing

$$h_S = h_N^*, \qquad k_S = -k_N^*. \tag{5.35}$$

Further using (5.4), one has that $h_1$ and $k_1$ are related to the real and imaginary parts of $h_N$ and $k_N$ by

$$\operatorname{Re} h_N = \frac{1 - h_1}{2}, \qquad k_1 = -2\mathrm{i} \operatorname{Im} k_N, \tag{5.36}$$

so that $\operatorname{Re} h_N$ is half-integer quantized. The extremal $\delta \to 0$ limit of the harmonic functions is

$$H = \frac{1-h_1}{r} + \frac{h_1}{r_1}, \qquad\qquad K = 2\operatorname{Im} k_N \left( \frac{1}{r} - \frac{1}{r_1} \right),$$
$$L = 1 + 2\operatorname{Re}\frac{k_N^2}{h_N}\frac{1}{r} - \frac{4(\operatorname{Im} k_N)^2}{h_1 r_1}, \qquad M = -\operatorname{Im}\frac{k_N^3}{h_N^2}\frac{1}{r} - \frac{4(\operatorname{Im} k_N)^3}{h_1^2 r_1}, \tag{5.37}$$

where $r$ is the distance to the origin of $\mathbb{R}^3$, where the horizon poles meet. The extremal solution is then parametrized by three real parameters $(\operatorname{Im} h_N, \operatorname{Re} k_N, \operatorname{Im} k_N)$ and an integer $(h_1)$. As we can see, the harmonic functions associated to this extremal solution are real, which implies that the Lorentzian continuation of the solution is also real. The charges of the extremal solution are simply obtained by imposing (5.35) in the expressions provided in (5.10),

$$J_+ = -6\pi \operatorname{Im} k_N \, \delta_1, \qquad J_- = 2\pi \left( \operatorname{Im}\frac{k_N^3}{h_N^2} + 4\frac{(\operatorname{Im} k_N)^3}{h_1^2} \right),$$
$$Q = 2\sqrt{3}\pi \left( \operatorname{Re}\frac{k_N^2}{h_N} - 2\frac{(\operatorname{Im} k_N)^2}{h_1} \right), \tag{5.38}$$

where

$$\delta_1 = -\frac{h_1}{3 \operatorname{Im} k_N} \operatorname{Im}\left[ h_N \left( \frac{k_N}{h_N} + \frac{2\mathrm{i} \operatorname{Im} k_N}{h_1} \right)^3 \right]. \tag{5.39}$$

This follows from solving (5.8) in the $\delta \to 0$ limit. In turn, the bubble potential $\Phi^{I_1}$ is given by

$$\Phi^{I_1} = -4\sqrt{3} \operatorname{Im} k_N. \tag{5.40}$$

This shows that charges $Q, J_\pm$, as well as $\Phi^{I_1}$, are real in the extremal limit. Additionally, one can verify that the constraint (5.31) is satisfied, further implying that the entropy is real. Instead, the chemical potentials associated to the extremal solution, which are given by

$$\omega_- = 4\pi\mathrm{i}\left( h_N^* - \frac{1}{2} \right), \qquad \varphi = 4\pi\sqrt{3}\, k_N^*, \tag{5.41}$$

remain complex. Consequently, the corresponding on-shell action is also complex. In contrast, to the extremal black hole with a spherical horizon analyzed in section 4.1 one associates real chemical potentials $\omega_-$, $\varphi$, and hence a real on-shell action, by using eq. (4.11) into (4.6), at least in the case where no shifts or orbifolds are considered. This is one of the distinctive features of these solutions.

**Topology of the Lorentzian solution.** Since in the extremal limit the horizon rod contracts to a point (the origin of $\mathbb{R}^3$), we are just left with one compact rod, $I_1$. Moreover, since the thermal circle is decompactified we can Wick-rotate back to Lorentzian time, $t = -i\tau$. The Lorentzian metric induced on the bolt associated to the surviving rod, $\mathcal{B}_{I_1}$, is given by

$$\mathrm{d}s^2_{\mathcal{B}_{I_1}} = -f^2 \big( \mathrm{d}t + \omega_\psi \, \mathrm{d}\psi_1 \big)^2 + f^{-1}H^{-1} \, \mathrm{d}\psi_1^2 + f^{-1}H \, \mathrm{d}z^2 \,, \tag{5.42}$$

where $\psi_1 = \psi + (1 - 2h_1)\phi$. The analysis of the topology of this bolt is analogous, to a large extent, to that of section 3.3. The main difference lies on the fact that the time coordinate $t$ is non-compact. In particular, the behavior of the metric functions near a bubbling center is the same as in the finite-$\beta$ solution, namely

$$f = \mathcal{O}(r_1^0), \qquad H = \frac{h_1}{r_1} + \mathcal{O}(r_1^0), \qquad \omega_\psi = \mathcal{O}(r_1). \tag{5.43}$$

Thus, we have again that the Killing vector $\partial_{\psi_1} = \partial_\psi$ contracts at the bubbling center, where space ends in a conical singularity if $|h_1| \neq 1$. On the contrary, the U(1) generated by this Killing vector has finite size at the horizon center, leading to the conclusion that $\mathcal{B}_{I_1}$ has $\mathbb{R} \times \mathrm{Disc}/\mathbb{Z}_{|h_1|}$ topology, with the $\mathbb{R}$ factor being parametrized by the Lorentzian time.

Let us now turn to the analysis of the near-horizon geometry. To this aim, it is convenient to recast the metric in the following form

$$\mathrm{d}s^2 = Y \big[ \mathrm{d}\psi + \chi - Y^{-1} f^2 \omega_\psi \, (\mathrm{d}t + \breve{\omega}) \big]^2 - Y^{-1} H^{-1} f \, (\mathrm{d}t + \breve{\omega})^2 + f^{-1}H \, \mathrm{d}s^2_{\mathbb{R}^3} \,, \tag{5.44}$$

where

$$Y = -f^2 \omega_\psi^2 + \frac{1}{fH} \,, \tag{5.45}$$

and expand it about the origin of $\mathbb{R}^3$, for $r \to 0$. The near-horizon behavior of the harmonic functions depends crucially on $h_1$, which forces us to consider separately the cases $h_1 \neq 1$ and $h_1 = 1$. These describe an extremal black lens and an extremal black ring, respectively.

**Near-horizon of extremal black lens ($h_1 \neq 1$).** The behavior of the metric functions near the horizon is

$$f = \hat{f} r + \mathcal{O}(r^2), \qquad Y = \frac{1 - h_1}{|h_N|^2 \hat{f}} + \mathcal{O}(r), \qquad \omega_\psi = \frac{\hat{\omega}_\psi}{r} + \mathcal{O}\big(r^0\big), \tag{5.46}$$

where $\hat{f}$ and $\hat{\omega}_\psi$ are constants. Using this information, one finds that the near-horizon expansion of (5.44) is given by

$$\begin{aligned}
\mathrm{d}s^2 = \frac{1 - h_1}{\hat{f}} & \left[ -\frac{|h_N|^2 \hat{f}^3}{(1 - h_1)^3} r^2 \, \mathrm{d}t^2 + \frac{\mathrm{d}r^2}{r^2} \right] \\
& + \frac{(1 - h_1)^3}{|h_N|^2 \hat{f}} \big( \mathrm{d}\hat{\psi} + \cos\theta \, \mathrm{d}\phi \big)^2 + \frac{1 - h_1}{\hat{f}} \big( \mathrm{d}\theta^2 + \sin^2\theta \, \mathrm{d}\phi^2 \big) + \dots \,,
\end{aligned} \tag{5.47}$$

where

$$\hat{\psi} = \frac{\psi - h_1 \phi}{1 - h_1} \,. \tag{5.48}$$

We recognize the AdS$_2$ factor characterizing the near-horizon geometry of extremal black holes in the first line of (5.47). In turn, the second line corresponds to the induced metric at the

horizon. Locally, this is the metric of a squashed three-sphere. However, the periodic identifications of the angular coordinates

$$\left(\phi, \hat{\psi}\right) \sim \left(\phi + 2\pi, \hat{\psi} + 2\pi\right) \sim \left(\phi, \hat{\psi} + \frac{4\pi}{1-h_1}\right), \tag{5.49}$$

tell us that, as in the non-extremal case, the horizon has lens-space topology

$$L(|1-h_1|, 1) \simeq S^3/\mathbb{Z}_{|1-h_1|}. \tag{5.50}$$

This corresponds to the extremal black lens of [19, 20]. A completely smooth geometry in the domain of outer communication is obtained for $h_1 = -1$, see below. By writing the explicit expression for $\hat{f}$,

$$\hat{f}^{-1} \equiv 2\mathrm{Re}\frac{\mathsf{k}_N^2}{h_N} + 4\frac{(\mathrm{Im}\mathsf{k}_N)^2}{1-h_1} = \frac{(\mathrm{Re}\mathsf{k}_N(1-h_1) + 2\mathrm{Im}\mathsf{k}_N\mathrm{Im}h_N)^2}{(1-h_1)|h_N|^2}, \tag{5.51}$$

we can verify that the combination $\frac{1-h_1}{\hat{f}}$ appearing in the near-horizon geometry (5.47) is positive, ensuring that the metric has the correct signature.

**Near-horizon of extremal black ring ($h_1 = 1$).** The behaviour of the metric functions for $r \to 0$ is

$$f^{-1} = \frac{4(\mathrm{Im}\mathsf{k}_N)^2 \delta_1}{r^2} + \mathcal{O}\left(\frac{1}{r}\right), \qquad Y = \frac{(\mathrm{Im}\mathsf{k}_N)^2}{|h_N|^2} + \mathcal{O}(r), \qquad \omega_\psi = \frac{8(\mathrm{Im}\mathsf{k}_N)^3 \delta_1^2}{r^3} + \mathcal{O}\left(\frac{1}{r^2}\right). \tag{5.52}$$

The near-horizon geometry reads

$$\begin{aligned} \mathrm{d}s^2 = {}& 4(\mathrm{Im}\mathsf{k}_N)^2 \left[-\frac{|h_N|^2}{16(\mathrm{Im}\mathsf{k}_N)^6} r^2 \mathrm{d}t^2 + \frac{\mathrm{d}r^2}{r^2}\right] \\ &+ \frac{(\mathrm{Im}\mathsf{k}_N)^2}{|h_N|^2}\mathrm{d}\tilde{\psi}^2 + 4(\mathrm{Im}\mathsf{k}_N)^2\left(\mathrm{d}\theta^2 + \sin^2\theta\,\mathrm{d}\phi^2\right) + \dots, \end{aligned} \tag{5.53}$$

where $\tilde{\psi} = \psi - \phi$. This corresponds to the near-horizon geometry of the extremal black ring found in [18], whose event horizon has $S^1 \times S^2$ topology.

### 5.4.2 The black lens

Among the solutions constructed in this section, two particular cases play a special role as they describe completely smooth supersymmetric non-extremal geometries. These correspond to solutions where $h_1 = \pm 1$, ensuring that the orbifold singularity at the bubbling center is absent. In the following we specialize the formulae for the on-shell action and chemical potential to these notable cases, focusing on the extremal $\beta \to \infty$ limits. Interestingly, the Lorentzian counterpart of these solutions is well known.

Let us start from the case $h_1 = -1$. From (5.47) we see that the extremal horizon has $L(2,1) \simeq S^3/\mathbb{Z}_2$ topology. This corresponds to the extremal black lens constructed in [19, 65]. This solution is described by the following harmonic functions [19]:

$$H = \frac{2}{r} - \frac{1}{r_1}, \qquad K = \hat{k}\left(\frac{1}{r} - \frac{1}{r_1}\right), \qquad L = 1 + \frac{\hat{\ell}}{r} + \frac{\hat{k}^2}{r_1}, \qquad M = \frac{\hat{m}}{r} - \frac{\hat{k}^3}{2r_1}. \tag{5.54}$$

These coefficients are related to the ones we used in section 5.4.1 by

$$\hat{k} = 2\mathrm{Im}\,\mathsf{k}_N, \qquad \hat{m} = \mathrm{Im}\left(\frac{\mathsf{k}_N^3}{h_N^2}\right), \qquad \hat{\ell} = 2\mathrm{Re}\left(\frac{\mathsf{k}_N^2}{h_N}\right). \tag{5.55}$$

The conserved charges carried by the solution are obtained by setting $h_1 = -1$ in (5.38) and (5.39). Moreover, the analysis of the extremal limit of section 5.4.1 implies the following constraint among the charges:

$$2J_1 + \Phi^{I_1}\left(Q - \frac{\pi}{12\sqrt{3}}\left(\Phi^{I_1}\right)^2\right) = 0\,. \tag{5.56}$$

The corresponding entropy is given by

$$\mathcal{S} = 4\sqrt{\pi}\sqrt{2\left(\frac{Q}{\sqrt{3}} - \frac{\pi}{24}(\Phi^{I_1})^2\right)^3 - \frac{\pi}{4}\left(J_2 - \frac{\pi}{24\sqrt{3}}(\Phi^{I_1})^3\right)^2}\,, \tag{5.57}$$

which is consistent with the results of [19, 65], up to conventions.

Similarly, the on-shell action for the supersymmetric extremal black lens is obtained by setting $p_1 = -1$ into (5.25),

$$I = \frac{\pi}{12\sqrt{3}}\left[\frac{\varphi^3}{\omega_1\omega_2} - \frac{\left(\varphi + \omega_2\Phi^{I_1}\right)^3}{\omega_2(2\omega_2 + \omega_1)}\right]\,, \tag{5.58}$$

where the chemical potentials turn out to be given by the following complex combinations

$$\omega_1 = 2\pi\left(\mathrm{Im}h_N + \mathrm{i}\right)\,, \qquad \omega_2 = -2\pi\,\mathrm{Im}h_N\,, \qquad \varphi = 4\sqrt{3}\pi\left(\mathrm{Re}k_N + \mathrm{i}\,\mathrm{Im}k_N\right)\,, \tag{5.59}$$

while $\Phi^{I_1}$ reads

$$\Phi^{I_1} = -4\sqrt{3}\,\mathrm{Im}k_N\,. \tag{5.60}$$

### 5.4.3 The black ring

Let us now set $h_1 = 1$. From (5.53), it follows that the extremal horizon has $S^1 \times S^2$ topology. This corresponds to the supersymmetric black ring studied in [18, 63, 64]. Setting $h_1 = 1$ in (5.37) indeed gives the harmonic functions that characterize such solution (see for instance [29]),

$$H = \frac{1}{r_1}\,, \qquad K = \hat{k}\left(\frac{1}{r} - \frac{1}{r_1}\right)\,, \qquad L = 1 + \frac{\hat{\ell}}{r} - \frac{\hat{k}^2}{r_1}\,, \qquad M = \frac{\hat{m}}{r} - \frac{\hat{k}^3}{2r_1}\,, \tag{5.61}$$

where the map between $(\hat{k}, \hat{l}, \hat{m})$ and the coefficients we used in section 5.4.1 is the same as in (5.55). The charges of the black ring must satisfy the constraint

$$2J_2 + \Phi^{I_1}\left(Q - \frac{\pi}{12\sqrt{3}}\left(\Phi^{I_1}\right)^2\right) = 0\,, \tag{5.62}$$

while the entropy takes the simple form

$$\mathcal{S} = \frac{\pi}{\sqrt{3}}|\Phi^{I_1}|\sqrt{Q^2 + \frac{\pi^2}{144}\left(\Phi^{I_1}\right)^4 - \frac{\pi}{\sqrt{3}}\Phi^{I_1}J_1}\,. \tag{5.63}$$

These expressions are in agreement with the results for the supersymmetric black ring obtained in [63].

We find that the angular velocity $\omega_1$ is real, while the other potentials are complex,

$$\omega_1 = 2\pi\,\mathrm{Im}h_N\,, \qquad \omega_2 = 2\pi\left(\mathrm{i} - \mathrm{Im}h_N\right)\,, \qquad \varphi = 4\sqrt{3}\pi\left(\mathrm{Re}k_N + \mathrm{i}\,\mathrm{Im}k_N\right)\,. \tag{5.64}$$

Again, the bubble potential is[19]

$$\Phi^{I_1} = -4\sqrt{3}\,\mathrm{Im}k_N\,. \tag{5.65}$$

Finally, the on-shell action for the supersymmetric black ring turns out to be

$$I = \frac{\pi}{12\sqrt{3}}\frac{\Phi^{I_1}}{\omega_1}\left(3\varphi^2 - 3\varphi\,\Phi^{I_1}\,\omega_2 + \left(\Phi^{I_1}\right)^2\omega_2^2\right)\,. \tag{5.66}$$

Therefore, starting from the non-extremal saddles and taking an extremal limit, on the one hand we have correctly reproduced the known extremal entropy of the supersymmetric black lens and black ring, and on the other hand we have assigned for the first time non-trivial chemical potentials and on-shell action to these solutions.

### 5.4.4  Topological solitons

Finally, we discuss a different limit leading to a horizonless topological soliton. Besides the reality conditions discussed above, another choice that ensures the charges $Q$, $J_\pm$ and the potential $\Phi^{I_1}$ remain real is the obvious one:

$$k_N = ik_N\,, \quad k_S = ik_S\,, \quad k_1 = ik_1\,, \qquad k_1 + k_N + k_S = 0\,, \tag{5.67}$$

with $k_{N,S,1}$, $h_{N,S,1} \in \mathbb{R}$. However, under this analytic continuation, the entropy given by (5.22) becomes purely imaginary. This follows from the fact that the argument of the square root in (5.22) turns negative:[20]

$$(1-h_1)\left(\frac{Q}{\sqrt{3}} + \frac{\pi(\Phi^{I_1})^2}{12h_1(1-h_1)}\right)^3 - \frac{\pi}{4}\left((h_1-1)J_- - \frac{\Phi^{I_1}}{2}Q - \frac{\pi(1+h_1)(\Phi^{I_1})^3}{24\sqrt{3}\,h_1^2(1-h_1)}\right)^2$$
$$= -\frac{\pi^3}{4h_N^4\,h_S^4}(h_N\,k_1 + k_N(1-h_1))^6 < 0\,. \tag{5.68}$$

Thus, the only possibility to get a real quantity is that $\mathcal{S}$ vanishes. This is equivalent to the following constraint among the charges[21]

$$\frac{4(1-h_1)}{3\sqrt{3}\pi}Q^2\left(Q + \frac{\sqrt{3}\pi}{4h_1(1-h_1)}(\Phi^{I_1})^2\right) + (1-h_1)J_1\left(J_2 - \frac{\pi}{12\sqrt{3}\,h_1^2(1-h_1)}(\Phi^{I_1})^3\right)$$
$$+ h_1 J_2\left(J_2 + h_1^{-1}\Phi^{I_1}Q\right) = 0\,, \tag{5.69}$$

leading to a horizonless solution [40]. From (5.21), one further deduces that this condition is equivalent to $\beta \to 0$, which in terms of the parameters reads

$$\frac{3}{2}\frac{k_N}{h_N} - \frac{h_S}{2\delta}\left(\frac{k_N}{h_N} - \frac{k_S}{h_S}\right)^3 - \frac{h_1}{2\delta_1 - \delta}\left(\frac{k_N}{h_N} - \frac{k_1}{h_1}\right)^3 = 0\,. \tag{5.70}$$

This constraint, together with (5.8), which we reproduce here for completeness,

$$\frac{3}{2}\frac{k_1}{h_1} - \frac{h_S}{2\delta_1 + \delta}\left(\frac{k_1}{h_1} - \frac{k_S}{h_S}\right)^3 - \frac{h_N}{2\delta_1 - \delta}\left(\frac{k_1}{h_1} - \frac{k_N}{h_N}\right)^3 = 0\,, \tag{5.71}$$

---

[19]This corresponds, up to normalization, to the *dipole charge* that characterizes the thermodynamics of the black ring [34,64]. The relation between these two quantities has been explained in [35].

[20]An alternative way to see this is to note that the inverse temperature $\beta$ in (5.9) also becomes imaginary under this continuation of the parameters.

[21]This constraint can be understood as imposing $P_0 = 0$ in (C.23).

constitutes the system of two *bubble equations* arising in three-center microstate geometries of [21, 22].

From a global perspective, in the $\beta \to 0$ limit what used to be the thermal isometry in (2.35) does not advance the Euclidean time anymore. As a result, the global identifications (2.34) become those relevant for a bubbling geometry,

$$(\tau, \phi, \psi) \sim \left(\tau, \phi + 2\pi, \psi + 2\pi(-1 + 2(h_1 + h_N))\right) \sim (\tau, \phi + 2\pi, \psi - 2\pi)$$
$$\sim (\tau, \phi, \psi + 4\pi)\,, \tag{5.72}$$

where consistency of the angular identifications requires $h_{N,S,1} \in \mathbb{Z}$ [21,22]. Once these conditions are imposed, we can safely Wick-rotate back to Lorentzian time, obtaining a well-behaved horizonless solution. The resulting spacetime is smooth up to certain discrete orbifold singularities, which we discuss below.

For this solitonic solution, the harmonic functions remain real, ensuring that the metric after Wick-rotating back to Lorentzian time is also real. However, the assigned chemical potentials become imaginary:

$$\omega_1 = 2\pi i h_S\,, \qquad \omega_2 = 2\pi i (h_N + h_1)\,, \qquad \varphi = -4\sqrt{3}\pi i k_S\,. \tag{5.73}$$

As a result, the on-shell action (5.25) is also imaginary. In fact, the angular velocities must be multiples of $2\pi i$, i.e. $\omega_{1,2} \in 2\pi i \mathbb{Z}$. Following the general argument developed in section 3.6, we can argue that also $\varphi$ becomes a multiple of $2\pi i$. This is the same outcome we noticed in the two-center solution discussed in section 4.1, and appears to be a general property of horizonless topological solitons. So these solutions appear to be relevant just in a particular limit of the gravitational index where the fugacities trivialize. It would be interesting to investigate the trace over microstates given by (1.6) in this limit.

The horizonless solution obtained sending $\beta \to 0$ has two compact rods, $I_1$ and $I_N$. The associated rod vectors are

$$\xi_{I_1} = \partial_\phi - (1 - 2h_1)\partial_\psi\,, \qquad \xi_{I_N} = \partial_\phi - (h_S - h_N - h_1)\partial_\psi\,. \tag{5.74}$$

Since both are bubbling rods, we have that the Killing vector $\partial_\psi$ contracts at all endpoints, implying that their fixed loci have $\mathbb{R} \times \Sigma_{[h_a, h_{a+1}]}$ topology. The analysis of the topology is analogous to that of section 3.3, with the exception that time is no more periodic. For an odd number of centers we can get rid of orbifold singularities by setting $|h_a| = 1$ for all $a$'s [24]. In the present three-center case, this leads to three distinct fully regular cases:

$$h_1 = h_N = -h_S = 1\,, \qquad h_1 = -h_N = h_S = 1\,, \qquad h_1 = -h_N = -h_S = -1\,. \tag{5.75}$$

# 6 Conclusions

In this paper, we considered pure five-dimensional supergravity with boundary conditions such that the gravitational path integral computes a grand-canonical supersymmetric index. We investigated candidate saddles of this gravitational index, namely supersymmetric yet non-extremal complexified solutions to the supergravity equations satisfying the boundary conditions and having finite action. Assuming $U(1)^3$ symmetry, we classified the solutions using the rod structure formalism, and studied their topology by analyzing the fixed loci of their isometries. We found a rich array of possibilities, which correspond in general to multi-horizon solutions with many bubbles. The fixed loci can have $S^3$, $S^2 \times S^1$ and lens space topologies, as well as versions of these geometries with conical singularities. We illustrated the relation of the angular velocities and electrostatic potential of each horizon with the chemical potentials

appearing in the index, and also discussed the role of the specific gauge potentials associated with the bubbles. Since the latter potentials are not associated with an asymptotic symmetry, they are not expected to represent additional variables of the index. Rather, it appears we should sum over their allowed values (when the gauge group is U(1), we have seen these are indeed integers).

For the notable case of the three-center solution, we explicitly computed the on-shell action providing the saddle-point contribution to the index. We also illustrated how these solutions are related to known Lorentzian solutions – BPS black rings or lenses and horizonless bubbling solutions – by taking suitable limits. We observed that the chemical potentials and on-shell action remain finite in this limit, and verified that the Legendre transform of the on-shell action reproduces the correct Bekenstein-Hawking entropy. While the action of the black ring and black lens can be taken with a positive real part, and the chemical potentials with a negative real part, ensuring naive convergence conditions, both the action and chemical potentials associated with the horizonless solutions are purely imaginary, making it unclear whether the latter should be regarded as true saddles of the grand-canonical index. It would be very interesting to compare these findings with a microscopic evaluation of the index in string theory.

Relatedly, an important issue that needs clarification is what the allowed complexifications of the parameters controlling the solutions in this paper are. More generally, it would be important to clarify what are the criteria for a candidate saddle of the gravitational index to be counted as a true saddle. After filtering the candidate saddles based on the allowability conditions, it should be possible to compare the on-shell actions of the competing saddles with different topologies for a given value of the chemical potentials. This would allow to discuss the various phases of the gravitational index and potentially find a recipe for resumming the saddle-point contributions. Different allowability criteria have been proposed recently, based on positive-definiteness of $p$-form kinetic terms [66,67] – assessed in a supersymmetric context in [68,69] – or wrapped brane stability [62]. However, no conclusive statement has been made so far. The ultimate criterion to decide whether a candidate saddle does or does not contribute to the path integral would be to determine whether it is crossed by the integration contour. This is a hard question which would require applying Picard-Lefschetz theory in the context of the gravitational path integral; one may try to attack it by working in a simplified setup at first. Again, a comparison with a microscopic computation, of the type that is sometimes possible when a conformal field theory holographic dual is available, would shed light on this issue.

Another natural question is whether one can exploit the technique of equivariant localization in order to efficiently compute the on-shell action of our five-dimensional solutions, as done for various classes of even-dimensional supersymmetric solutions starting with [70]. Assuming that a solution exists, this may allow to determine the on-shell action just based on the rod structure and the boundary conditions, without knowing its full explicit form. For the simplest case with just one compact rod, this problem was addressed in [40] by localizing with respect to a Killing vector with a fixed circle. For a general rod structure, however, the approach of [40] needs to be revisited. This is because in the situation where different U(1) isometries degenerate and, relatedly, the gauge field $A$ is not globally defined, it is currently unknown how to construct the globally well-defined equivariantly-closed form needed to implement the BVAB localization argument. Also the fact that the finite-temperature supersymmetric solutions presented here do not appear to allow for a real positive-definite metric represents a further complication, since the standard form of the localization theorem assumes a Riemannian manifold.

One may also consider dimensionally reducing along one of the isometries so as to apply equivariant localization in four dimensions, which is better understood. Reducing along the orbits of $\partial_\psi$ (which commutes with the action of the preserved supercharges) would allow to make contact with the study performed in four-dimensional gauged supergravity in [71,72],

where gravitational instanton solutions are studied that have a structure similar to the bubbling solutions discussed here. Reducing instead along the orbits of the supersymmetric Killing vector $\partial_\tau$ would perhaps make it possible to apply the method developed in [73] in the context of five-dimensional gauged supergravity, which however would also need to be extended to locally defined gauge fields. We leave the investigation of how to efficiently combine the rod structure information with equivariant localization for future work.

While here we provided the general framework for solutions with multiple horizon components, it would be interesting to study in detail explicit examples of these configurations, as well as to consider solutions where the centers are not necessarily aligned along a symmetry axis. Very recently, saddles of the four-dimensional gravitational index comprising these features have been studied in [45], and it would be interesting to make the connection between the four-dimensional and the five-dimensional cases more explicit.

Although in this paper we worked in minimal supergravity and thus with a single U(1) gauge field, our results straightforwardly generalize to the case where vector multiplets are included. Multi-charge saddles of the gravitational index with a single compact rod were presented in [40], see also [38] for the three-charge case. Recently, the case corresponding to the D1-D5-$p$ brane system with one of the three charges turned off was studied in [74], where a finite-temperature version of the small black ring was given. We note that the solutions of [74] necessarily have running scalars, so there is no limit in which they reduce to the solutions of minimal supergravity presented here, which in the D1-D5-$p$ system correspond to solutions with the three charges set equal. It will be interesting to investigate rod structures in the generic three-charge case, thus exploring the index associated with the full D1-D5-$p$ brane system.

## Acknowledgments

We would like to thank Nikolay Bobev, Jan Boruch, Edoardo Colombo, Roberto Emparan, Jerome Gauntlett, Stefano Giusto and Sameer Murthy for useful discussions.

**Funding information** DC is supported in part by the MUR-PRIN contract 2022YZ5BA2 - Effective quantum gravity. AR is supported by a postdoctoral fellowship associated to the MIUR-PRIN contract 2020KR4KN2 - String Theory as a bridge between Gauge Theories and Quantum Gravity. AR further thanks the High Energy Theory group at INFN and University of Padova for hospitality and financial support. ET thanks the Theoretical Physics group at King's College London for hospitality during the completion of part of this work. ET further thanks the COST Action CA22113 for financial support during the stay at King's College London.

## A   Expression for $\breve{\omega}$

The purpose of this appendix is to derive eq. (2.10). Namely, we want to solve

$$\star_3 \mathrm{d}\breve{\omega} = H\mathrm{d}M - M\mathrm{d}H + \frac{3}{2}\left(K\mathrm{d}L - L\mathrm{d}K\right), \tag{A.1}$$

for the general choice of harmonic functions made in (2.6), under the assumption that all centers are placed on the $z$-axis. We start noticing that

$$\mathrm{i}\breve{\omega} = \sum_{a=1}^{s}\left[h_0 m_a - m_0 h_a + \frac{3}{2}\left(k_0 \ell_a - \ell_0 k_a\right)\right]\cos\theta_a \mathrm{d}\phi + \mathrm{i}\breve{\omega}_*, \tag{A.2}$$

where $\breve{\omega}_*$ satisfies the same equation as $\breve{\omega}$ but ignoring the constant term in all the harmonic functions. This implies that

$$\star_3 \mathrm{d}\,i\breve{\omega}_* = \sum_a \sum_b C_{ab} \frac{\mathrm{d}r_b^{-1}}{r_a} = \sum_a \sum_{b>a} C_{ab} \left( \frac{\mathrm{d}r_b^{-1}}{r_a} - \frac{\mathrm{d}r_a^{-1}}{r_b} \right), \tag{A.3}$$

where

$$C_{ab} = h_a \mathsf{m}_b - h_b \mathsf{m}_a + \frac{3}{2}(\mathsf{k}_a \ell_b - \mathsf{k}_b \ell_a). \tag{A.4}$$

We can solve (A.3) for each pair of centers labelled by $a, b$ separately, and then sum up all the contributions. The result is

$$i\breve{\omega}_* = \left[ c_\omega - \sum_a \sum_{b>a} \frac{C_{ab}}{\delta_{ab}} \frac{r_a + \delta_{ab}\cos\theta_a}{r_b} \right] \mathrm{d}\phi, \tag{A.5}$$

where $c_\omega$ is an arbitrary integration constant and $\delta_{ab} \equiv z_a - z_b$ (emphasis is put on the fact that there is no absolute value). Now we note that

$$\begin{aligned}
\frac{r_a + \delta_{ab}\cos\theta_a}{r_b} &= (1 + \cos\theta_a)\frac{r_a + \delta_{ab}}{r_b} - \frac{\delta_{ab} + r_a\cos\theta_a}{r_b} \\
&= (1 + \cos\theta_a)\left( -1 + \frac{r_a + \delta_{ab}}{r_b} \right) + \cos\theta_a - \cos\theta_b + 1.
\end{aligned} \tag{A.6}$$

Hence, fixing $c_\omega = \sum_a \sum_{b>a} \frac{C_{ab}}{\delta_{ab}}$, we have

$$i\breve{\omega}_* = \sum_a \sum_{b>a} \frac{C_{ab}}{\delta_{ab}} (\cos\theta_b - \cos\theta_a)\,\mathrm{d}\phi + i\breve{\omega}_{\mathrm{reg}}, \tag{A.7}$$

where

$$i\breve{\omega}_{\mathrm{reg}} = \sum_a \sum_{b>a} \frac{C_{ab}}{\delta_{ab}} (1 + \cos\theta_a)\left( 1 - \frac{r_a + \delta_{ab}}{r_b} \right) \mathrm{d}\phi. \tag{A.8}$$

One can check that $\breve{\omega}_{\mathrm{reg}}$ is regular in the entire $z$-axis, being this statement independent of the sign of $\delta_{ab}$. Finally, we want to re-write the first term in (A.7) in the form $\sum_a \mathrm{coeff}_a \cos\theta_a \mathrm{d}\phi$, so that it combines with the first term in (A.2). To this aim, we assume that the centers are ordered in a way such that $\delta_{ab} > 0$ if $b > a$. Then,

$$\begin{aligned}
\sum_a \sum_{b>a} \frac{C_{ab}}{\delta_{ab}} (\cos\theta_b - \cos\theta_a) &= \sum_a \sum_{b<a} \frac{C_{ba}}{\delta_{ba}} \cos\theta_a - \sum_a \sum_{b>a} \frac{C_{ab}}{\delta_{ab}} \cos\theta_a \\
&= -\sum_a \sum_{b\neq a} \frac{C_{ab}}{|\delta_{ab}|} \cos\theta_a,
\end{aligned} \tag{A.9}$$

where in the first equality we have exchanged the labels $a, b$ in the first term, while in the second equality we have used that $C_{ba} = -C_{ab}$ and that $\delta_{ab} = |\delta_{ab}|$ if $b > a$ and $\delta_{ba} = -\delta_{ab} = |\delta_{ab}|$ if $b < a$. All in all, the general expression for $\breve{\omega}$ is

$$i\breve{\omega} = \sum_a iw_a \cos\theta_a \mathrm{d}\phi + \sum_a \sum_{b>a} \frac{C_{ab}}{\delta_{ab}} (1 + \cos\theta_a)\left( 1 - \frac{r_a + \delta_{ab}}{r_b} \right) \mathrm{d}\phi, \tag{A.10}$$

where

$$iw_a = h_0 \mathsf{m}_a - \mathsf{m}_0 h_a + \frac{3}{2}(\mathsf{k}_0 \ell_a - \ell_0 \mathsf{k}_a) - \sum_{b\neq a} \frac{C_{ab}}{|\delta_{ab}|}. \tag{A.11}$$

Note that because of the assumption $\delta_{ab} > 0$ if $a < b$, only the absolute value of $\delta_{ab}$ appears in (2.10).

# B Lens spaces

We recall here the definition of a Lens space and highlight some properties that are needed in the main text.

Given a unit $S^3$ parameterized by $(Z_1, Z_2) \in \mathbb{C}^2$, with $|Z_1|^2 + |Z_2|^2 = 1$, and given coprime integers $(\mathsf{p}, \mathsf{q})$, the Lens space $L(\mathsf{p}, \mathsf{q}) \simeq S^3/\mathbb{Z}_\mathsf{p}$ is defined by the orbifold identification

$$(Z_1, Z_2) \sim \left( e^{\frac{2\pi i}{\mathsf{p}}} Z_1, \, e^{\frac{-2\pi i \mathsf{q}}{\mathsf{p}}} Z_2 \right). \tag{B.1}$$

Since $\mathsf{p}$ and $\mathsf{q}$ are coprime, the orbifold is freely acting and the resulting space is a smooth manifold.

We can take $Z_1 = \sin\vartheta \, e^{i\phi_1}$, $Z_2 = \cos\vartheta \, e^{i\phi_2}$, with $\vartheta \in [0, \frac{\pi}{2}]$ and $\phi_1, \phi_2$ satisfying the identifications

$$(\phi_1, \phi_2) \sim (\phi_1 + 2\pi, \phi_2) \sim (\phi_1, \phi_2 + 2\pi). \tag{B.2}$$

These are the angular coordinates of footnote 5. The orbits of $\partial_{\phi_1}$ and $\partial_{\phi_2}$ are $2\pi$-periodic and smoothly collapse to zero size at either endpoint of the $\vartheta$ interval. The orbifold introduces the new identification

$$(\phi_1, \phi_2) \sim \left( \phi_1 + \frac{2\pi}{\mathsf{p}}, \, \phi_2 - \frac{2\pi\mathsf{q}}{\mathsf{p}} \right), \tag{B.3}$$

hence the nowhere vanishing vector

$$\lambda = \frac{1}{\mathsf{p}} \partial_{\phi_1} - \frac{\mathsf{q}}{\mathsf{p}} \partial_{\phi_2}, \tag{B.4}$$

also has $2\pi$-periodic orbits. It describes an $S^1$ fibration over a two-dimensional base which in general is a spindle. The same expression can be written as

$$\partial_{\phi_1} = \mathsf{q} \, \partial_{\phi_2} + \mathsf{p} \, \lambda, \tag{B.5}$$

showing that going once around the circle that smoothly collapses at $\vartheta = 0$ is the same as going $\mathsf{q}$ times around the circle that smoothly collapses at $\vartheta = \frac{\pi}{2}$ and $\mathsf{p}$ times around the non-vanishing fibre.

In the main text, we deal with orbifold identifications of the type

$$(\phi_1, \phi_2) \sim \left( \phi_1 + \frac{2\pi\mathsf{q}_1}{\mathsf{p}}, \, \phi_2 - \frac{2\pi\mathsf{q}_2}{\mathsf{p}} \right), \tag{B.6}$$

where both $\mathsf{q}_1, \mathsf{q}_2$ are coprime to $\mathsf{p}$. We thus have a $2\pi$-periodic nowhere-vanishing vector $\tilde{\lambda}$ such that

$$\mathsf{q}_1 \, \partial_{\phi_1} = \mathsf{q}_2 \, \partial_{\phi_2} + \mathsf{p} \, \tilde{\lambda}. \tag{B.7}$$

It is not hard to see that this orbifold in fact defines the lens space $L(\mathsf{p}, \mathsf{a} \, \mathsf{q}_2)$, where $\mathsf{a}$ is an integer determined by the equation

$$\mathsf{a} \, \mathsf{q}_1 + \mathsf{b} \, \mathsf{p} = 1, \qquad \mathsf{a}, \mathsf{b} \in \mathbb{Z}, \tag{B.8}$$

which can always be solved for coprime $\mathsf{q}_1, \mathsf{p}$ by the Bézout lemma. The equivalence is seen as follows: applying $\mathsf{a}$ times the identification (B.6) and using $\frac{\mathsf{a}\mathsf{q}_1}{\mathsf{p}} = \frac{1}{\mathsf{p}} - \mathsf{b}$, we land on (B.3) with $\mathsf{q} = \mathsf{a} \, \mathsf{q}_2$; conversely, applying $\mathsf{q}_1$ times the identification (B.3) with $\mathsf{q} = \mathsf{a} \, \mathsf{q}_2$, and using $\mathsf{q}_1 \frac{\mathsf{a}\mathsf{q}_2}{\mathsf{p}} = \frac{\mathsf{q}_2}{\mathsf{p}} - \mathsf{b} \, \mathsf{q}_2$, we arrive at (B.6).

Starting from (B.7), it is straightforward to see that the vector $\lambda$ specifying the lens space circle fibration is given by

$$\lambda = \frac{1}{\mathsf{q}_1} \tilde{\lambda} + \frac{\mathsf{b} \, \mathsf{q}_2}{\mathsf{q}_1} \partial_{\phi_2}. \tag{B.9}$$

# C  The on-shell action and its Legendre transform

## C.1  Evaluation of the on-shell action

Here we compute the on-shell action $I$ of the three-center solution studied in section 5. This is given by the sum

$$
\begin{aligned}
I &= I_{\text{bulk}} + I_{\text{GHY}} \\
&= -\frac{1}{16\pi} \int_{\mathcal{M}} \left( R \star_5 1 - \frac{1}{2} F \wedge \star_5 F + \frac{\mathrm{i}}{3\sqrt{3}} F \wedge F \wedge A \right) \\
&\quad - \frac{1}{8\pi} \int_{\partial\mathcal{M}} \mathrm{d}^4 x \left( \sqrt{h}\,\mathcal{K} - \sqrt{h_{\text{bkg}}}\,\mathcal{K}_{\text{bkg}} \right),
\end{aligned}
\tag{C.1}
$$

where $\partial\mathcal{M}$ is the cutoff boundary of the regulated spacetime at large but finite distance, $h$ denotes the determinant of the induced metric at such boundary and $\mathcal{K}$ its extrinsic curvature. To remove divergences, we perform a background subtraction using a reference flat background with the same asymptotics, characterized by $h_{\text{bkg}}$ and $\mathcal{K}_{\text{bkg}}$.

A subtle issue arises in the computation of the on-shell action due to the presence of the Chern-Simons term. Since $A$ is not a globally defined one-form, the integral $\int F \wedge F \wedge A$ is ill-defined and must be treated with care. A similar issue was encountered in early studies of black rings [75], where it was noted that the Page charge, used to compute the electric charge and angular momentum, is not well-defined for the same reason. The way out of this problem involves introducing an appropriate number of patches and summing their contributions while including a compensating term along the patch overlaps.

To address this issue, we introduce the relevant patches. We define the open set

$$
\mathcal{U}_{I_1} \equiv \left\{ (\tau, \psi, \rho, \phi, z) \,:\, \frac{\rho^2}{\varepsilon^2} + \frac{\left(z - \frac{2\delta_1 + \delta}{4}\right)^2}{\left(\varepsilon + \frac{2\delta_1 - \delta}{4}\right)^2} < 1, \quad \varepsilon > 0 \right\},
\tag{C.2}
$$

while $\mathcal{U}_{\mathcal{H}}$ is defined as the complement of $\mathcal{U}_{I_1}$. The patch $\mathcal{U}_{\mathcal{H}}$ coincides with the one defined in (3.85); this covers the horizon rod and extends to the asymptotic region. Instead, $\mathcal{U}_{I_1}$ does not extend to the asymptotic region. The parameter $\varepsilon$, which controls the size of the region $\mathcal{U}_{I_1}$, is a positive small quantity, whose exact value should not affect the final result. Indeed, many expressions simplify in the limit $\varepsilon \to 0$, as we discuss below.

The boundary of the closure of $\mathcal{U}_{I_1}$, denoted by $\partial\mathcal{U}_{I_1}$, is the surface parametrized by

$$
\rho = \varepsilon \sin\theta, \qquad z = \frac{2\delta_1 + \delta}{4} + \left( \varepsilon + \frac{2\delta_1 - \delta}{4} \right) \cos\theta.
\tag{C.3}
$$

Taking the limit $\varepsilon \to 0$, this region collapses to $\rho = 0$, with $z = \frac{2\delta_1(1+\cos\theta) + \delta(1-\cos\theta)}{4}$. This shows that in this limit, the set $\mathcal{U}_{I_1}$ shrinks onto the bolt $\mathcal{B}_{I_1}$, leading to the simplifications we will exploit below.

The gauge field can be constructed patch-by-patch. The gauge fields in the two patches are related by the transformation

$$
A_1 = A_{\mathcal{H}} - \mathrm{i}\sqrt{3}\,\mathrm{d}(\alpha_1 - \alpha_{\mathcal{H}}),
\tag{C.4}
$$

where

$$
\alpha_1 = -\frac{\mathsf{k}_1}{h_1} \left( \phi + \psi - \frac{4\pi h_N}{\beta} \tau \right) - \frac{4\pi \mathsf{k}_N}{\beta} \tau, \qquad \alpha_{\mathcal{H}} = \frac{4\pi \mathsf{k}_S}{\beta} \tau,
\tag{C.5}
$$

as we showed in (3.87) and (5.15). Having introduced the two patches, we now extend the approach of [75] to properly define the integral of the Chern-Simons form in our setup. In order to define the integral properly, we start from a gauge invariant quantity. The natural quantity to consider is the six-form $F \wedge F \wedge F = \mathrm{d}(F \wedge F \wedge A)$. To define an appropriate integration manifold, we assume the existence of a smooth six-dimensional extension of the spacetime, denoted by $\mathcal{M}_6$, such that its boundary coincides with the five-dimensional spacetime under consideration. This ensures that $\mathcal{M}$ is cobordant to a point. Additionally, we extend the two patches $\mathcal{U}_{\mathcal{H}}$ and $\mathcal{U}_{I_1}$ smoothly into $\mathcal{M}_6$, denoting these extensions by $\widehat{\mathcal{U}}_{\mathcal{H}}$ and $\widehat{\mathcal{U}}_{I_1}$. We can then write a well-defined integral as

$$\int_{\mathcal{M}_6} F \wedge F \wedge F = \int_{\widehat{\mathcal{U}}_{\mathcal{H}} \cap \mathcal{M}_6} F \wedge F \wedge F + \int_{\widehat{\mathcal{U}}_{I_1} \cap \mathcal{M}_6} F \wedge F \wedge F. \tag{C.6}$$

Integrating by parts gives

$$\int_{\mathcal{M}_6} F \wedge F \wedge F = \int_{\mathcal{U}_{\mathcal{H}}} F \wedge F \wedge A_{\mathcal{H}} + \int_{\mathcal{U}_{I_1}} F \wedge F \wedge A_1 + \int_{\partial \widehat{\mathcal{U}}_{I_1} \cap \mathcal{M}_6} (A_{\mathcal{H}} - A_1) \wedge F \wedge F. \tag{C.7}$$

Using the gauge map (C.4), this expression simplifies to

$$\int_{\mathcal{M}_6} F \wedge F \wedge F = \int_{\mathcal{U}_{\mathcal{H}}} F \wedge F \wedge A_{\mathcal{H}} + \int_{\mathcal{U}_{I_1}} F \wedge F \wedge A_1 - \mathrm{i}\sqrt{3} \int_{\partial \mathcal{U}_{I_1}} \mathrm{d}(\alpha_1 - \alpha_{\mathcal{H}}) \wedge A_1 \wedge F. \tag{C.8}$$

Following [75], we take this sum of three terms as the formal definition of the Chern-Simons integral in five dimensions:

$$\int_{\mathcal{M}} F \wedge F \wedge A \equiv \int_{\mathcal{M}_6} F \wedge F \wedge F. \tag{C.9}$$

We now turn to the explicit evaluation of the on-shell action (C.1). Using the trace of Einstein equations, the bulk term simplifies to

$$I_{\text{bulk}} = \frac{1}{48\pi} \int_{\mathcal{U}_{\mathcal{H}}} F \wedge G_{\mathcal{H}} + \frac{1}{48\pi} \int_{\mathcal{U}_{I_1}} F \wedge G_1 - \frac{1}{48\pi} \int_{\partial \mathcal{U}_{I_1}} \mathrm{d}(\alpha_1 - \alpha_{\mathcal{H}}) \wedge A_1 \wedge F, \tag{C.10}$$

where $G \equiv \star_5 F - \frac{\mathrm{i}}{\sqrt{3}} A \wedge F$. The size of the patch $\mathcal{U}_{I_1}$ is controlled by the parameter $\varepsilon$ introduced in (C.2). In the limit $\varepsilon \to 0$, the volume of $\mathcal{U}_{I_1}$ collapses to zero. Consequently, any smooth five-form integrated over $\mathcal{U}_{I_1}$ must vanish. In particular,

$$\lim_{\varepsilon \to 0} \int_{\mathcal{U}_{I_1}} F \wedge G_1 = 0. \tag{C.11}$$

This follows from the fact that any smooth form in $\mathcal{U}_{I_1}$, such as $F \wedge G_1$, must have a component along the direction dual to $\xi_{I_1}$ that vanishes on $\mathcal{B}_{I_1}$, where $\xi_{I_1}$ contracts. We may integrate by parts the first term in (C.10) and reduce it to a boundary term by using the Maxwell equations,

$$\int_{\mathcal{U}_{\mathcal{H}}} F \wedge G_{\mathcal{H}} = \int_{\partial \mathcal{M}} A_{\mathcal{H}} \wedge \star_5 F - \int_{\partial \mathcal{U}_{I_1}} A_1 \wedge \star_5 F - \mathrm{i}\sqrt{3} \int_{\partial \mathcal{U}_{I_1}} \mathrm{d}(\alpha_1 - \alpha_{\mathcal{H}}) \wedge \star_5 F. \tag{C.12}$$

For the same reason as above, we conclude that

$$\lim_{\varepsilon \to 0} \int_{\partial \mathcal{U}_{I_1}} A_1 \wedge \star_5 F = 0. \tag{C.13}$$

However, $\mathrm{d}(\alpha_1 - \alpha_{\mathcal{H}})$ is not a smooth one-form in $\mathcal{U}_{I_1}$, in particular its norm diverges at the bolt $\mathcal{B}_{I_1}$. As a result, terms involving $\mathrm{d}\alpha$ contribute additional non-trivial corrections to the on-shell action, which would be absent in the case of a globally defined gauge field. Taking this contribution into account, the bulk action simplifies to

$$I_{\text{bulk}} = \frac{1}{48\pi} \int_{\partial \mathcal{M}} A_{\mathcal{H}} \wedge \star_5 F - \frac{\mathrm{i}\sqrt{3}}{48\pi} \int_{\partial \mathcal{U}_{I_1}} \mathrm{d}(\alpha_1 - \alpha_{\mathcal{H}}) \wedge G_{\mathcal{H}} + \mathcal{O}(\varepsilon). \tag{C.14}$$

It is straightforward to show that the contribution from the asymptotic region of the spacetime yields

$$\frac{1}{48\pi} \int_{\partial \mathcal{M}} A_{\mathcal{H}} \wedge \star_5 F = -\frac{\beta \Phi}{3} Q. \tag{C.15}$$

The second term can be computed explicitly in the $\varepsilon \to 0$ limit, giving

$$-\frac{\mathrm{i}\sqrt{3}}{48\pi} \lim_{\varepsilon \to 0} \int_{\partial \mathcal{U}_{I_1}} \mathrm{d}(\alpha_1 - \alpha_{\mathcal{H}}) \wedge G_1 = -\frac{\mathrm{i}\sqrt{3}}{24} \left[ \iota_{\xi_{I_1}} \mathrm{d}(\alpha_1 - \alpha_{\mathcal{H}}) \right] \int_{\mathcal{B}_{I_1}} G_1$$

$$= \frac{\Phi^{I_1}}{3} \mathcal{Q}^{I_1}, \tag{C.16}$$

where to obtain the second line we used (3.92), (C.5), (5.6), (5.20).

We are now left to compute the boundary terms of (C.1) [40]:

$$I_{\text{GHY}} = -\frac{1}{8\pi} \lim_{r \to +\infty} \int_{\partial \mathcal{M}} \mathrm{d}^4 x \, \frac{r^2}{2} \partial_r f^{-1} \sin\theta = \frac{\beta}{\sqrt{3}} Q. \tag{C.17}$$

Putting everything together, we finally obtain that the on-shell action is given by

$$I = \frac{1}{3} \left( \Phi^{I_1} \mathcal{Q}^{I_1} - \varphi Q \right)$$

$$= \frac{\pi}{12\sqrt{3}} \left[ \frac{\varphi^3}{\omega_1 \omega_2} - \frac{\left( p_1 \varphi - \omega_2 \Phi^{I_1} \right)^3}{p_1^2 \omega_2 \left( p_1 \omega_1 + (p_1 - 1) \omega_2 \right)} \right], \tag{C.18}$$

which is consistent with (5.25).

## C.2 Legendre transform

The entropy (5.21), expressed as a microcanonical function of the conserved charges, is related to the on-shell action (5.25) by a Legendre transform. This transformation is performed with respect to the thermodynamic variables $\varphi$ and $\omega_{1,2}$, subject to the constraint $\omega_+ = 2\pi \mathrm{i}$ [12]. According to the quantum statistical relation (5.24), the Legendre transform is implemented by extremizing the following functional:

$$\mathcal{S} = \mathrm{ext}_{\{\varphi, \omega_1, \omega_2, \Lambda\}} \left[ -I - \varphi Q - \omega_1 J_1 - \omega_2 J_2 - \Lambda(\omega_1 + \omega_2 - 2\pi \mathrm{i}) \right], \tag{C.19}$$

where $\Lambda$ is a Lagrange multiplier enforcing the constraint on the angular velocities. Since $I$ is an homogeneous function of degree one with respect to $\varphi$, $\omega_{1,2}$, it follows that

$$\mathcal{S} = 2\pi \mathrm{i}\Lambda. \tag{C.20}$$

The extremization equations

$$Q = -\frac{\partial I}{\partial \varphi}, \qquad J_{1,2} = -\frac{\partial I}{\partial \omega_{1,2}} - \Lambda, \tag{C.21}$$

yield a quadratic formula for $\Lambda$,

$$P_0 + P_1\Lambda + \Lambda^2 = 0\,, \tag{C.22}$$

where the coefficients are given by

$$P_1 = (1-p_1)J_1 + (1+p_1)J_2 + \Phi^{I_1}\left(Q - \frac{\pi}{12\sqrt{3}}\frac{\left(\Phi^{I_1}\right)^2}{p_1^2}\right),$$

$$P_0 = \frac{4(1-p_1)}{\pi}\left(\frac{Q}{\sqrt{3}} + \frac{\pi}{12p_1(1-p_1)}\left(\Phi^{I_1}\right)^2\right)^3 - \left(J_2 - \frac{\pi}{12\sqrt{3}\,p_1^2(1-p_1)}\left(\Phi^{I_1}\right)^3\right)^2 \tag{C.23}$$

$$+\left(J_2 - \frac{\pi}{12\sqrt{3}\,p_1^2(1-p_1)}\left(\Phi^{I_1}\right)^3\right)P_1\,.$$

Solving for $\Lambda$,

$$\Lambda = -\frac{\mathrm{i}}{2}\sqrt{4P_0 - P_1^2} - \frac{P_1}{2}\,, \tag{C.24}$$

we find the entropy

$$\mathcal{S} = 4\sqrt{\pi}\sqrt{(1-p_1)\left(\frac{Q}{\sqrt{3}} + \frac{\pi\,(\Phi^{I_1})^2}{12p_1(1-p_1)}\right)^3 - \frac{\pi}{4}\left((p_1-1)J_- - \frac{\Phi^{I_1}}{2}Q - \frac{\pi(1+p_1)(\Phi^{I_1})^3}{24\sqrt{3}\,p_1^2(1-p_1)}\right)^2}$$

$$-2\pi\mathrm{i}\left[J_+ - p_1 J_- + \frac{\Phi^{I_1}}{2}\left(Q - \frac{\pi}{12\sqrt{3}}\frac{\left(\Phi^{I_1}\right)^2}{p_1^2}\right)\right]. \tag{C.25}$$

Using the explicit expression for the charges (5.10), one can verify that this expression agrees with the Bekenstein-Hawking entropy (5.21).

Finally, the electric flux (5.20) can be determined by differentiating the action with respect to $\Phi^{I_1}$:

$$\mathcal{Q}^{I_1} = \frac{\partial I}{\partial \Phi^{I_1}} = -\frac{\partial \mathcal{S}}{\partial \Phi^{I_1}} = \frac{\pi}{4\sqrt{3}}\frac{\left(p_1\,\varphi - \Phi^{I_1}\omega_2\right)^2}{p_1^2(p_1\omega_1 + (p_1-1)\,\omega_2)}\,. \tag{C.26}$$

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
