# Peer review of "Bubbling saddles of the gravitational index"

_SciPost Physics, doi:SciPost Phys. 19, 134 (2025)_

## Round 1 · Referee Report · Anonymous (Referee 1) · 2025-9-14

Disclosure of Generative AI use

The referee discloses that the following generative AI tools have been used in the preparation of this report:

AI has been used to improve the language of this report.

Strengths

  1. Original contribution: The systematic classification of bubbling saddles and explicit construction of new solutions significantly extends the current literature on gravitational index saddles.

  2. Technical thoroughness: The use of multi-center harmonic functions, rod formalism, and careful action evaluation (with detailed appendices) demonstrates strong technical command.

  3. Relevance: The results are timely and of clear interest to the AdS/CFT and black hole microstate communities. The discussion of contributions to the index is well motivated.

  4. Connections: The work relates known examples (black rings, black lenses, bubbling microstates) to the new families in a coherent framework.

Weaknesses

  1. Physical interpretation of complex saddles: Many saddles are complex Euclidean solutions. While this is acknowledged, the physical role of such saddles and especially the appearance of imaginary on-shell actions for bubbling solutions remains somewhat ambiguous.

  2. Regularity arguments: The justification that singularities can be removed by complexified coordinate changes in the rod formalism is nontrivial.

  3. Accessibility: The presentation is very technical, especially in sections 2–3, and might be difficult for non-specialists to follow.

Report

The paper investigates new semiclassical saddles of the five-dimensional supergravity path integral relevant to the gravitational index. The authors classify bubbling configurations using rod structures, generalizing previously known black hole saddles to include black rings, black lenses, and horizonless bubbling solutions. They compute on-shell actions, study regularity conditions in a complexified setup, and analyze the contributions of these saddles to the gravitational index. The work connects with recent progress in holographic index computations, black hole microstates, and bubbling geometries.

The manuscript makes a solid and original contribution to the understanding of gravitational index saddles in supergravity and is suitable for publication after minor revision.

Requested changes

The only obvious correction I would suggest is replacing “well-definite” with “well-defined.”

In addition, here are some questions and suggestions that the authors may or may not wish to address:

  • The usage of the term “finite temperature” is confusing when referring to the periodicity of a Euclidean circle that cannot be Wick rotated to Lorentzian signature. The underlying solutions do not appear to admit a natural interpretation as thermal horizons, since real time is absent. It may be clearer to use terminology such as “fictitious temperature” or simply “finite circle size.”

  • Is there a meaningful sense in which each class of Euclidean rod structure could be argued to connect smoothly to a Lorentzian solution by contracting the rod or segments of it to a point, or could there be cases where this is obstructed?

  • The admissibility of Euclidean saddles in the gravitational index is presented as an open question. This is somewhat puzzling, since already in the second paragraph of the introduction the authors state that they have defined the boundary conditions for the index. Given their detailed analysis, one would expect a more definitive statement on admissibility, from a purely mathematical perspective. It would be also useful to discuss what changes in a different ensemble (fixing the asymptotic charges).

  • Related to the above, the discussion of a “microscopic evaluation of the index in string theory” is left quite vague. The authors should be able to comment more concretely, in particular on whether the boundary conditions they define are compatible with a string-theoretic setting. In addition, I believe there already exist microscopic calculations for some of the black hole systems under consideration (though perhaps only in the extremal Lorentzian case?). If so, a direct comparison would be very valuable.

Recommendation

Ask for minor revision

  • validity: high
  • significance: high
  • originality: high
  • clarity: good
  • formatting: excellent
  • grammar: excellent

Author:  Davide Cassani  on 2025-09-27  [id 5868]

(in reply to Report 1 on 2025-09-14)

We would like to thank the referee for the careful reading and the useful suggestions. In the following we address the four main remarks in the report.

1 - The usage of the term “finite temperature” is confusing when referring to the periodicity of a Euclidean circle that cannot be Wick rotated to Lorentzian signature. The underlying solutions do not appear to admit a natural interpretation as thermal horizons, since real time is absent. It may be clearer to use terminology such as “fictitious temperature” or simply “finite circle size.”

At page 5, in the third paragraph, we have specified what we mean by finite temperature, emphasizing that the size of the circle, beta, may be complex in addition to being finite.

2 - Is there a meaningful sense in which each class of Euclidean rod structure could be argued to connect smoothly to a Lorentzian solution by contracting the rod or segments of it to a point, or could there be cases where this is obstructed?

This is an interesting question. In the case where there is just one Euclidean horizon rod, it should indeed always be possible to smoothly contract it to a point and in this way obtain a Lorentzian solution. In particular, our eqs. (4.7) and (5.9) show (for the two-center and three-center solutions, respectively) that when the length delta of the horizon rod goes to zero, the inverse temperature beta goes to infinity, meaning that the solution becomes extremal. However, in general a solution may contain multiple horizon rods, in which case we do not have a definite answer. Indeed one should work out the precise map between the length of each of these rods and beta. Studying the bubble equations, which impose consistency conditions between the lengths of the different rods, it should be possible to see whether the different rods can be contracted independently, or only simultaneously. We believe that the second case would give rise to a good Lorentzian solution. Since we feel we do not have a sharp statement to make, we have chosen not to comment on this issue in the text. This may be addressed when we will study the multi-horizon case in detail, which we are planning to do in the future.

3 - The admissibility of Euclidean saddles in the gravitational index is presented as an open question. This is somewhat puzzling, since already in the second paragraph of the introduction the authors state that they have defined the boundary conditions for the index. Given their detailed analysis, one would expect a more definitive statement on admissibility, from a purely mathematical perspective. It would be also useful to discuss what changes in a different ensemble (fixing the asymptotic charges).

The boundary conditions respected by all the solutions we study are indeed those for the index. However, a field configuration extremizing the action and satisfying the assigned boundary conditions (i.e., a candidate saddle) does or does not contribute to the path integral (i.e., it is or is not a true saddle) depending on whether the integration contour passes through it. A precise definition of the integration contour and the determination of the true saddles in the context of the gravitational path integral is a hard question which goes beyond the scope of our work. We have slightly expanded the third paragraph in section 6 emphasizing this issue.

4 - Related to the above, the discussion of a “microscopic evaluation of the index in string theory” is left quite vague. The authors should be able to comment more concretely, in particular on whether the boundary conditions they define are compatible with a string-theoretic setting. In addition, I believe there already exist microscopic calculations for some of the black hole systems under consideration (though perhaps only in the extremal Lorentzian case?). If so, a direct comparison would be very valuable.

The reason why we postpone a concrete comparison with a microscopic evaluation of the index in string theory is that we believe there are some questions to be answered beforehand. The microscopic computations we are aware of rely on indices defined in the near-horizon region of a supersymmetric and extremal solution, which a priori depend on different variables than the index in our paper, which is defined in the asymptotically flat region. A precise map between these two partition functions has not been established yet (some progress has been recently made in arXiv:2501.17909). Since this analysis is quite different from the main focus of our paper, we have decided to postpone it to future work. No changes have been made in the text.

Also, we have replaced “well-definite” with “well-defined” everywhere in the text.

---

## Round 1 · Referee Report · Anonymous (Referee 3) · 2025-9-15

Report

The paper studies supersymmetric solutions of pure five-dimensional supergravity, assuming $U(1)^3$ symmetry. These assumptions allow the authors to describe a solution in terms of a $2d$ orbit space, whose boundary encodes the structure of the fixed locus of the symmetry in terms of a rod structure.
The authors focus on analytic continuations of Lorentzian supersymmetric solutions, and extend the study of the regularity of the latter. After a necessarily technical though well-presented section discussing the regularity properties of general solutions, the authors move to the discussion of two special cases, namely a 2-center solution representing an analytic continuation of the BPS black hole, and a 3-center solution representing an analytic continuation of the BPS black ring/lens.

The analysis is thorough and well-presented, and the paper represents a timely and significant contribution to the literature on complex solutions contributing to the gravitational path integral.

Requested changes

  1. Could the authors please expand on the competition between the different saddles to the gravitational path integral? Is it possible that solutions with different topologies contribute to the same index? If so, what is the expectation based on the action?

  2. Some additional minor points a) is the coordinate $r_a$ used in (2.6) defined in (2.7)? b) is the expression (2.9) still locally defined? If so, where? c) the discussion about the "cap" above (2.17) seems a bit confusing, since later on the authors also discuss solutions without "horizon" rods. Should (2.17) be $\forall a$? d) (2.23) what is $r$? e) p. 14 "the Gibbons-Hawking fibre $\partial_\psi$" perhaps would be more properly characterized as something along the lines of "the vector $\partial_\psi$ along the Gibbons-Hawking fibre" f) p. 16 and p. 42 it should be "a Euclidean" rather than "an Euclidean"

Recommendation

Ask for minor revision

  • validity: -
  • significance: -
  • originality: -
  • clarity: -
  • formatting: -
  • grammar: -

Author:  Davide Cassani  on 2025-09-27  [id 5870]

(in reply to Report 3 on 2025-09-15)

We thank the referee for the interesting comments. In the following we address the points raised in the report, specifying the corresponding changes that we have made in the paper:

1 - “Could the authors please expand on the competition between the different saddles to the gravitational path integral? Is it possible that solutions with different topologies contribute to the same index? If so, what is the expectation based on the action?”

Yes, we expect that solutions with different topologies can in principle contribute to the same index. For a given value of the chemical potentials, the dominant saddle should be determined by comparing their on-shell actions. However, not all possible candidate saddles are expected to contribute to the index, only those satisfying appropriate allowability criteria should compete. Clarifying this likely requires a careful analysis of the integration contour for the path integral. This is a complicated analysis, and goes beyond the scopes of the present work. Therefore, a detailed study of the competition between different topologies should be delayed to future work. We have added a comment on this point in the last paragraph of page 55.

2 - “Some additional minor points a) is the coordinate r_a used in (2.6) defined in (2.7)? b) is the expression (2.9) still locally defined? If so, where? c) the discussion about the "cap" above (2.17) seems a bit confusing, since later on the authors also discuss solutions without "horizon" rods. Should (2.17) be ∀a? d) (2.23) what is r ? e) p. 14 "the Gibbons-Hawking fibre ∂ψ" perhaps would be more properly characterized as something along the lines of "the vector ∂ψ along the Gibbons-Hawking fibre" f) p. 16 and p. 42 it should be "a Euclidean" rather than "an Euclidean”.”

a. Yes, we added a comment to clarify this under (2.6).
b. The one-form is singular on the vertical z-axis. In section 3.2 we showed that this singularity is, however, just a coordinate singularity that can be removed by performing suitable coordinate transformations involving the Euclidean time coordinate. Imposing that the coordinate transformations required to remove the singularity along the whole z-axis are compatible with each other leads to a set of quantization conditions, that we discussed.
c. Yes, Eq. (2.17) (now (2.18)), should be applied to each center, both for “horizon centers” and “bubbling centers”. We added “for each center” to clarify this below (2.18).
d. r is the radial direction in a system of spherical coordinates in R^3 centered at its origin. We added Eq. (2.7) to clarify this.
e. We implemented the suggested change.
f.  We implemented the corrections.

---

## Round 1 · Referee Report · Anonymous (Referee 2) · 2025-9-15

Report

The paper classifies and systematically studies semiclassical saddles of 5D minimal ungauged supergravity with U(1)^3 symmetry. Many of the specific examples that they discuss are novel, and a general framework for analyzing any saddle with U(1)^3 symmetry is presented. The authors generalize the techniques that have been used to classify supersymmetric Lorentzian asymptotically-flat solutions to the case of complex Euclidean saddles, while at the same time following a blueprint that has been successfully used for asymptotically-AdS saddles in the literature. This combination of different techniques is an original contribution of this work, and it opens the door to various new research directions.

The analysis carried out in the paper is very thorough, and well connected to various other works in the literature. I recommend publication, after minor revisions.

Requested changes

1- Minor nitpick: the coordinate r (without the subscript "a" or the tilde) is never properly defined prior to (2.23), where it first appears; even if it is not hard to guess what this coordinate represents, it would be better if it was clarified.

2- Section 3.2 would be more complete if the case of vanishing f and H was commented, even very briefly. In analogy with lemma 1 of arxiv.org/abs/1712.07092, do the authors expect that the non-vanishing of K^2 + H L is sufficient to ensure that the metric is smooth and invertible? Or are situations where f and H vanish censored by the fact that saddles cap off at the horizon and the h_a coefficient are complex?

3- In sections 4 and 5, the authors discuss horizonless solitonic solutions by starting with saddles with a horizon and taking the limit of beta going to zero, and are thus able to reproduce the physical properties of Lorentzian topological solitons. In section 6, the authors mention that it is not clear whether horizonless solutions are true saddles of the grand-canonical index, given that they have purely imaginary action. The question that I have for the authors is the following: could it be that the purely imaginary action is an artifact of the beta-to-zero limit, and a finite-beta soliton may have an action that is not purely imaginary? Or do the authors have an argument against this possibility?

Recommendation

Ask for minor revision

  • validity: high
  • significance: high
  • originality: high
  • clarity: good
  • formatting: perfect
  • grammar: perfect

Author:  Davide Cassani  on 2025-09-27  [id 5869]

(in reply to Report 2 on 2025-09-15)

We thank the referee for their positive assessment of our work, and for their suggestions for improvement. Below we detail the changes we have made in the manuscript and provide answers to address the points raised in the report.

  • Minor nitpick: the coordinate r (without the subscript "a" or the tilde) is never properly defined prior to (2.23), where it first appears; even if it is not hard to guess what this coordinate represents, it would be better if it was clarified.

We have fixed this by introducing all the sets of spherical coordinates used in the paper in page 9.

  • Section 3.2 would be more complete if the case of vanishing f and H was commented, even very briefly. In analogy with lemma 1 of arxiv.org/abs/1712.07092, do the authors expect that the non-vanishing of K^2 + H L is sufficient to ensure that the metric is smooth and invertible? Or are situations where f and H vanish censored by the fact that saddles cap off at the horizon and the h_a coefficient are complex?

It is not clear to us whether critical surfaces where H and f vanish are part of the solutions we study. The answer to this question will certainly depend on the complexification we assume. For instance, in the two-center case analyzed in section 4, one can choose a Euclidean section in which the metric is real, simplifying the regularity analysis. In this case, we show that the positivity of the combination K^2 + H L (which ensures that the metric remains real and positive-definite) is equivalent to requiring the coefficients of the harmonic function H — and thus H itself — to be positive, thereby forbidding the presence of critical surfaces where both H and f vanish. In the more general setting in which the parameters of the harmonic functions can be complex, the presence of such critical surfaces would also require a complexification of the coordinates. Thus, it is not entirely clear to us how the analysis of 1712.07092 should be extended in this context. Nevertheless, since the local form of the solutions is the same as in the Lorentzian case analyzed in 1712.07092, we expect that the non-vanishing of K^2 + H L must be a necessary condition to be imposed so that the metric is finite and invertible at these potential critical surfaces. We have added footnote 9 to mention this issue.

  • In sections 4 and 5, the authors discuss horizonless solitonic solutions by starting with saddles with a horizon and taking the limit of beta going to zero, and are thus able to reproduce the physical properties of Lorentzian topological solitons. In section 6, the authors mention that it is not clear whether horizonless solutions are true saddles of the grand-canonical index, given that they have purely imaginary action. The question that I have for the authors is the following: could it be that the purely imaginary action is an artifact of the beta-to-zero limit, and a finite-beta soliton may have an action that is not purely imaginary? Or do the authors have an argument against this possibility?

As far as we understand there is no way we can obtain an action which is not purely imaginary for the topological solitons. Indeed, before assuming a concrete complexification, the on-shell action and chemical potentials of the finite-beta soliton are generically complex. However, such grand-canonical quantities cannot be associated a priori to a physically relevant Lorentzian solution (note that the Lorentzian solution obtained after Wick rotation is not even real for generic complex values of the chemical potentials). As discussed in detail in section 5.4, there are two limits (beta-to-zero and beta-to-infinity) yielding physically sensible Lorentzian solutions, if a concrete complexification, which turns out to be different for each limit, is assumed.  n the case of the beta-to-zero limit, the complexification we need in order to make sense of the Wick-rotated solution implies that the chemical potentials and the action become purely imaginary.

---

## Round 2 · Referee Report · Anonymous (Referee 3) · 2025-9-29

Report

I would like to thank the authors for addressing the points I raised.
I believe that the paper is ready for publication.

Recommendation

Publish (meets expectations and criteria for this Journal)

---

## Round 2 · Referee Report · Anonymous (Referee 2) · 2025-10-1

Report

The authors have thoroughly addressed all the questions and requested changes. I recommend this paper for publication.

Recommendation

Publish (meets expectations and criteria for this Journal)

---

## Round 2 · Referee Report · Anonymous (Referee 1) · 2025-10-25

Report

I recommend the revised manuscript for publication.

Recommendation

Publish (easily meets expectations and criteria for this Journal; among top 50%)

---

## Round 2 · Author Response

Dear Editor,

We have addressed the remarks of the referees. Details can be found in our response letter to each of the three reports.

---

## Round 2 · List of Changes

In addition to the changes detailed in our response letters to each of the three reports, we have made a few very minor changes in the wording with no impact on the meaning of the corresponding sentences.

---

## Editorial Decision

published